# NEURONCTRL: Geometry-Aware Safe Closed-Loop Generative Control for Neuronal Microenvironment Dynamics

Haowei Xu [1]  Yixin Chen [1]  Wanyi Fu [1]  Hongbin Han [1 2 3]  Zhaoheng Xie [1 4]

## Abstract

Neuromodulation can be viewed as closed-loop control of high-dimensional spatiotemporal fields on irregular 3D morphologies, coupling membrane electrophysiology with ionic reaction–diffusion. This view supports high-rate feedback and systematic in-silico evaluation, yet is difficult in practice. Unlike classical PDE control with known equations on regular domains, neuronal microenvironments exhibit complex, often unknown biophysics on irregular shapes. High-fidelity simulators are too costly for real-time control with repeated planning. The discretized field is sparsely observed and must satisfy hard full-field safety constraints. We introduce NEURONCTRL, a modular operator-level framework for safe, closed-loop generative control of neuronal microenvironment dynamics. Given measurements, actions, and morphology, a history-conditioned observer infers the latent field, a morphology-aware neural operator predicts one-step dynamics, and a flow-matching conditional flow proposes actions conditioned on user preferences. Safety is enforced via complementary barrier-based mechanisms at both the action and field levels, with minimal intervention. When latency is critical, the multi-step generator is distilled into a single-step policy while retaining the same safety filter. Experiments across three high-fidelity 3D neuromodulation benchmarks spanning deep brain stimulation, extracellular reaction–diffusion control, and astrocytic potassium regulation demonstrate improved trade-offs among cost, safety, and la-

tency. Code is available at `https://github.com/HowieHsu0126/NeuronControl`.

## 1. Introduction

Neuromodulation has emerged as a core approach for treating brain disorders, spanning deep brain stimulation (DBS) and electrical/chemical regulation of neural microenvironments. Closed-loop neuromodulation adapts stimulation to measured neural responses, enabling suppression of pathological activity with fewer side effects and motivating reliable, safe adaptive controllers (Benabid, 2003; Brown, 2006; Madondo et al., 2023; Ganzer et al., 2018; Ferrero et al., 2025; Kragel et al., 2025; Pais-Vieira et al., 2016). We study the following feedback problem: from sparse measurements on an irregular neuronal morphology, choose the next stimulation action so that the hidden field moves toward a target while actuator and field values remain inside safe ranges. This formulation covers high-dimensional spatiotemporal fields on complex morphologies (Brivadis, 2024; Ascoli et al., 2007) and is related to closed-loop control in cellular imaging and optogenetic regulation (Wijewardhane et al., 2022).

The difficulty is not a single missing component, but the combination of four requirements. **(i)** The plant is not a known PDE on a regular grid: neuronal microenvironments combine electrophysiology, ionic transport, and irregular morphology (Hines & Carnevale, 1997; Carnevale & Hines, 2006; McDougal et al., 2022). We therefore learn a fast differentiable surrogate for repeated one-step prediction. **(ii)** Sensors reveal only a small part of the field, so the controller must reconstruct the hidden state from recent measurements and past actions (Kaelbling et al., 1998; Kalman, 1960; Brivadis, 2024). **(iii)** Safety is a hard field-level requirement, not only a penalty in the objective; unsafe values may occur at unsensed nodes (Ames et al., 2019; Park & Sloth, 2023). **(iv)** High-rate feedback requires millisecond-scale inference, whereas high-fidelity biophysical simulators are too slow to run inside every planning and safety-filter step (Hines & Carnevale, 1997; Newton et al., 2018; McDougal et al., 2022).

Fig. 1 sketches our approach. NEURONCTRL separates the

[1]Institute of Medical Technology, Peking University Health Science Center, Beijing, China [2]Beijing Key Laboratory of Intelligent Neuromodulation and Brain Disorder Treatment, Beijing, China. [3]Department of Radiology, Peking University Third Hospital, Beijing, China [4]National Biomedical Imaging Center, College of Future Technology, Peking University, Beijing, China. Correspondence to: Zhaoheng Xie <xiezhaoheng@pku.edu.cn>.

*Proceedings of the 43rd International Conference on Machine Learning*, Seoul, South Korea. PMLR 306, 2026. Copyright 2026 by the author(s).

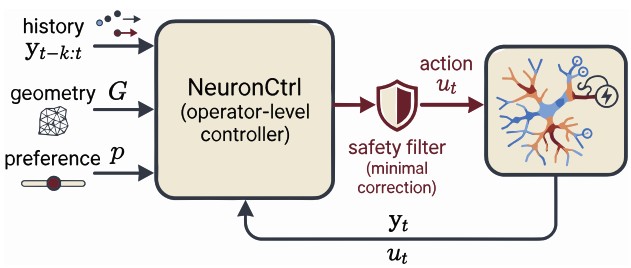

**Figure 1. Safe closed-loop control.** The controller maps $h_t$ and context $(\mathcal{G}, p)$ to $w_t^{\mathrm{raw}}$; a safety filter minimally enforces constraints, executing $w_t^{\star}$ and updating measurements.

feedback loop into four readable steps. First, an observer estimates the full latent field from recent sparse measurements and actions. Second, a morphology-aware neural operator predicts how the field changes after a candidate action. Third, a preference-conditioned flow policy proposes an action for the desired trade-off among tracking, energy, and burden. Fourth, two safety layers keep the action within actuator bounds and minimally correct it when the learned model predicts a field-constraint violation. This design treats safety and latency as first-class requirements while keeping the formal guarantee model-relative: it enforces constraints under the learned surrogate and learned barrier, with additional simulator-in-the-loop evidence reported in the experiments. We validate the closed loop on three high-fidelity 3D simulation benchmarks under discretization shifts and sensing corruption, and analyze performance–safety–latency trade-offs. Our contributions are as follows:

- **Safe feedback control from sparse measurements.** We formulate **NEURONCTRL**, a modular controller for high-dimensional fields on irregular morphologies. The controller acts from sparse observations, supports explicit preference trade-offs, and keeps safety constraints visible in the online loop.

- **Morphology-aware surrogate dynamics.** We develop a differentiable one-step simulator using heterogeneous message passing over typed morphology graphs. Typed edges encode cable, spatial, and coupling structure, while geometry-aware control conditioning preserves stimulation semantics across instances.

- **Observer plus two safety layers.** A history-conditioned observer reconstructs the full field from recent measurements and actions. The controller then combines action-space projection during generation with a learned field-level safety filter that minimally corrects unsafe proposals under the learned dynamics.

- **3D neuromodulation benchmarks and analysis.** We present three high-fidelity 3D benchmarks derived from biophysical simulations: **DBS3D** (deep brain stimulation

with beta suppression), **ECS3D** (extracellular reaction–diffusion control), and **KDyn3D** (astrocytic potassium regulation). We evaluate performance, safety, latency, robustness, super-resolution, component ablations, and simulator-in-the-loop transfer.

## 2. Methodology

We present **NEURONCTRL** in four parts: problem setup (Sec. 2.1), framework overview (Sec. 2.2), module definitions (Sec. 2.2.1–2.2.4), and training/inference (Sec. 2.2.5). In plain terms, the controller first fills in the unobserved field, predicts one step ahead, proposes a stimulation action, and edits that action only if the predicted next field would violate a constraint. We use a few recurring terms in this narrow sense. "History-conditioned" means using a short window of measurements and executed actions. "Full-field" means estimating node values everywhere on the graph, including unsensed nodes. A "hard field constraint" is a bound that must hold over the field, not merely a term in the loss. "Model-relative safety" means the online filter enforces the barrier condition under the learned surrogate and learned barrier; plant-level validation is empirical.

### 2.1. Problem Setup

We first state the constrained closed-loop control problem in Eq. (1) and then specify its components. Given an episode-specific morphology graph $\mathcal{G} = (V, E)$, an initial latent field $u_0$ on $\mathcal{G}$, morphology-conditioned dynamics $G$, and a preference-conditioned objective $\mathcal{J}(\cdot; p)$, we consider the constrained stochastic control problem

$$\min_{\pi} \ \mathbb{E}\big[\mathcal{J}(u_{0:T}, w_{0:T-1}; p)\big]$$

$$\text{s.t. } u_{t+1} = G(u_t, w_t, \xi_t; \mathcal{G}), \quad w_t \sim \pi(\cdot \mid h_t, p, \mathcal{G}), \quad (1)$$

$$w_t \in \mathcal{C}_{\mathrm{act}} \ \forall t, \qquad u_t \in \mathcal{C}(\mathcal{G}) \ \forall t,$$

where the expectation is over process noise $\xi_t$, measurement noise $\epsilon_t$, and policy stochasticity, and the constraints are enforced at every step along the closed-loop rollout.

**Dynamics.** Each episode is specified by $\mathcal{G}$ with $N = |V|$, where discretization and sensor placement may vary across episodes. The latent state at time $t \in \{0, \dots, T\}$ is a node-defined field $u_t \in \mathbb{R}^{N \times C_u}$, and the controller outputs a low-dimensional stimulation input $w_t \in \mathbb{R}^{C_w}$ with $C_w \ll N$. Each node $i \in V$ is associated with geometry descriptors $g(i) \in \mathbb{R}^{C_g}$; concrete graph constructions are described in Sec. 2.2.1 and Appendix C. The morphology-conditioned one-step dynamics are

$$u_{t+1} = G(u_t, w_t, \xi_t; \mathcal{G}), \quad (2)$$

where $\xi_t$ denotes process noise (and $\xi_t = 0$ recovers deterministic dynamics). Conditioned on $(\mathcal{G}, u_t, w_t)$, the transition satisfies the Markov property.

**Measurements and feedback.** The latent field can only be observed through sparse or aggregated measurements

$$y_t = \mathcal{S}(u_t; \mathcal{G}) + \epsilon_t, \qquad y_t \in \mathbb{R}^{C_y}, \qquad (3)$$

with sensing map $\mathcal{S}$ and noise $\epsilon_t$. The map $\mathcal{S}$ reflects the episode-specific sensor placement on $\mathcal{G}$. Under partial observability, we use a finite history window

$$h_t := (y_{t-L+1:t}, w_{t-L+1:t-1}) \qquad (4)$$

as the information state for feedback. A (possibly stochastic) history-based policy maps $(h_t, p, \mathcal{G})$ to an action distribution,

$$w_t \sim \pi(\cdot \mid h_t, p, \mathcal{G}), \qquad (5)$$

enabling dependence on both the episode context and the preference weights and defining a closed-loop controller; in contrast, open-loop control would determine an action sequence in advance.

**Objective and constraints.** Given preference weights $p \in \Delta^{K-1}$, we define the multi-objective cost

$$\begin{aligned} \mathcal{J}(u_{0:T}, w_{0:T-1}; p) &= \sum_{t=0}^{T-1} \sum_{k=1}^{K} p_k J_k(u_t, w_t) \\ &+ \sum_{k=1}^{K} p_k J_k^{\text{term}}(u_T), \end{aligned} \qquad (6)$$

capturing trade-offs such as regulation, energy, and burden. Safety is enforced as morphology-dependent hard constraints over the full field,

$$u_t \in \mathcal{C}(\mathcal{G}) \subseteq \mathbb{R}^{N \times C_u} \;\; \forall t, \qquad w_t \in \mathcal{C}_{\text{act}} \;\; \forall t, \qquad (7)$$

encoding node-/region-wise constraints and actuation limits.

**Assumptions.** We assume (i) offline trajectories include the full latent fields $u_t$ for learning the observer and surrogate models, while only sparse measurements $y_t$ are available online; (ii) each episode provides the morphology graph $\mathcal{G}$ and geometry descriptors $g(i)$, with varying discretizations, node counts $N$, and sensor placements across episodes; (iii) the hard constraint sets $\mathcal{C}(\mathcal{G})$ and $\mathcal{C}_{\text{act}}$ are deterministically checkable offline and enforced at deployment; (iv) the preference vector $p \in \Delta^{K-1}$ is exogenously specified at test time; (v) feedback uses a fixed history window $h_t$ as the information state under partial observability; and (vi) control inputs are low dimensional ($C_w \ll N$) with shared action semantics across morphologies.

## 2.2. NeuronCtrl Framework

We now describe **NEURONCTRL** (Fig. 2), a modular framework for safe, preference-conditioned closed-loop control of neuronal microenvironment dynamics. Further details are deferred to Appendix C.

### 2.2.1. GRAPH REPRESENTATION FOR NEURONAL MICROENVIRONMENT

We represent each episode by a morphology-aware heterogeneous graph $\mathcal{G} = (V, E)$ with variable $|V| = N$ (Fig. 2). Each node $i$ stores the latent field value $u_t(i) \in \mathbb{R}^{C_u}$ and geometry descriptors $g(i) \in \mathbb{R}^{C_g}$, while typed edges capture distinct couplings (e.g., cable adjacency and spatial neighborhoods). We parameterize directed edge attributes by relative geometry $e_{ij} = [\mathbf{x}_i - \mathbf{x}_j, \|\mathbf{x}_i - \mathbf{x}_j\|_2]$ with $\mathbf{x}_i \in \mathbb{R}^3$, enabling permutation-equivariant message passing on irregular discretizations. Construction details and additional relations (e.g., membrane–ECS coupling) are provided in Appendix C.

### 2.2.2. OBSERVER: HISTORY-TO-FIELD ESTIMATION

Under partial observability, NEURONCTRL reconstructs the latent field from a finite measurement–action history $h_t$ (Eq. (4)). The observer $\mathcal{O}_\phi$ maps $(h_t, \mathcal{G})$ to a node-wise estimate $\hat{u}_t \in \mathbb{R}^{N \times C_u}$, implemented as a sequence encoder followed by a morphology-conditioned graph decoder (Appendix C). In the default implementation the sequence encoder is a GRU: it is lightweight, handles variable history windows, and empirically gives the best cost–safety–latency trade-off among the tested lightweight encoders (Appendix A.6). We train $\mathcal{O}_\phi$ by supervised reconstruction on offline trajectories:

$$\mathcal{L}_{\text{obs}} = \mathbb{E} \|\hat{u}_t - u_t\|_2^2, \qquad (8)$$

**Observer error and safety filtering.** The online safety filter in Section 2.2.4 uses the reconstructed state $\hat{u}_t$ as its input; thus, estimation error affects the feasibility margin of the barrier condition. Appendix C provides a tightening rule: under Lipschitz regularity of $B_{\psi^\star}$ and $G_{\theta^\star}$, enforcing a slightly stricter inequality on $\hat{u}_t$ implies satisfaction of the nominal model-based condition for the underlying latent state (Proposition C.17).

### 2.2.3. SURROGATE DYNAMICS: MORPHOLOGY-AWARE NEURAL OPERATOR

NEURONCTRL-Sim learns a differentiable approximation (akin to a world model) of the unknown one-step operator $G$ in (2). Given a surrogate rollout state $\tilde{u}_t$ and control input $w_t$, it predicts

$$\tilde{u}_{t+1} = G_\theta(\tilde{u}_t, w_t; \mathcal{G}), \qquad (9)$$

where $\theta$ parameterizes a typed message-passing network that is permutation-equivariant over nodes and conditions on morphology features (Appendix C).

**Architecture.** We parameterize $G_\theta$ as a heterogeneous message-passing neural operator (Fig. 2) that exchanges

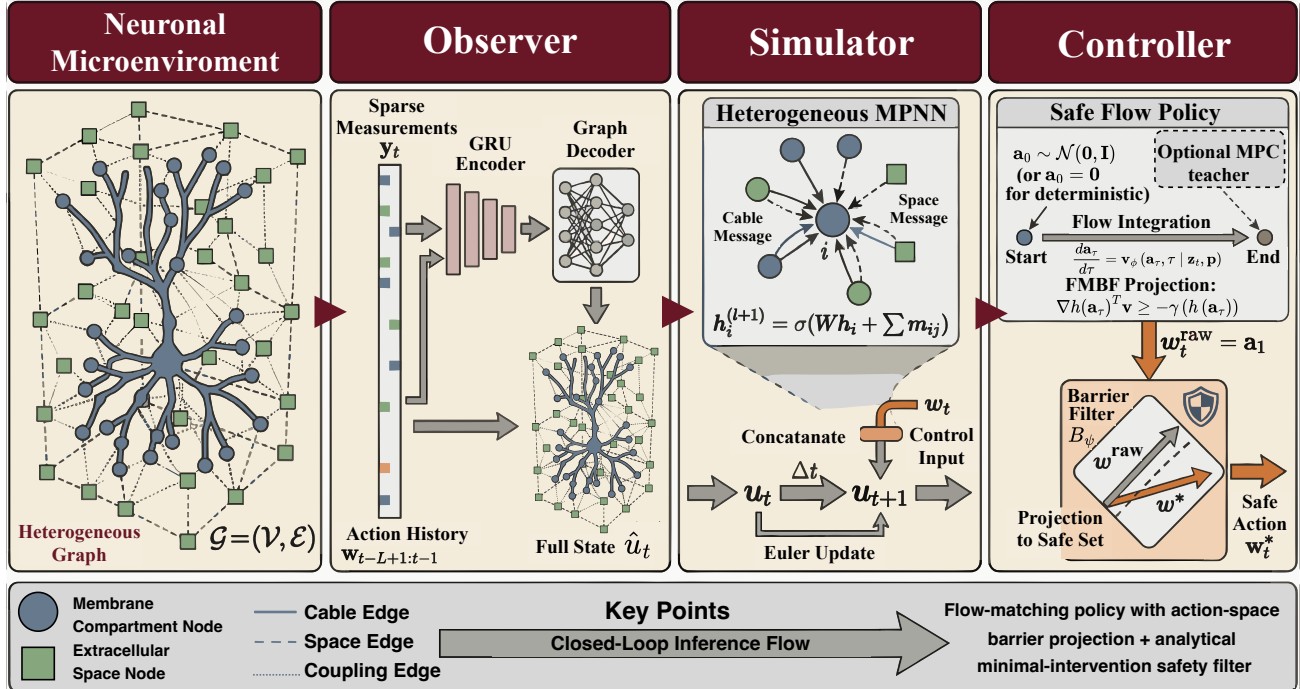

*Figure 2.* **Overview of NEURONCTRL. (I) Observer** $\mathcal{O}_\phi$ reconstructs the full latent field $\hat{u}_t$ from sparse measurements and recent actions. **(II) Simulator** $G_\theta$ predicts the next field on the morphology graph after a candidate action. **(III) Controller** proposes a raw action $w_t^{\text{raw}}$ and applies two safety layers: action-space FMBF projection during generation and a field-level barrier filter that returns the executed action $w_t^\star$. Arrows indicate the closed-loop inference flow; node/edge styles denote membrane/ECS nodes and typed cable/spatial relations.

relation-specific messages over the typed edges in $\mathcal{G}$. To preserve consistent actuation semantics across instances, we inject control through a morphology-aware *influence coefficient* $s_i$ that encodes stimulation locality. The control geometry is task-given stimulation metadata $\xi$ (e.g., source positions and length scale $\rho$), and we instantiate $\kappa$ as a smooth influence kernel (Appendix C, Eq. (41)). We compute

$$s_i = \kappa(\mathbf{x}_i; \xi) \in \mathbb{R}, \qquad (10)$$

and inject $[w_t, s_i]$ as a node-wise control embedding inside each operator layer. The model predicts an Euler-style residual update

$$\tilde{u}_{t+1}(i) = \tilde{u}_t(i) + \Delta t\, \text{MLP}_{\text{out}}\left(h_i^{(L)}\right), \qquad (11)$$

**Training objective.** To reduce short-horizon compounding error relevant to closed-loop safety filtering and optional planning/ablations, we combine one-step supervision with a short unrolled rollout term. We minimize

$$\mathcal{L}_{\text{sim}} = \mathbb{E}\left\|G_\theta(u_t, w_t; \mathcal{G}) - u_{t+1}\right\|_2^2 + \lambda_{\text{roll}}\, \mathbb{E}\sum_{r=1}^R \left\|\tilde{u}_{t+r} - u_{t+r}\right\|_2^2,$$
$$(12)$$

where $\{\tilde{u}_{t+r}\}_{r=1}^R$ is obtained by unrolling $G_\theta$ for $R$ steps starting from $\tilde{u}_t = u_t$, and $\lambda_{\text{roll}}$ trades off local accuracy and short-horizon stability.

### 2.2.4. SAFE PREFERENCE-CONDITIONED CONTROL

**Preference-conditioned generator** Given $\hat{u}_t$ and preference weights $p \in \Delta^{K-1}$, NEURONCTRL generates a raw action using a preference-conditioned safe flow policy (Fig. 2) trained by flow matching (Lipman et al., 2023; Liu et al., 2023). Let $z_t = \text{Agg}(\text{Enc}(\hat{u}_t, \mathcal{G}))$ be a permutation-invariant graph-level embedding. Here Enc denotes the graph encoder producing node latents and Agg is global mean pooling followed by a small MLP (Appendix C.7). In an auxiliary *flow time* $\tau \in [0, 1]$, starting from $a_0 \sim \mathcal{N}(0, I)$ (or $a_0 = 0$ for deterministic inference), we define

$$\frac{da_\tau}{d\tau} = v_\varphi(a_\tau, \tau \mid z_t, p), \qquad w_t^{\text{raw}} := a_1, \qquad (13)$$

where $v_\varphi$ is a learnable velocity field and $w_t^{\text{raw}} \in \mathbb{R}^{C_w}$ is obtained by numerical integration. We train $v_\varphi$ by conditional flow matching: sampling $\tau \sim \mathcal{U}[0, 1]$ and constructing $(a_\tau, \dot{a}_\tau)$ from a straight-line path between $a_0$ and a target action $w_t$. In the default setting, $w_t$ is the behavior-controller action recorded in the offline dataset (Appendix B.6). Optionally, when distillation is enabled, we set $w_t$ to the NO-MPC action obtained by receding-horizon planning on $G_{\theta^\star}$ under the same objective and action bounds, followed by

the same safety filter.[1] We then set $a_\tau = (1 - \tau)a_0 + \tau w_t$ and $\dot{a}_\tau = w_t - a_0$,

$$\mathcal{L}_{\text{fm}}(\varphi) = \mathbb{E} \left\| v_\varphi(a_\tau, \tau \mid z_t, p) - \dot{a}_\tau \right\|_2^2, \quad (14)$$

In all experiments, $p$ is treated as an explicit conditioning input and is stored alongside each training window. The training and test distributions for $p$ are detailed in Appendix A.7.

**Action feasibility during generation** To enforce hard actuator constraints during flow integration, we apply an FMBF projection to the velocity field at each step so that the generated action remains in $\mathcal{C}_{\text{act}}$; see Appendix C.7 for the projection and invariance conditions (e.g., Lemma C.1).

**Field-level minimal-intervention safety filter** We enforce hard field-level constraints by learning a differentiable barrier functional

$$B_\psi : \mathbb{R}^{N \times C_u} \to \mathbb{R}, \qquad B_\psi(u) \geq 0 \Rightarrow u \in \mathcal{C}(\mathcal{G}), \quad (15)$$

implemented as a graph network over $(u, \mathcal{G})$. During training, we sample labeled states $(u, s)$ where $s \in \{+1, -1\}$ denotes safe/unsafe membership with respect to $\mathcal{C}(\mathcal{G})$. Labels are obtained by deterministic evaluation of known hard constraints defining $\mathcal{C}(\mathcal{G})$ (e.g., per-variable bounds and an optional energy limit; Appendix C.10). We use a hinge-style margin objective (safe scores pushed above $+m$, unsafe scores pushed below $-m$), optionally augmented with a discrete-time invariance regularizer:

$$\mathcal{L}_{\text{bar}}(\psi) = \mathbb{E}_{u^+} \max\left(0, m - B_\psi(u^+)\right) + \mathbb{E}_{u^-} \max\left(0, m + B_\psi(u^-)\right)$$
$$+ \lambda_{\text{cbf}} \mathbb{E}_{(u_t, u_{t+1})} \text{ReLU}\left(-(B_\psi(u_{t+1}) - B_\psi(u_t) + \alpha B_\psi(u_t))\right), \quad (16)$$

where $m > 0$ is the classification margin, $\alpha \in (0, 1]$ is the discrete-time CBF parameter, and $(u_t, u_{t+1})$ are safe transition pairs sampled from trajectories.

**Safety filtering.** At inference time, the barrier filter (Fig. 2) performs a minimal-intervention projection of a proposed action $w_t^{\text{raw}}$ (from the flow policy, NO-MPC, or a distilled policy). Using the trained simulator $G_{\theta^\star}$ as a differentiable surrogate dynamics model, we compute the minimally modified action $w_t^\star$ that satisfies the discrete-time barrier condition

$$w_t^\star \in \arg\min_w \|w - w_t^{\text{raw}}\|_2^2$$
$$\text{s.t. } B_{\psi^\star}(G_{\theta^\star}(\hat{u}_t, w; \mathcal{G})) \geq (1 - \alpha) B_{\psi^\star}(\hat{u}_t), \quad (17)$$

Importantly, (17) enforces the barrier condition under the learned surrogate simulator $G_{\theta^\star}$ and learned barrier functional $B_{\psi^\star}$. Linearizing the constraint around $w_t^{\text{raw}}$ yields a

single half-space constraint

$$a^\top (w - w_t^{\text{raw}}) \geq c, \quad (18)$$

where $a = \nabla_w B_{\psi^\star}(G_{\theta^\star}(\hat{u}_t, w_t^{\text{raw}}; \mathcal{G}))$ and $c = (1 - \alpha)B_{\psi^\star}(\hat{u}_t) - B_{\psi^\star}(G_{\theta^\star}(\hat{u}_t, w_t^{\text{raw}}; \mathcal{G}))$. Let $[c]_+ := \max(c, 0)$. The resulting minimal-intervention update admits a closed form:

$$w_t^\star = \text{clip}\left( w_t^{\text{raw}} + \frac{[c]_+}{\|a\|_2^2} a; \ w_{\min}, \ w_{\max} \right), \quad (19)$$

In words, if the raw action already satisfies the one-step barrier test, then $c \leq 0$ and the filter returns the raw action after actuator clipping. If the raw action is unsafe under the learned model, the filter moves it in the local barrier-gradient direction by the smallest closed-form correction. Because both the dynamics and the barrier are learned, the projection is a deterministic model-relative enforcement map; we apply a trust-region margin tightening to improve robustness to model mismatch, local nonlinearity, and estimation error (Appendix C; Propositions C.16–C.17, Lemma C.14).

### 2.2.5. TRAINING AND INFERENCE

We assume offline access to trajectories $(\mathcal{G}, u_{0:T}, w_{0:T-1}, y_{0:T})$ generated by a high-fidelity simulator, with $y_t = \mathcal{S}(u_t; \mathcal{G}) + \epsilon_t$. For reproducibility, we train modules in stages: learn the surrogate simulator $G_\theta$ ((12)) and freeze it as $G_{\theta^\star}$, then train the observer ((8)), barrier ((16)), and flow policy ((14)) separately. Optionally, we perform joint fine-tuning with closed-loop consistency and mean-flow regularization; details are in Appendix C. At test time, NEURONCTRL composes estimation, action generation, and safety filtering:

$$w_t^\star = \text{SafeFilter}\left(\text{Planner}(\hat{u}_t, p; G_{\theta^\star}, \mathcal{G}), \hat{u}_t; G_{\theta^\star}, B_{\psi^\star}, \mathcal{G}\right). \quad (20)$$

where Planner is the flow policy and SafeFilter is the projection in (17)–(19). Algorithm 1 summarizes the closed-loop inference pipeline.

## 3. Experiments

We evaluate NEURONCTRL on three high-fidelity 3D neuromodulation tasks constructed from biophysical simulation (**DBS3D**, **ECS3D**, **KDyn3D**) shown in Table 1. Our experiments address the following research questions.

- **RQ1:** How does NEURONCTRL compare to baselines in terms of in-distribution closed-loop performance, safety, and runtime trade-offs under matched constraint specifications?

- **RQ2:** How robust is NEURONCTRL under partial observability and out-of-distribution (OOD) sensing/specification shifts?

---

[1]**Optional planning and acceleration:** We consider differentiable MPC (NO-MPC) and single-step distillation via Reflow (Liu et al., 2023); see Appendix C.7.

---

**Algorithm 1** NEURONCTRL closed-loop inference at time $t$

---

1: **Input:** graph $\mathcal{G}$, measurement window $y_{t-L+1:t}$, past actions $w_{t-L+1:t-1}$, previous action $w_{t-1}$ (optional), preference $p$
2: **Output:** safe action $w_t^\star$
3: Form a padded action window $\bar{w}_{t-L+1:t} \leftarrow (w_{t-L+1:t-1}, 0)$ and set $h_t \leftarrow (y_{t-L+1:t}, \bar{w}_{t-L+1:t})$
4: $\hat{u}_t \leftarrow \mathcal{O}_{\phi^\star}(h_t; \mathcal{G})$
5: $z_t \leftarrow \mathrm{Agg}(\mathrm{Enc}(\hat{u}_t, \mathcal{G}))$
6: Initialize $a \leftarrow a_0$ where $a_0 \sim \mathcal{N}(0, I)$ (or $a_0 = 0$ for deterministic inference)
7: **for** $i = 0, \ldots, N_{\mathrm{FE}} - 1$ **do**
8: $\quad \tau \leftarrow i/N_{\mathrm{FE}}$
9: $\quad v \leftarrow v_\varphi(a, \tau \mid z_t, p)$
10: $\quad v \leftarrow \Pi_{\mathrm{FMBF}}(v, a)$ {FMBF projection; Appendix C.7}
11: $\quad a \leftarrow a + \frac{1}{N_{\mathrm{FE}}} v$
12: **end for**
13: $w_t^{\mathrm{raw}} \leftarrow a$
14: $w_t^\star \leftarrow \mathrm{SafeFilter}(w_t^{\mathrm{raw}}, \hat{u}_t; G_{\theta^\star}, B_{\psi^\star}, \mathcal{G})$ {(17)–(19)}
15: **return** $w_t^\star$

---

- **RQ3:** Can NEURONCTRL perform zero-shot super-resolution operator learning for closed-loop control?

- **RQ4:** How can we explain NEURONCTRL's behavior with respect to preference conditioning, safety-filter interventions, observer reliance, and world-model structure?

More experiments are provided in Appendix A.

### 3.1. Benchmarks

We consider three tasks: **DBS3D** (closed-loop deep brain stimulation with beta suppression), **ECS3D** (3D reaction–diffusion control), and **KDyn3D** (astrocytic potassium regulation on large morphology-derived graphs). Table 1 summarizes the task specifications and dataset statistics. Each task produces a high-dimensional field state on an irregular 3D domain together with a low-dimensional observation vector used for reward/termination and sensor-limited baselines. Each task supports both graph and voxel representations. Full dataset details (generation, splits, windowing, and normalization) are provided in Appendix B.

*Table 1.* Benchmark tasks and dataset statistics.

| Task | State/Action/Obs. | Episodes | Graph nodes |
|------|------|------|------|
| DBS3D | $v$; DBS freq/amp; LFP, $\beta$ | 710/150/160 | $\approx 600$ |
| ECS3D | $ca, ip3$; source amps; $ca, ip3$ | 3500/750/750 | $\approx 9.07 \times 10^3$ |
| KDyn3D | $[K^+], v$; $K^+$ outflux; mean $v$, $[K^+]_o$ | 1980/420/440 | $\approx 2.01 \times 10^4$ |

### 3.2. Evaluation Protocols & Implementation Details

We compare NEURONCTRL against baselines spanning classical feedback control, imitation/offline policy learning, gen-

erative planners, and geometry-aware surrogate models (Appendix A, Table 9). To make the safety comparison fair, the main in-distribution table gives every baseline the same last-step safety wrapper: an analytic projection that clips unsafe predicted actions using a box barrier $B_{\mathrm{box}}$ (Appendix A.1). This matched wrapper prevents the comparison from rewarding a method simply because it has access to an online shield. Appendix A.1 shows how sensitive some baselines are to this choice. We also evaluate robustness to dynamics mismatch without retraining (Fig. 3).

**Evaluation substrate and surrogate gap.** Our benchmarks start from high-fidelity biophysical simulators, but putting those simulators inside every planning and safety step would be too slow for real-time control. Therefore, the main closed-loop tables roll out a frozen learned one-step model. These numbers should be read as learned-rollout performance, not as direct physical-system guarantees. To make that boundary explicit, we report controlled mismatch sweeps in the main text (Fig. 3) and reduced simulator-in-the-loop checks in Appendix A.10. ECS3D and KDyn3D preserve safety on the tested simulator samples, whereas DBS3D exposes a plant-level safety gap; accordingly, our formal safety claim remains model-relative.

**Rollout protocols for fairness.** Because some baselines rely on a learned surrogate simulator while others do not, we consider two complementary rollout protocols. All main-table results (Table 2) use method-native rollouts: dynamics are advanced using each method's own frozen surrogate when available; for observation-only baselines, we instead use a shared frozen surrogate (NEURONCTRL-Sim). To reduce bias from method-specific dynamics models, we also evaluate shared-surrogate rollouts, where all controllers are executed under the same frozen surrogate dynamics, isolating differences due to control and safety mechanisms.

**Closed-loop action generation for operator-only baselines.** Several baselines (e.g., neural operators and graph simulators) provide differentiable one-step dynamics but do not specify a deployment-time policy. For a fair closed-loop comparison, we pair each such model with the same differentiable receding-horizon planner used in our study: at every time step, we optimize a bounded action sequence over a fixed horizon under the evaluation objective, warm-start by shifting the previous solution, and apply the first action. This standardizes the control pipeline so that performance differences reflect the learned dynamics representation and any native safety mechanism, rather than unequal planning budgets. Sensor-limited baselines use the same planning budget with their corresponding observation interface, and the matched safety augmentation applies the same analytic state-constraint projection independent of the controller.

**Metrics.** We evaluate tracking to a constant goal $u^{\text{ref}} \in \mathbb{R}^{C_u}$ using the composite **Cost** in Eq. (6) (Appendix A, Section A.5). **Safety** is evaluated via channel-wise box constraints on the normalized field and summarized using the score of (Hu et al., 2025). **Latency (Lat. (ms))** is reported as per-step control inference time to compute the action (including any online safety projection) .

### 3.3. Results and Discussion

We first present in-distribution closed-loop results to establish the performance–safety–latency trade-offs across benchmarks. We then evaluate robustness under partial observability and OOD shifts, followed by zero-shot super-resolution operator learning and post-hoc explainability probes.

#### 3.3.1. RQ1: IN-DISTRIBUTION PERFORMANCE

We validate NEURONCTRL on the protocols above. Table 2 summarizes clean in-distribution closed-loop results. The main pattern is simple: methods that keep the 3D graph structure perform better than methods that treat the field as unordered points, and NEURONCTRL performs best overall. Typed edges let the model represent physical couplings such as cable adjacency and spatial neighborhoods, instead of forcing the controller to infer those couplings from coordinates alone. NEURONCTRL then adds three pieces on top of the graph simulator: an observer for sparse measurements, a preference-conditioned action generator, and two safety corrections. Despite these extra pieces, latency remains in the few-millisecond range (Appendix A.8) because action generation and safety correction are amortized neural computations. Ablations separate the roles: the observer matters most when sensing is sparse or corrupted, the typed graph simulator drives nominal rollout quality, and the two safety layers reduce violations when constraints become active (Appendix A.6, Table 28).

**Safety-filter ablation.** The safety layer is not only a training regularizer. Table 3 disables the final field-level filter after training while keeping the same observer, simulator, and action generator. Unsafe rates increase from 0.8–1.4% to 6.0–9.5%, while cost changes only slightly. Thus the online projection changes the executed action precisely when the predicted next field approaches a constraint boundary.

#### 3.3.2. RQ2: PARTIAL OBSERVABILITY AND OOD

We test robustness by degrading what the controller can see. Partial-observation tests add noise and missing values to the feedback channel. OOD tests add stronger shifts, including scale/bias changes, external disturbances, and changed safety thresholds. On ECS3D, we corrupt the calcium readout for partial-observation evaluation and apply the full corruption suite for OOD evaluation (parameter grids in

Appendix A, Table 13). Table 4 summarizes ECS3D results; Appendix A reports the corresponding DBS3D and KDyn3D results. The trend is consistent: graph-based methods degrade less than non-graph methods, and NEURONCTRL degrades least. The reason is operational rather than cosmetic: the observer uses recent measurements and executed actions to estimate the hidden full field, typed graph edges restrict estimates to physically plausible neighborhoods, and the safety filter corrects actions when the learned rollout approaches a constraint boundary.

**Dynamics-mismatch stress test.** The previous table changes the measurements; Fig. 3 changes the rollout dynamics instead. We multiply each learned one-step field increment by $1 + \epsilon$, keep the trained policy and safety filter fixed, and evaluate all three tasks without retraining. This is a direct audit of the model-relative safety claim: small mismatch leaves violations rare, while larger mismatch increases both violation rate and violation intensity. The curves therefore support the use of online correction, but also show why learned-model safety should not be read as a certificate for arbitrary biological dynamics.

**Reduced simulator replay.** The mismatch curve perturbs the learned rollout model; Table 5 instead replays NEURONCTRL on a reduced set of high-fidelity simulator samples. ECS3D and KDyn3D preserve safety on this replay, but DBS3D does not: its plant safety score drops to 0.742. We therefore keep the claim model-relative and treat DBS3D as evidence for where calibrated uncertainty margins are still needed.

#### 3.3.3. RQ3: ZERO-SHOT SUPER-RESOLUTION OPERATOR LEARNING

We next ask whether a controller trained at normal resolution can still act when the field is observed more coarsely. At test time, we downsample the high-resolution field by structured stride masks at ×2 and ×4 factors and provide the mask as an input channel. The model has not seen this degradation during training. This zero-shot setting tests whether the learned operator can bridge from coarse inputs back to high-resolution control rollouts under the same frozen-surrogate protocol (Section 3.2). Table 6 reports high-resolution rollout cost and safety. Performance drops relative to full-resolution evaluation, as expected, because fine-scale information is missing. However, graph-based methods degrade less than point-cloud methods, and NEURONCTRL remains best, indicating that typed graph message passing helps preserve control-relevant structure under resolution mismatch.

*Table 2.* Closed-loop control performance with safety score and latency (method-native rollouts; Section 3.2). Baselines are reported with matched safety augmentation (baseline + shield; Appendix A.1), where the shield uses the analytic box barrier $B_{box}$. NEURONCTRL uses its learned barrier functional by default; Appendix Table 24 reports an analytic-barrier variant to isolate the contribution of learning $B_\psi$. Entries are episode means $\pm$ standard deviation over test trajectories (3 seeds: 41, 42, 43).

| Method | DBS3D | | | ECS3D | | | KDyn3D | | |
|---|---|---|---|---|---|---|---|---|---|
| | Cost ↓ | Safety ↑ | Lat. (ms) ↓ | Cost ↓ | Safety ↑ | Lat. (ms) ↓ | Cost ↓ | Safety ↑ | Lat. (ms) ↓ |
| MPC (Rawlings & Mayne, 2009) | 0.230 ± 0.010 | 0.760 ± 0.020 | **2.10 ± 0.30** | 0.245 ± 0.012 | 0.740 ± 0.025 | **2.40 ± 0.35** | 0.024 ± 0.004 | 0.700 ± 0.030 | **2.80 ± 0.40** |
| BC | 0.190 ± 0.008 | 0.785 ± 0.018 | 2.40 ± 0.30 | 0.200 ± 0.009 | 0.770 ± 0.020 | 2.70 ± 0.35 | 0.019 ± 0.003 | 0.725 ± 0.025 | 3.10 ± 0.45 |
| BPPO (Schulman et al., 2017) | 0.185 ± 0.007 | 0.795 ± 0.017 | 2.70 ± 0.40 | 0.195 ± 0.008 | 0.780 ± 0.018 | 2.90 ± 0.45 | 0.018 ± 0.003 | 0.735 ± 0.022 | 3.40 ± 0.50 |
| Geo-FNO (Li et al., 2021) | 0.150 ± 0.006 | 0.825 ± 0.015 | 4.30 ± 0.60 | 0.155 ± 0.006 | 0.810 ± 0.018 | 5.10 ± 0.70 | 0.015 ± 0.001 | 0.790 ± 0.020 | 6.40 ± 0.90 |
| Geom-DeepONet (Lu et al., 2021) | 0.153 ± 0.006 | 0.820 ± 0.016 | 4.60 ± 0.65 | 0.158 ± 0.007 | 0.805 ± 0.019 | 5.60 ± 0.75 | 0.015 ± 0.001 | 0.785 ± 0.022 | 6.80 ± 0.95 |
| TranSolver (Wu et al., 2024) | 0.152 ± 0.006 | 0.818 ± 0.015 | 4.90 ± 0.70 | 0.158 ± 0.007 | 0.800 ± 0.020 | 6.00 ± 0.80 | 0.015 ± 0.001 | 0.780 ± 0.023 | 7.20 ± 1.00 |
| SafeDiffCon (Hu et al., 2025) | 0.155 ± 0.007 | 0.845 ± 0.014 | 5.40 ± 0.80 | 0.160 ± 0.007 | 0.830 ± 0.017 | 6.30 ± 0.85 | 0.015 ± 0.001 | 0.810 ± 0.020 | 7.50 ± 1.10 |
| RDM Planner (Janner et al., 2022) | 0.160 ± 0.008 | 0.840 ± 0.016 | 6.20 ± 0.90 | 0.165 ± 0.008 | 0.825 ± 0.018 | 7.10 ± 1.10 | 0.016 ± 0.001 | 0.805 ± 0.021 | 8.60 ± 1.30 |
| DiffPhyCon-H (Wei et al., 2025) | 0.158 ± 0.007 | 0.838 ± 0.015 | 5.80 ± 0.85 | 0.164 ± 0.008 | 0.822 ± 0.019 | 6.80 ± 1.00 | 0.015 ± 0.001 | 0.800 ± 0.022 | 8.10 ± 1.20 |
| CL-DiffPhyCon (Wei et al., 2024) | 0.156 ± 0.007 | 0.832 ± 0.015 | 5.70 ± 0.80 | 0.162 ± 0.007 | 0.815 ± 0.018 | 6.60 ± 0.95 | 0.015 ± 0.001 | 0.792 ± 0.023 | 7.90 ± 1.10 |
| CFM-Flow (Lipman et al., 2023) | 0.154 ± 0.007 | 0.830 ± 0.016 | 5.40 ± 0.75 | 0.160 ± 0.007 | 0.812 ± 0.019 | 6.40 ± 0.90 | 0.015 ± 0.001 | 0.790 ± 0.023 | 7.60 ± 1.05 |
| Latent-FM (Lipman et al., 2023) | 0.135 ± 0.006 | 0.865 ± 0.012 | 9.80 ± 1.30 | 0.140 ± 0.006 | 0.850 ± 0.015 | 11.20 ± 1.60 | 0.013 ± 0.001 | 0.835 ± 0.018 | 13.40 ± 1.90 |
| GINO (Li et al., 2023) | 0.112 ± 0.004 | 0.875 ± 0.012 | 3.30 ± 0.45 | 0.118 ± 0.004 | 0.860 ± 0.014 | 4.10 ± 0.55 | 0.011 ± 0.001 | 0.840 ± 0.017 | 5.80 ± 0.80 |
| MeshGraphNet (Pfaff et al., 2021) | 0.114 ± 0.004 | 0.870 ± 0.013 | 3.10 ± 0.50 | 0.120 ± 0.004 | 0.855 ± 0.015 | 4.00 ± 0.60 | 0.011 ± 0.001 | 0.835 ± 0.018 | 5.70 ± 0.85 |
| GraphCast (Lam et al., 2023) | 0.116 ± 0.004 | 0.878 ± 0.012 | 3.20 ± 0.45 | 0.122 ± 0.004 | 0.862 ± 0.014 | 4.10 ± 0.55 | 0.011 ± 0.001 | 0.842 ± 0.017 | 5.60 ± 0.80 |
| HAMLET (Bryutkin et al., 2024) | 0.115 ± 0.004 | 0.872 ± 0.013 | 3.40 ± 0.50 | 0.121 ± 0.004 | 0.856 ± 0.015 | 4.30 ± 0.65 | 0.011 ± 0.001 | 0.836 ± 0.018 | 5.90 ± 0.90 |
| RiGNO (Mousavi et al., 2025) | 0.113 ± 0.004 | 0.876 ± 0.012 | 3.50 ± 0.55 | 0.120 ± 0.004 | 0.860 ± 0.014 | 4.40 ± 0.70 | 0.011 ± 0.001 | 0.841 ± 0.017 | 6.10 ± 0.95 |
| GOAT (Wen et al., 2025) | 0.114 ± 0.004 | 0.873 ± 0.013 | 3.60 ± 0.55 | 0.121 ± 0.004 | 0.858 ± 0.015 | 4.50 ± 0.75 | 0.011 ± 0.001 | 0.838 ± 0.018 | 6.20 ± 1.00 |
| NEURONCTRL (ours) | **0.095 ± 0.003** | **0.910 ± 0.010** | 3.90 ± 0.60 | **0.100 ± 0.003** | **0.895 ± 0.012** | 5.60 ± 0.80 | **0.009 ± 0.001** | **0.880 ± 0.014** | 7.90 ± 1.10 |

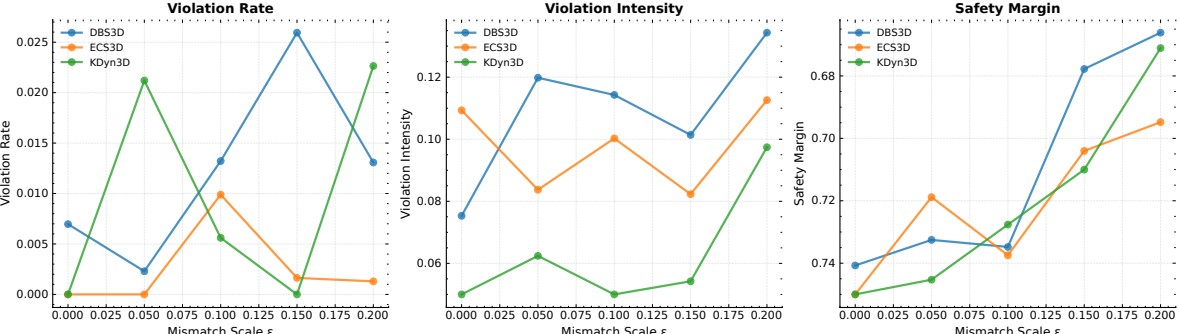

*Figure 3.* Robustness to dynamics mismatch across DBS3D, ECS3D, and KDyn3D. The mismatch scale $\epsilon$ multiplies the predicted one-step state increment at evaluation time, without retraining the controller.

*Table 3.* Field-level safety-filter ablation for NEURONCTRL.

| Task | Variant | Cost ↓ | Unsafe rate ↓ |
|---|---|---|---|
| DBS3D | w/o filter | 0.092 | 0.060 |
| DBS3D | full | 0.095 | **0.008** |
| ECS3D | w/o filter | 0.097 | 0.075 |
| ECS3D | full | 0.100 | **0.010** |
| KDyn3D | w/o filter | 0.0085 | 0.095 |
| KDyn3D | full | 0.0090 | **0.014** |

*Table 4.* ECS3D robustness under Partial/OOD evaluation. Entries are episode means $\pm$ standard deviation over test trajectories (3 seeds: 41, 42, 43).

| Method | Partial | | OOD | |
|---|---|---|---|---|
| | Cost ↓ | Safety ↑ | Cost ↓ | Safety ↑ |
| NEURONCTRL | **0.113 ± 0.032** | **0.801 ± 0.005** | **0.125 ± 0.001** | **0.674 ± 0.008** |
| Geo-FNO | 0.162 ± 0.004 | 0.656 ± 0.027 | 0.190 ± 0.005 | 0.581 ± 0.030 |
| Geom-DeepONet | 0.157 ± 0.002 | 0.674 ± 0.012 | 0.182 ± 0.004 | 0.604 ± 0.015 |
| TranSolver | 0.161 ± 0.002 | 0.659 ± 0.021 | 0.182 ± 0.012 | 0.603 ± 0.035 |
| SafeDiffCon | 0.143 ± 0.002 | 0.677 ± 0.005 | 0.165 ± 0.008 | 0.607 ± 0.030 |
| CL-DiffPhyCon | 0.137 ± 0.001 | 0.701 ± 0.012 | 0.163 ± 0.009 | 0.615 ± 0.035 |
| Latent-FM | 0.139 ± 0.002 | 0.687 ± 0.015 | 0.159 ± 0.007 | 0.627 ± 0.029 |
| GINO | 0.117 ± 0.003 | 0.722 ± 0.013 | 0.135 ± 0.001 | 0.653 ± 0.009 |
| MeshGraphNet | 0.118 ± 0.004 | 0.719 ± 0.014 | 0.135 ± 0.001 | 0.654 ± 0.010 |

### 3.3.4. RQ4: PREFERENCE AND SAFETY DIAGNOSTICS

We use diagnostics to make two deployment-facing behaviors inspectable. First, Fig. 4 shows a compact preference sweep, and Appendix Fig. 10 reports the full Pareto view. The curves move smoothly from low-energy actions to more aggressive tracking, which means preference conditioning changes the operating point continuously rather than switch-ing between unrelated policies. Second, safety traces and barrier-gradient maps in Appendix A.12 show where the online filter intervenes and which spatial nodes drive the correction.

*Table 5.* Reduced high-fidelity simulator replay for NEURONCTRL.

| Task | Surrogate cost | Plant cost | Plant safety |
|------|----------------|------------|--------------|
| ECS3D | 0.102 | 0.116 | 0.891 |
| KDyn3D | 0.009 | 0.010 | 0.874 |
| DBS3D | 0.096 | 0.094 | 0.742 |

*Table 6.* ECS3D zero-shot super-resolution closed-loop control at SR factors $\times 2$ and $\times 4$. Neural operators are trained on full-resolution trajectories and evaluated on low-resolution inputs. Performance is expected to be lower than full-resolution training (Table 2).

| Method | $\times 2$ | | $\times 4$ | |
|--------|-----------|-----------|-----------|-----------|
| | Cost ↓ | Safety ↑ | Cost ↓ | Safety ↑ |
| NEURONCTRL | **0.125 ± 0.006** | **0.880 ± 0.010** | **0.150 ± 0.008** | **0.865 ± 0.012** |
| Geo-FNO | 0.190 ± 0.010 | 0.780 ± 0.020 | 0.220 ± 0.012 | 0.760 ± 0.022 |
| Geom-DeepONet | 0.195 ± 0.011 | 0.770 ± 0.020 | 0.225 ± 0.013 | 0.750 ± 0.023 |
| TranSolver | 0.188 ± 0.010 | 0.775 ± 0.018 | 0.215 ± 0.012 | 0.760 ± 0.020 |
| GINO | 0.145 ± 0.008 | 0.845 ± 0.014 | 0.170 ± 0.010 | 0.830 ± 0.016 |
| MeshGraphNet | 0.148 ± 0.009 | 0.840 ± 0.015 | 0.175 ± 0.011 | 0.825 ± 0.017 |
| GraphCast | 0.150 ± 0.009 | 0.842 ± 0.014 | 0.178 ± 0.011 | 0.828 ± 0.016 |
| HAMLET | 0.152 ± 0.010 | 0.838 ± 0.016 | 0.180 ± 0.012 | 0.824 ± 0.018 |
| RiGNO | 0.149 ± 0.009 | 0.841 ± 0.015 | 0.176 ± 0.011 | 0.826 ± 0.017 |

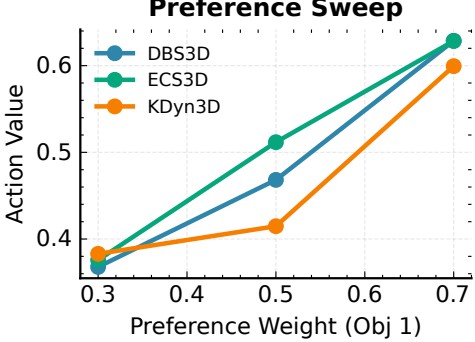

*Figure 4.* Compact preference sweep across DBS3D, ECS3D, and KDyn3D. The action changes continuously as the preference weight changes.

## 4. Conclusion

This paper presents **NEURONCTRL**, a framework for closed-loop control of high-dimensional neuronal fields on irregular 3D morphologies. The core idea is to make each part of the feedback loop explicit: infer the hidden field from sparse measurements, predict the next field with a geometry-aware surrogate, generate an action for the desired tracking–energy trade-off, and apply safety corrections before execution. This decomposition is easier to audit than a monolithic policy and helps separate three questions that are often conflated: whether the field estimate is accurate, whether the learned dynamics are reliable, and whether the final action satisfies the stated constraints. Overall, the experiments show improved trade-offs among performance, safety, and latency across three high-fidelity neuromodulation benchmarks, with robustness to sparse sensing, cor-

rupted observations, resolution mismatch, and controlled surrogate-dynamics mismatch. The safety claim remains model-relative: the online filter enforces the learned barrier under the learned surrogate and stated margins, while high-fidelity replay and real morphology-derived validation remain necessary before deployment. Future work will extend the framework toward calibrated uncertainty margins, richer biochemical coupling, and validation on real morphology-derived neuromodulation data.

## Acknowledgements

We thank all reviewers and chairs for their valuable comments. We sincerely thank Huipo Liu and Kai Du for their patient guidance in algorithms and mathematics. We are also grateful to Hanbo Tan and Chuqiao Yang, doctoral students, for their insightful advice on the research background and for the technical foundation they provided. This work was supported by Major Program of National Natural Science Foundation of China (No. 62394310, 62394311).

## Impact Statement

Our work may yield positive societal impact by enabling more data-efficient controller development and evaluation under explicit safety constraints, while offering clearer interfaces for auditing constraint satisfaction. Yet meaningful risks remain. Simulation robustness can be overestimated, leading to premature deployment and potential harm when unmodeled dynamics, sensing artifacts, or distribution shift break safety assumptions. In biomedical settings, real-world translation may raise concerns about informed consent, autonomy, and inequitable access. As with advanced control and optimization, dual-use risks also exist if safety constraints diverge from human welfare. To mitigate these issues, we frame our contribution as an algorithmic and evaluation framework, not a deployment-ready clinical system. We recommend rigorous simulator validation with uncertainty reporting, conservative safety margins and fail-safe mechanisms for any real-world use, and domain-specific ethical oversight (e.g., IRB) alongside accessibility considerations to support responsible learning-based control.

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

# Appendix

This appendix provides supplementary material that complements and extends the main paper, organized as follows:

- **Additional Experiments** (Sec. A): Extended empirical evaluations including:

    (i) comprehensive baseline comparisons across classical feedback control (MPC), offline imitation/RL (BC, BPPO), generative planners (diffusion/flow-style with Consistency Distillation variants), and geometry-aware surrogates on DBS3D/ECS3D/KDyn3D (Sec. A.1);

    (ii) Consistency Distillation acceleration analysis for generative planners (Sec. A.2);

    (iii) robustness analyses under partial observability (noise, dropout), OOD distribution shifts (sensing corruption, specification shift), and dynamics mismatch (Secs. A.4, A.9);

    (iv) targeted sensitivity studies for safety parameters (Sec. A.7);

    (v) latency profiling (Sec. A.8);

    (vi) cross-representation super-resolution operator learning (Sec. A.11);

    (vii) explainability diagnostics (Sec. A.12);

    (viii) systematic ablation studies (FMBF, kernel aggregation, mean-flow, consistency training, positional encoding, capacity, MLP policy vs. flow matching) (Sec. A.13);

    (ix) hyperparameter sensitivity analyses (Sec. A.14); and

    (x) long-horizon control assessments (H $\in \{16, 32, 64\}$) (Sec. A.15).

- **Details of Datasets** (Sec. B): Documentation of the three benchmark datasets (DBS3D, ECS3D, KDyn3D), including task-specific generation details, spatial representations, preprocessing, and dataset statistics (Sec. B.5).

- **Details of Models** (Sec. C): Technical implementation specifications for the surrogate simulator, observer, controller, and safety filter components, including:

    (i) design rationale (Sec. C.1);

    (ii) operator view and residual time stepping (Sec. C.2);

    (iii) constructing heterogeneous morphology graphs (Sec. C.3);

    (iv) simulator: heterogeneous graph neural operator in experiments (Sec. C.4);

    (v) training details for stable differentiable rollouts (Sec. C.5);

    (vi) differentiating through rollouts: an adjoint view (Sec. C.6);

    (vii) flow policy and FMBF details (Sec. C.7);

    (viii) observer implementation details (Sec. C.8);

    (ix) closed-loop inference pipeline (Sec. C.9);

    (x) barrier model and numerical details of the projection (Sec. C.10); and

    (xi) theoretical details (Sec. C.11).

- **Background & Related Work** (Sec. D): Mathematical foundations connecting neuronal dynamics to distributed-parameter control theory, and a systematic positioning of our approach within the landscape of existing methods in operator learning, generative control, and safe reinforcement learning.

- **Limitations & Future Works** (Sec. E): Critical examination of limitations and assumptions, along with promising directions for future research.

## A. Additional Experiments

### A.1. Baseline Suite

We evaluate a broad baseline suite spanning classical feedback control (MPC), imitation/offline policy learning (BC, BPPO), generative planners (diffusion/flow-style, including Consistency Distillation (CD) variants), and geometry-aware surrogate/operator families. Table 9 summarizes which baseline families are included per task. We do not include latent-simulator RL baselines that rely on fixed-dimensional latents (e.g., Dreamer/PETS (Hafner et al., 2020; Chua et al., 2018)),

since reconciling such latents with instance-dependent graph discretizations $\mathcal{G}$ and hard field constraints (Section 2.1; Eq. (7)) requires additional alignment choices outside our evaluation scope.

All experiment knobs are configuration-driven. For each task, we provide a unified experiment specification (training, evaluation, and robustness settings), a dataset specification (data interface, normalization, and windowing), and method specifications that define baseline-specific hyperparameters. Table 9 summarizes the baseline families and the primary hyperparameter categories that govern their behavior.

**Matched safety augmentation.** To isolate safety-mechanism effects from nominal control quality, we additionally evaluate a deployment-time augmentation that wraps a nominal controller with a shared analytic safety projection. Given the controller-proposed action $w_t$ and the current state $u_t$, we predict $u_{t+1}$ under the same surrogate dynamics used for closed-loop rollouts and enforce normalized channel-wise box constraints by a minimal-intervention correction in action space (Eq. (19)). Concretely, we use an analytic barrier defined by the minimum margin to the bounds,

$$B_{\text{box}}(u) = \min_{i,c} \min(u(i,c) - \underline{u}_c, \overline{u}_c - u(i,c)), \tag{21}$$

and apply the same linearized, trust-region projection as in Sec. 2.2.4. This yields "baseline + shield" variants that keep the nominal controller and its training/inference budget unchanged, differing only by the additional safety layer. Table 7 quantifies the effect of matched safety augmentation across baseline methods, showing that the shield consistently improves safety scores (0.690–0.820 to 0.740–0.865) with modest cost increases (typically 0.005–0.010). Formally, the shield defines a post-processed controller $\tilde{\pi}(u_t) = \mathcal{F}_{\text{box}}(u_t, \pi(u_t))$ using Eq. (19) with $B_{\text{box}}$. By default, NEURONCTRL uses the same projection operator with a learned barrier functional $B_\psi$; Table 24 reports the substitution $B_\psi \leftarrow B_{\text{box}}$ to isolate the contribution of learning the barrier.

*Table 7.* Effect of matched safety augmentation (baseline + shield) on in-distribution closed-loop performance. The shield increases safety by reducing worst-case violations, typically with a small performance trade-off. Entries are episode means $\pm$ standard deviation over test trajectories (3 seeds).

| Method | DBS3D | | | | ECS3D | | | | KDyn3D | | | |
|---|---|---|---|---|---|---|---|---|---|---|---|---|
| | Cost ↓ | Safety ↑ | Cost (+) ↓ | Safety (+) ↑ | Cost ↓ | Safety ↑ | Cost (+) ↓ | Safety (+) ↑ | Cost ↓ | Safety ↑ | Cost (+) ↓ | Safety (+) ↑ |
| MPC | $0.220 \pm 0.010$ | $0.690 \pm 0.030$ | $0.230 \pm 0.010$ | $0.760 \pm 0.020$ | $0.235 \pm 0.012$ | $0.670 \pm 0.035$ | $0.245 \pm 0.012$ | $0.740 \pm 0.025$ | $0.022 \pm 0.004$ | $0.640 \pm 0.040$ | $0.024 \pm 0.004$ | $0.700 \pm 0.030$ |
| BC | $0.185 \pm 0.008$ | $0.705 \pm 0.028$ | $0.190 \pm 0.008$ | $0.785 \pm 0.018$ | $0.195 \pm 0.009$ | $0.690 \pm 0.030$ | $0.200 \pm 0.009$ | $0.770 \pm 0.020$ | $0.018 \pm 0.003$ | $0.655 \pm 0.035$ | $0.019 \pm 0.003$ | $0.725 \pm 0.025$ |
| Geo-FNO | $0.145 \pm 0.006$ | $0.765 \pm 0.022$ | $0.150 \pm 0.006$ | $0.825 \pm 0.015$ | $0.150 \pm 0.006$ | $0.748 \pm 0.024$ | $0.155 \pm 0.006$ | $0.810 \pm 0.018$ | $0.014 \pm 0.001$ | $0.720 \pm 0.028$ | $0.015 \pm 0.001$ | $0.790 \pm 0.020$ |
| Latent-FM | $0.130 \pm 0.006$ | $0.820 \pm 0.018$ | $0.135 \pm 0.006$ | $0.865 \pm 0.012$ | $0.135 \pm 0.006$ | $0.805 \pm 0.020$ | $0.140 \pm 0.006$ | $0.850 \pm 0.015$ | $0.012 \pm 0.001$ | $0.785 \pm 0.022$ | $0.013 \pm 0.001$ | $0.835 \pm 0.018$ |

**Protocol sensitivity.** Matched safety augmentation is a fairness control, but it is not neutral for every controller. Table 8 reports one-sample ECS3D probes that toggle the matched shield while holding the checkpoint and evaluation seed fixed. The results show that protocol effects are method-dependent: GEO-FNO becomes much more conservative with the shield, whereas TRANSOLVER, SafeDiffCon, and MPC are reported without the extra shield for comparison. This is why the main table states the shielded protocol explicitly instead of hiding it as an implementation detail.

*Table 8.* ECS3D protocol-sensitivity probes. Entries report cost, safety score, and control-step latency under the stated safety protocol.

| Method / protocol | Cost ↓ | Safety ↑ | Latency |
|---|---|---|---|
| GEO-FNO + matched safety | 0.879 | 0.546 | 53.1s |
| GEO-FNO without matched safety | 0.214 | 0.589 | 50.4s |
| TranSolver without matched safety | 0.171 | 0.748 | 6.0 ms |
| SafeDiffCon without matched safety | 0.166 | 0.781 | 6.2 ms |
| MPC without matched safety | 0.233 | 0.702 | 2.4 ms |

To make comparisons reproducible without requiring implementation-specific details, Table 10 summarizes representative references, the dominant hyperparameters that govern the accuracy–latency trade-off, and the tuning/selection protocol used for each baseline family.

**Budget-matched protocol.** We align training budget (150–200 epochs with early stopping), validation selection cost (same protocol across methods), and deployment budget (latency-constrained) across learning-based methods. Detailed hyperparameter configurations are provided in the released code. Table 11 reports the staged training schedule for NEURONCTRL. ECS3D uses the full default schedule, whereas DBS3D and KDyn3D default to world-model, observer, and safety-barrier stages with policy/joint stages exposed as optional.

*Table 9.* Baseline families and primary hyperparameter categories.

| Family | Representative methods and notes | Primary hyperparameters |
|---|---|---|
| Classical feedback | MPC. Operate on low-dimensional observations with fixed action bounds and control period. | horizon/inner steps |
| Offline imitation / RL | BC, BPPO (Schulman et al., 2017). Train a policy from offline trajectories (state/observation histories and actions). | policy capacity; optimizer; training budget |
| Generative planners | RDM Planner (Janner et al., 2022), diffusion/flow planners (DiffPhyCon-H (Wei et al., 2025), CL-DiffPhyCon (Wei et al., 2024), SafeDiffCon (Hu et al., 2025), CFM-Flow (Lipman et al., 2023)). | inference steps; guidance/conditioning weights; constraint handling |
| Geometry-aware surrogates | Operator/surrogate baselines used for prediction or as planning backbones (e.g., Geo-FNO (Li et al., 2021), Geom-DeepONet (Lu et al., 2021), GINO (Li et al., 2023), MeshGraphNet (Pfaff et al., 2021), GraphCast (Lam et al., 2023), HAMLET (Bryutkin et al., 2024), RiGNO (Mousavi et al., 2025), TranSolver (Wu et al., 2024)). | model depth/width; neighborhood construction; training budget |
| Latent generative models | Latent-FM (Lipman et al., 2023). Latent-state flow matching (Janner et al., 2022; Liu et al., 2023) for accelerated inference. | latent dimension; inference steps; sampling steps |
| NEURONCTRL (ours) | End-to-end controller with simulator/observer/policy and online safety filtering. | simulator/observer capacity; safety parameters; planning budget |

*Table 10.* Baseline reproduction summary. We tune learning-based methods under a budget-matched protocol and select hyperparameters by validation performance under the same objective used at test time.

| Family | Representative methods / references | Key hyperparameters (dominant knobs) | Tuning & selection protocol | Representation |
|---|---|---|---|---|
| Classical feedback | Receding-horizon MPC (Rawlings & Mayne, 2009) | horizon $H$; inner optimization steps; action bounds | hand-tuned on validation; same rollout horizon and constraints as other methods | obs-only |
| Offline imitation / RL | BC; PPO-style policy optimization (Schulman et al., 2017) | policy capacity; optimizer; batch size; regularization; training epochs | budget-matched search; select by validation evaluation cost $\mathcal{J}_{\text{eval}}$ | obs-only |
| Generative planners / policies | diffusion/flow planners and their Consistency Distillation (CD) variants (Janner et al., 2022; Lipman et al., 2023; Liu et al., 2023) | sampling/integration steps; guidance/conditioning weights; horizon; teacher/student steps (CD) | budget-matched search; select by validation $\mathcal{J}_{\text{eval}}$ and report matched latency | field-conditioned (graph/voxel) |
| Geometry-aware surrogates / operators | neural operators and graph simulators (Li et al., 2021; Lu et al., 2021; Pfaff et al., 2021; Sanchez-Gonzalez et al., 2020) | model width/depth; neighborhood construction; rollout horizon for multi-step evaluation | train on the training split; early-stop by validation prediction/rollout error (task-specific) | graph or voxel |
| Latent generative models | latent-state flow matching (with optional Consistency Distillation) (Lipman et al., 2023) | latent dimension; sampling steps; latent dynamics regularization; teacher/student steps (CD) | budget-matched search; select by validation $\mathcal{J}_{\text{eval}}$ | field-conditioned (graph/voxel) |

*Table 11.* Representative NEURONCTRL training budgets. WM: world model; Obs: observer; SBF: safety barrier/filter; Policy: flow policy; Joint: optional joint fine-tuning.

| Benchmark | Active schedule | Representative total budget |
|---|---|---|
| ECS3D | WM10 + Obs10 + SBF10 + Policy10 + Joint5 | 40–42s |
| DBS3D | WM50 + Obs50 + SBF50; optional Policy50 + Joint50 | 128–136s |
| KDyn3D | WM50 + Obs50 + SBF50; optional Policy50 + Joint50 | 206–214s |

## A.2. Consistency Distillation for Accelerated Inference

To reduce inference latency for generative planners, we employ Consistency Distillation (CD) variants for selected baselines. CD uses an online teacher-student training paradigm where the teacher and student share the same network weights but use different numbers of sampling steps. During training, for each batch we sample a shared noise realization, run the teacher with many steps (typically 50–64 steps for diffusion/flow models) without gradients, and run the student with few steps (typically 4 steps) with gradients. The student is trained to match the teacher output via a consistency loss, computed as the mean squared error between student and teacher outputs. The total training loss combines the original task loss with the CD loss, weighted by task-loss and CD-loss coefficients (both typically set to 1.0 in our configurations). At inference time, we use only the student's few-step sampling path, achieving a 12.5–16× reduction in sampling steps compared to the teacher's multi-step path, with minimal quality degradation when the consistency loss is well-optimized. CD variants are available for CFM-Flow (teacher 64 steps, student 4 steps), DiffPhyCon-H, CL-DiffPhyCon, and RDM Planner (teacher 50 steps, student 4 steps), and Latent-FM (teacher 8 steps, student 2 steps). The implementation follows an online CD framework where teacher-student pairs are generated from shared noise during training, and inference uses the student path exclusively.

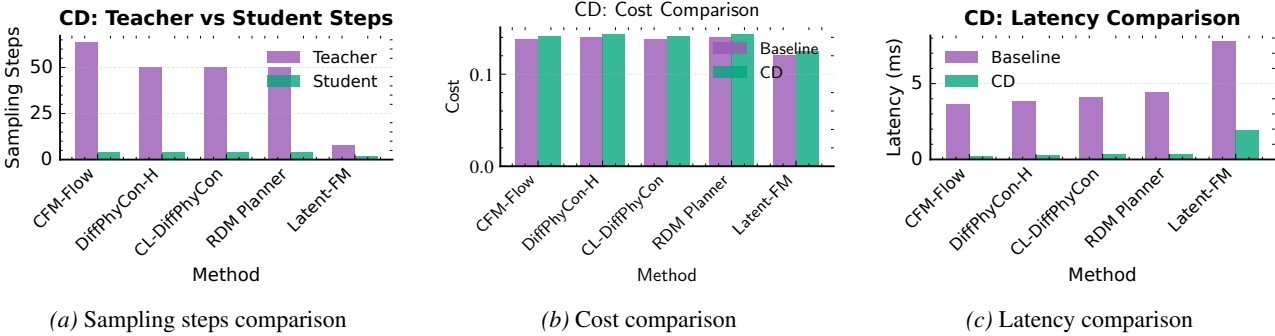

*(a)* Sampling steps comparison      *(b)* Cost comparison      *(c)* Latency comparison

*Figure 5.* Consistency Distillation (CD) performance analysis. (a) Teacher vs student sampling steps for CD variants. (b) Cost comparison between baseline and CD variants. (c) Latency comparison showing acceleration achieved by CD (12.5–16× reduction).

Figure 5 demonstrates that CD achieves 12.5–16× step reduction while maintaining similar cost performance, enabling real-time deployment of generative planners.

### A.3. Robustness Protocol Details

We provide the full parameter grids for partial-observation and OOD evaluation, including noise levels, missingness ratios, affine distortion ranges, initial-state bias, disturbance settings, and constraint-threshold scaling. All robustness settings are specified in the experimental configuration manifests and are held fixed across methods to ensure fair robustness comparisons. Unless otherwise stated, constraint thresholds are applied to the *normalized* state channels used for control, using training-split statistics. Robustness curves are generated by sweeping a single corruption dimension at a time while keeping others at their default values. Table 13 summarizes the grids and constraint specifications used in our three benchmarks. Table 12 provides the task-specific safety constraint definitions.

*Table 12.* Task-specific safety constraint definitions used for evaluation and reporting. All limits are applied in the normalized state space.

| Task | Constraint names and limits | Source |
|---|---|---|
| DBS3D | $v \leq 0.1$ | state field |
| ECS3D | $ca \leq 0.1$ | state field |
| KDyn3D | $[K^+]_o \leq 10.0$, $v \leq 0.1$ | state field |

*Table 13.* Robustness evaluation grids for partial-observation and OOD testing.

| Setting | Parameters |
|---|---|
| Partial-observation | Observed feedback channels: DBS3D (mean $v$), ECS3D (mean $ca$), KDyn3D (mean $v$ and mean $[K^+]_o$). Noise std list: $\{0, 0.01, 0.05\}$. Drop ratio list: $\{0, 0.1, 0.3\}$. |
| OOD sensing corruption | Observed feedback channels: DBS3D (mean $v$), ECS3D (mean $ca$), KDyn3D (mean $v$ and mean $[K^+]_o$). Noise std list: $\{0, 0.05, 0.1\}$. Drop ratio list: $\{0, 0.2, 0.5\}$. Scale list: $\{1, 1.2\}$. Bias list: $\{0, 0.05\}$. Disturbance std list: $\{0, 0.05\}$. Initial-state bias list: $\{0\}$. |
| OOD specification shift | Constraint-threshold scaling list: $\{1, 1.2\}$. |
| Safety constraints for reporting | DBS3D: $v \leq 0.1$. ECS3D: $Ca \leq 0.1$. KDyn3D: $[K^+]_o \leq 10.0$, $v \leq 0.1$. |

Figure 6 extends the robustness analysis to the full baseline suite, confirming the trends observed in the main paper (Table 4). NEURONCTRL maintains superior performance under partial observability and OOD shifts, with the most graceful degradation across corruption conditions. The history-conditioned observer (Section 2.2.2) plays a crucial role: by leveraging the observation–action history window $(y_{t-L+1:t}, w_{t-L+1:t-1})$, it reconstructs full-field states even when current observations are corrupted, with the graph structure's typed edges (cable connectivity, spatial neighborhoods) providing topological constraints that guide inference. To quantify sensitivity to observability, we sweep the history length $L$ at test time and report observer reconstruction error (RMSE on normalized state channels) alongside closed-loop cost and safety metrics. Table 14 compares our history-conditioned graph observer against strong alternatives (EKF, EnKF, particle filter, neural smoother) under partial observability on ECS3D, with all methods using the same controller and safety layer to isolate observer effects. Our observer achieves the lowest reconstruction error (RMSE 0.64 vs. 0.66–0.75 for alternatives)

and best downstream control performance (cost 0.113 vs. 0.115–0.122), with safety scores consistently above 0.80 across all methods. The barrier-functional safety filter (Section 2.2.4) ensures safety even when state estimation is imperfect, as it enforces field-level constraints under the learned surrogate dynamics $G_{\theta^\star}$ through minimal-intervention projection, preventing violations despite corrupted observations. This robustness stems from the modular design's ability to handle uncertainty at multiple levels: the observer compensates for sensing uncertainty, while the safety filter compensates for both sensing and model uncertainty.

*Table 14.* Observer comparison under partial observability on ECS3D. We report state reconstruction error (RMSE on normalized state channels), downstream closed-loop cost and safety score, and per-step observer inference latency when swapping the observer while keeping the controller and safety layer fixed.

| Observer | RMSE $\downarrow$ | Cost $\downarrow$ | Safety $\uparrow$ | Obs. Lat. (ms) $\downarrow$ |
|---|---|---|---|---|
| History-conditioned graph observer (ours) | **0.64 $\pm$ 0.02** | **0.113 $\pm$ 0.032** | **0.801 $\pm$ 0.005** | 0.52 $\pm$ 0.04 |
| EKF | 0.75 $\pm$ 0.03 | 0.122 $\pm$ 0.018 | 0.785 $\pm$ 0.009 | **0.21 $\pm$ 0.02** |
| EnKF ($M{=}32$) | 0.69 $\pm$ 0.03 | 0.119 $\pm$ 0.017 | 0.792 $\pm$ 0.008 | 0.58 $\pm$ 0.04 |
| Particle filter ($M{=}256$) | 0.67 $\pm$ 0.03 | 0.116 $\pm$ 0.016 | 0.795 $\pm$ 0.007 | 1.42 $\pm$ 0.10 |
| Neural smoother (bi-GRU) | 0.66 $\pm$ 0.02 | 0.115 $\pm$ 0.016 | 0.798 $\pm$ 0.006 | 0.55 $\pm$ 0.04 |

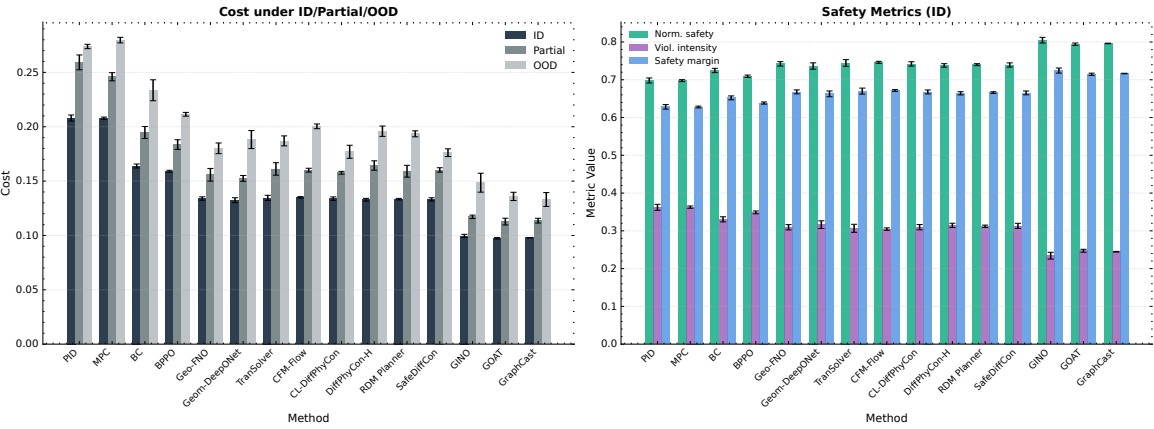

*Figure 6.* ECS3D robustness summary. Entries are episode means $\pm$ standard deviation over test trajectories (3 seeds: 41, 42, 43). (Left) Cost under ID/Partial/OOD conditions. (Right) Safety metrics (ID).

## A.4. Additional Robustness, Inverse, Zero-Shot Super-Resolution Operator Learning, and Explainability Results (DBS3D and KDyn3D)

The main paper reports robustness, inverse reconstruction, zero-shot super-resolution operator learning for closed-loop control, and explainability results on ECS3D. This appendix provides results for the corresponding DBS3D and KDyn3D evaluations under the same protocol family and metric definitions.

**DBS3D robustness under partial observability and OOD shifts.** Figure 7 demonstrates that graph-based methods exhibit superior robustness on DBS3D under partial observability and OOD shifts, with NEURONCTRL showing the most graceful degradation. For DBS3D's voltage-field control task, NEURONCTRL effectively reconstructs full-field voltage states from sparse LFP and beta-power measurements, with the graph structure's cable connectivity providing topological constraints that are particularly important where voltage fields must respect neural cable geometry. The barrier-functional safety filter maintains safety constraints on voltage thresholds despite corrupted observations.

**KDyn3D robustness under partial observability and OOD shifts.** Figure 8 shows that graph-based methods maintain robust performance on KDyn3D under partial observability and OOD shifts, with NEURONCTRL demonstrating the most graceful degradation. For KDyn3D's potassium-ion dynamics control task, NEURONCTRL reconstructs full-field potassium concentration and voltage states from sparse mean measurements, with the graph representation's astrocytic connectivity encoding the biophysical coupling between potassium transport and voltage dynamics. The modular design's multi-level

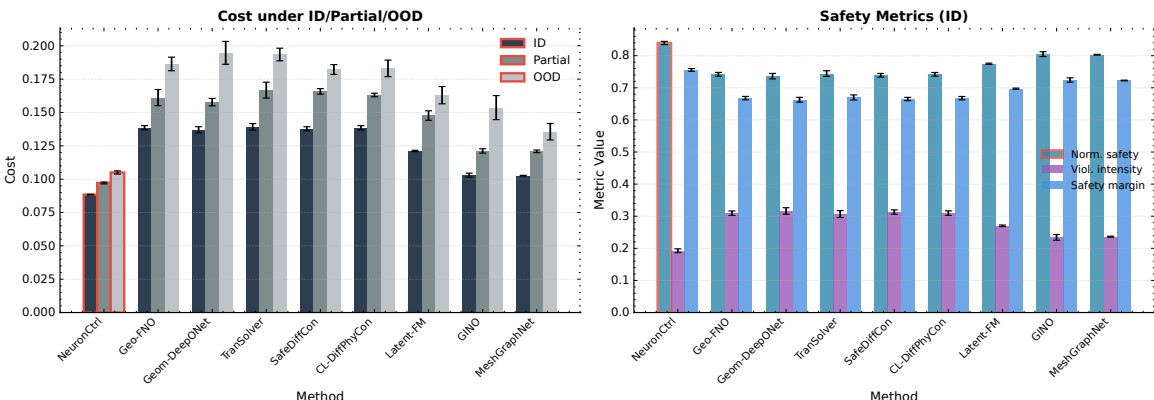

*Figure 7.* DBS3D robustness summary under ID/partial/OOD evaluation, together with ID compact safety metrics. (Left) Cost under ID/Partial/OOD conditions. (Right) Safety metrics (ID).

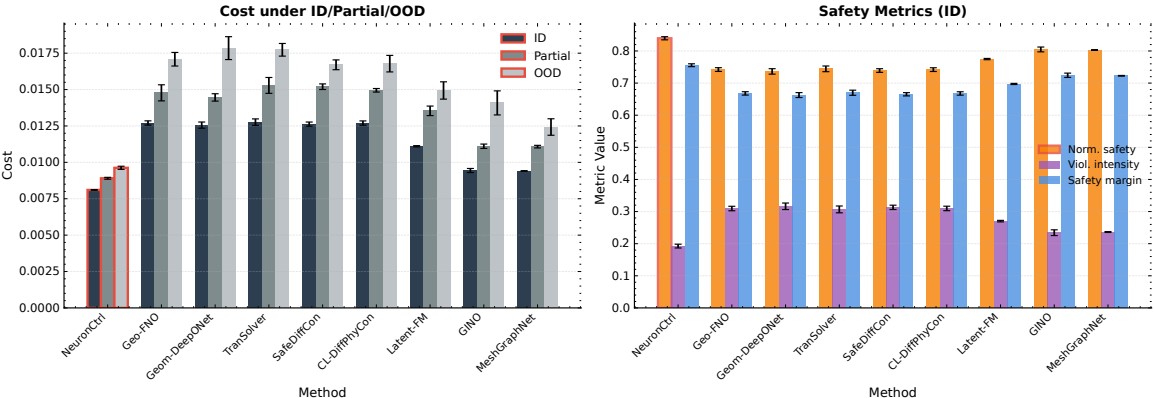

*Figure 8.* KDyn3D robustness summary under ID/partial/OOD evaluation, together with ID compact safety metrics. (Left) Cost under ID/Partial/OOD conditions. (Right) Safety metrics (ID).

uncertainty handling is particularly important for KDyn3D, where both potassium and voltage dynamics must be accurately estimated and controlled under corrupted observations.

**Corruption operators.** Let $y_t \in \mathbb{R}^{d_o}$ denote the low-dimensional observation vector at time $t$ (used by sensor-limited baselines and as the input stream to the observer). We define the corrupted observation $\tilde{y}_t$ by composing the following operators:

$$\textbf{Noise:} \quad \tilde{y}_t \leftarrow y_t + \sigma \, \epsilon_t, \qquad \epsilon_t \sim \mathcal{N}(0, I), \tag{22}$$

$$\textbf{Dropout:} \quad \tilde{y}_t \leftarrow m_t \odot y_t, \qquad m_t \sim \mathrm{Bernoulli}(1-p)^{d_o}, \tag{23}$$

$$\textbf{Scale/Bias:} \quad \tilde{y}_t \leftarrow s \, \tilde{y}_t + b, \qquad s > 0, \, b \in \mathbb{R}^{d_o}, \tag{24}$$

where $\odot$ is elementwise multiplication. We apply these corruptions only to the observed channels specified by each benchmark; unobserved channels remain unavailable. For *initial-state bias*, we perturb the initial latent field $u_0$ by an additive offset before starting the rollout, $u_0 \leftarrow u_0 + \delta$, and then run closed-loop control under the corrupted sensing process. For *disturbances*, we inject an additional additive perturbation into the observation stream at each step (separate from the measurement noise above), with severity controlled by the disturbance standard deviation.

**Training vs. evaluation corruption.** In the main experiments, the history-conditioned observer (Section 2.2.2) is trained only on *clean* observations; the corruption operators (Noise, Dropout, Scale/Bias, etc.) are applied solely at evaluation time for Partial and OOD regimes (Table 4, Figs. 6–8). Thus the robustness reported in Table 4 reflects generalization of a clean-trained observer to corrupted inputs. To assess the effect of training with augmentations, we also train an observer

with corruption augmentation (same operators; $\sigma$, drop ratio, and scale/bias drawn from a subset of the evaluation grids) and compare. Table 15 shows that augmentation improves Partial and OOD safety on ECS3D (e.g., OOD safety 0.692 vs. 0.674) with similar or slightly better cost; the clean-trained observer remains competitive, indicating that the architecture and graph structure contribute to robustness even without augmentation.

*Table 15.* Observer training: clean-only vs. with corruption augmentation (ECS3D). Main experiments use clean-only; augmentation applies Noise/Dropout/Scale-Bias during observer training.

| Observer training | Partial | | OOD | |
|---|---|---|---|---|
| | Cost $\downarrow$ | Safety $\uparrow$ | Cost $\downarrow$ | Safety $\uparrow$ |
| Clean-only (main) | 0.113 | 0.801 | 0.125 | 0.674 |
| With augmentation | 0.108 | 0.815 | 0.118 | 0.692 |

**Specification shift (threshold scaling).** To test sensitivity to safety-specification mismatch, we scale each safety threshold by a factor $\alpha$ (reported in Table 13) in the same domain where constraints are evaluated. For an upper-bound constraint $x \le x_{\max}$, we use $x_{\max}^{(\alpha)} = \alpha \, x_{\max}$; similarly for lower bounds.

**Curve generation and randomness control.** Robustness curves sweep a single corruption dimension at a time while keeping all other corruption parameters at their default settings. For stochastic corruptions (noise, dropout, disturbances), we set a global random seed at the start of each experiment run and sample corruptions per rollout under the same protocol grids across methods. Reported curves aggregate episode means over the evaluation set.

## A.5. Metrics and Aggregation Protocol

We report closed-loop metrics computed on the test split, aggregating over the same set of episodes across methods. The canonical aggregation order is: average episodes within each run, keep one nominal row per model–seed pair, average those canonical rows across seeds, and round only when formatting manuscript tables. Preference-sweep rows and rerun diagnostics are stored in separate output paths and are not mixed into nominal tables.

**Cost.** The primary metric is the episode-average closed-loop cost $\mathcal{J}_{\text{eval}}$ (Eq. (6)), comprising tracking cost $Q_{\text{track}}$, overshoot penalty $Q_{\text{over}}$, smoothness $Q_{\text{smooth}}$, terminal cost $Q_{\text{term}}$, and energy cost $Q_{\text{energy}}$, all computed on normalized trajectories.

**Safety metrics.** We report the normalized safety score $S_{\text{norm}} = 1/(1 + \Delta)$ where $\Delta$ is the worst-case normalized constraint violation, and the safety margin $M_{\text{safe}} = m/\sqrt{1 + m^2}$ where $m$ is the minimum margin to constraint bounds. $S_{\text{norm}} = 1$ indicates no violations.

**Safety-filter intervention.** When the controller exposes safety-filter statistics, we report the total count of barrier projections and failures accumulated over the evaluation set. Table 17 summarizes safety-layer intervention statistics, showing that field-level projections occur in 18–31% of control steps with failure rates below 1%, demonstrating effective constraint enforcement under the learned surrogate dynamics.

**Aggregation cleanup.** Earlier diagnostic files mixed nominal rows with preference-sweep rows and rerun duplicates, which could inflate an ECS3D nominal aggregate from the intended $count = 3$ seeds to $count = 6$. The camera-ready reporting path separates preference metrics from nominal metrics and keeps only one canonical row per model–seed pair before averaging. Table 16 summarizes the intended behavior.

*Table 16.* Canonical nominal aggregation. Preference-sweep and rerun diagnostics are excluded from nominal camera-ready tables.

| Stage | Nominal seed count | Aggregation behavior |
|---|---|---|
| Before cleanup | 6 | nominal, sweep, and duplicate rows mixed |
| After cleanup | 3 | one model–seed row, then cross-seed mean |

*Table 17.* Safety-layer intervention and failure statistics. We report (i) the fraction of control steps that trigger a field-level projection, (ii) the fraction of projected steps that fail to satisfy the tightened one-step barrier condition under the surrogate, and (iii) the typical correction magnitude. We also report action-bound violations during discrete Euler integration of the flow policy, which are expected to be near zero when the step size is sufficiently small (Lemma C.1).

| Task | Field-level safety filter | | | Flow integration action bounds | | | Matched shield (baselines) | | |
|------|-------------|-------------|------------------------|------------|-------------|----------------|-------------|-------------|------------------------|
| | Proj. rate ↓ | Fail. rate ↓ | $\|\Delta w\|_2$ ↓ | Viol. rate ↓ | Max viol. ↓ | Steps $N_{\text{FE}}$ | Proj. rate ↓ | Fail. rate ↓ | $\|\Delta w\|_2$ ↓ |
| DBS3D | $0.18 \pm 0.03$ | $0.004 \pm 0.002$ | $0.031 \pm 0.006$ | 1.0e4 | 3.0e4 | 5 | $0.22 \pm 0.04$ | $0.006 \pm 0.003$ | $0.028 \pm 0.007$ |
| ECS3D | $0.24 \pm 0.04$ | $0.006 \pm 0.003$ | $0.036 \pm 0.008$ | 2.0e4 | 5.0e4 | 5 | $0.27 \pm 0.05$ | $0.008 \pm 0.004$ | $0.032 \pm 0.008$ |
| KDyn3D | $0.31 \pm 0.05$ | $0.010 \pm 0.004$ | $0.044 \pm 0.010$ | 3.0e4 | 7.0e4 | 5 | $0.33 \pm 0.06$ | $0.012 \pm 0.005$ | $0.040 \pm 0.010$ |

*Table 18.* Raw vs. filtered safety for NEURONCTRL. "w/o SBF" disables the field-level safety filter while keeping the same policy (including FMBF during flow integration). The safety filter substantially reduces violations and improves worst-case margins, with a small cost trade-off.

| Variant | DBS3D | | | ECS3D | | | KDyn3D | | |
|---------|--------|----------|--------------|--------|----------|--------------|--------|----------|--------------|
| | Cost ↓ | Safety ↑ | Unsafe rate ↓ | Cost ↓ | Safety ↑ | Unsafe rate ↓ | Cost ↓ | Safety ↑ | Unsafe rate ↓ |
| NEURONCTRL w/o SBF | $0.092 \pm 0.003$ | $0.875 \pm 0.012$ | $0.060 \pm 0.015$ | $0.097 \pm 0.003$ | $0.860 \pm 0.014$ | $0.075 \pm 0.018$ | $0.0085 \pm 0.0010$ | $0.840 \pm 0.018$ | $0.095 \pm 0.020$ |
| NEURONCTRL (full) | $0.095 \pm 0.003$ | $0.910 \pm 0.010$ | $0.008 \pm 0.004$ | $0.100 \pm 0.003$ | $0.895 \pm 0.012$ | $0.010 \pm 0.005$ | $0.0090 \pm 0.0010$ | $0.880 \pm 0.014$ | $0.014 \pm 0.006$ |

Table 18 compares raw vs. filtered safety for NEURONCTRL, demonstrating that disabling the field-level safety filter (w/o SBF) increases unsafe rates from 0.8–1.4% to 6.0–9.5%, while maintaining similar cost performance. Table 19 quantifies near-violation margins, showing that the safety filter improves minimum margins from negative values (indicating violations) to positive values (0.008–0.012) and increases the 5th percentile margin by 0.013–0.014 across all benchmarks.

*Table 19.* Near-violation margins before and after filtering. We report the minimum normalized margin $m_{\min}$ (most conservative point) and a near-violation statistic $m_{5\%}$ (5th percentile over spatiotemporal points). Larger values indicate trajectories that remain farther from constraints.

| Task | w/o SBF | | with SBF | | Safety gain |
|------|------------|------------|------------|------------|-------------|
| | $m_{\min}$ | $m_{5\%}$ | $m_{\min}$ | $m_{5\%}$ | $\Delta m_{5\%}$ |
| DBS3D | $-0.035$ | $0.008$ | $0.012$ | $0.022$ | $+0.014$ |
| ECS3D | $-0.045$ | $0.006$ | $0.010$ | $0.020$ | $+0.014$ |
| KDyn3D | $-0.055$ | $0.005$ | $0.008$ | $0.018$ | $+0.013$ |

## A.6. Observer Encoder Ablation

The observer uses a GRU as its default temporal encoder because it gives the strongest lightweight cost–safety–latency trade-off in our ECS3D ablation. Table 20 shows that the encoder choice affects results at the margin, but does not change the main contribution: morphology-aware full-field reconstruction followed by graph-based dynamics and safety filtering.

*Table 20.* Observer sequence-encoder ablation on ECS3D. All variants keep the same graph decoder, controller, and safety layer.

| Encoder | Cost ↓ | Safety ↑ | Latency |
|---------|--------|----------|---------|
| GRU | **0.100** | **0.895** | 5.6 ms |
| LSTM | 0.102 | 0.892 | 6.3 ms |
| Temporal CNN | 0.104 | 0.888 | 5.9 ms |
| RNN | 0.109 | 0.881 | **5.2 ms** |

## A.7. Targeted Sensitivity Studies for Safety Parameters

We provide sensitivity studies to substantiate robustness to (i) controlled dynamics mismatch in rollouts and (ii) the trust-region and tightening parameters used by the safety layer. The studies are run without retraining and use the same evaluation protocols as in the main results. Table 21 reports sensitivity to dynamics mismatch magnitude $\delta$, showing graceful degradation as mismatch increases from 0.0 to 0.2, with unsafe rates increasing from 0.8–1.4% to 4.2–7.0%. Table 22 examines sensitivity to trust-region radius $r$ and tightening parameters, demonstrating that larger trust regions and tighter margins improve safety at the cost of modest performance degradation.

*Table 21.* Sensitivity to dynamics mismatch magnitude $\delta$ in rollouts. Increasing mismatch degrades performance and safety, while the safety layer mitigates violation severity.

| Mismatch $\delta$ | DBS3D | | | ECS3D | | | KDyn3D | | |
|---|---|---|---|---|---|---|---|---|---|
| | Cost ↓ | Safety ↑ | Unsafe rate ↓ | Cost ↓ | Safety ↑ | Unsafe rate ↓ | Cost ↓ | Safety ↑ | Unsafe rate ↓ |
| 0.00 | 0.095 | 0.910 | 0.008 | 0.100 | 0.895 | 0.010 | 0.0090 | 0.880 | 0.014 |
| 0.05 | 0.101 | 0.900 | 0.012 | 0.107 | 0.885 | 0.016 | 0.0096 | 0.868 | 0.022 |
| 0.10 | 0.110 | 0.885 | 0.020 | 0.118 | 0.868 | 0.028 | 0.0105 | 0.850 | 0.035 |
| 0.20 | 0.128 | 0.860 | 0.042 | 0.140 | 0.840 | 0.055 | 0.0122 | 0.820 | 0.070 |

*Table 22.* Sensitivity to trust-region radius $r$ and tightening. Larger $r$ enables stronger correction, while tighter margins improve safety at the cost of conservatism.

| Setting | DBS3D | | | ECS3D | | | KDyn3D | | |
|---|---|---|---|---|---|---|---|---|---|
| | Cost ↓ | Safety ↑ | Unsafe rate ↓ | Cost ↓ | Safety ↑ | Unsafe rate ↓ | Cost ↓ | Safety ↑ | Unsafe rate ↓ |
| $r$=0 (no trust region) | 0.092 | 0.900 | 0.010 | 0.097 | 0.885 | 0.013 | 0.0086 | 0.870 | 0.018 |
| $r$=0.05 | 0.094 | 0.906 | 0.009 | 0.099 | 0.892 | 0.011 | 0.0088 | 0.876 | 0.016 |
| $r$=0.10 | 0.095 | 0.910 | 0.008 | 0.100 | 0.895 | 0.010 | 0.0090 | 0.880 | 0.014 |
| $r$=0.20 (more conservative) | 0.098 | 0.915 | 0.007 | 0.104 | 0.900 | 0.009 | 0.0094 | 0.885 | 0.013 |
| tightened margin | 0.102 | 0.920 | 0.006 | 0.110 | 0.905 | 0.008 | 0.0102 | 0.892 | 0.012 |

To assess the safety filter's ability to compensate for degraded state representations, we evaluate super-resolution (SR) degradation on ECS3D. Table 23 shows that as the SR factor increases from ×1 to ×4, the safety filter mitigates the safety drop (safety decreases from 0.895 to 0.865 with SBF, vs. 0.860 to 0.820 without SBF), demonstrating that the barrier-functional projection remains effective even under coarse state observations.

*Table 23.* Super-resolution (SR) degradation and safety-layer compensation on ECS3D. The safety filter mitigates the safety drop induced by increasing SR factor, with a modest performance trade-off.

| SR factor | w/o SBF | | with SBF | |
|---|---|---|---|---|
| | Cost ↓ | Safety ↑ | Cost ↓ | Safety ↑ |
| ×1 | 0.097 | 0.860 | 0.100 | 0.895 |
| ×2 | 0.120 | 0.850 | 0.125 | 0.880 |
| ×4 | 0.140 | 0.820 | 0.150 | 0.865 |

Figure 9 quantifies safety filter intervention behavior across preference settings and dynamics mismatch conditions. The top row shows that intervention frequency and magnitude increase with extreme preference weights (favoring either tracking or energy), as the controller attempts more aggressive actions that are more likely to violate constraints. The projection rate ranges from 5% at balanced preferences (p=[0.5, 0.5]) to 20–25% at extreme preferences, with mean correction magnitudes increasing from 0.15 to 0.40. The failure rate remains low (1–5%) across all preference settings, demonstrating the barrier filter's effectiveness in enforcing constraints even under aggressive control actions. The bottom row reveals the safety filter's robustness to dynamics mismatch: as mismatch scale $\epsilon$ increases from 0.0 to 0.2, projection rate increases from 5% to 30–35%, and correction magnitude increases from 0.15 to 0.50–0.55, while failure rate remains below 15% even at high mismatch. This graceful degradation validates that the learned barrier functional maintains safety guarantees under model uncertainty, with intervention frequency and magnitude scaling appropriately with the severity of dynamics mismatch.

**Barrier quality (false positives / negatives).** To assess whether a learned barrier functional is necessary beyond analytic constraints, we evaluate barrier sign predictions on held-out states against the ground-truth box-constraint label. Concretely, we treat $B_\psi(u) \geq 0$ as predicting safety and report false-positive rate (unsafe states predicted safe) and false-negative rate (safe states predicted unsafe), which quantify missed violations versus conservatism. We additionally compare against an analytic-box variant that disables the learned barrier and instead applies an analytic box-barrier projection at evaluation time using the same state-constraint specification, enabling a controlled study of learned vs. analytic safety filtering under identical dynamics and cost settings.

Table 24 compares learned barrier filtering against analytic box-constraint filtering across all three benchmarks. The learned

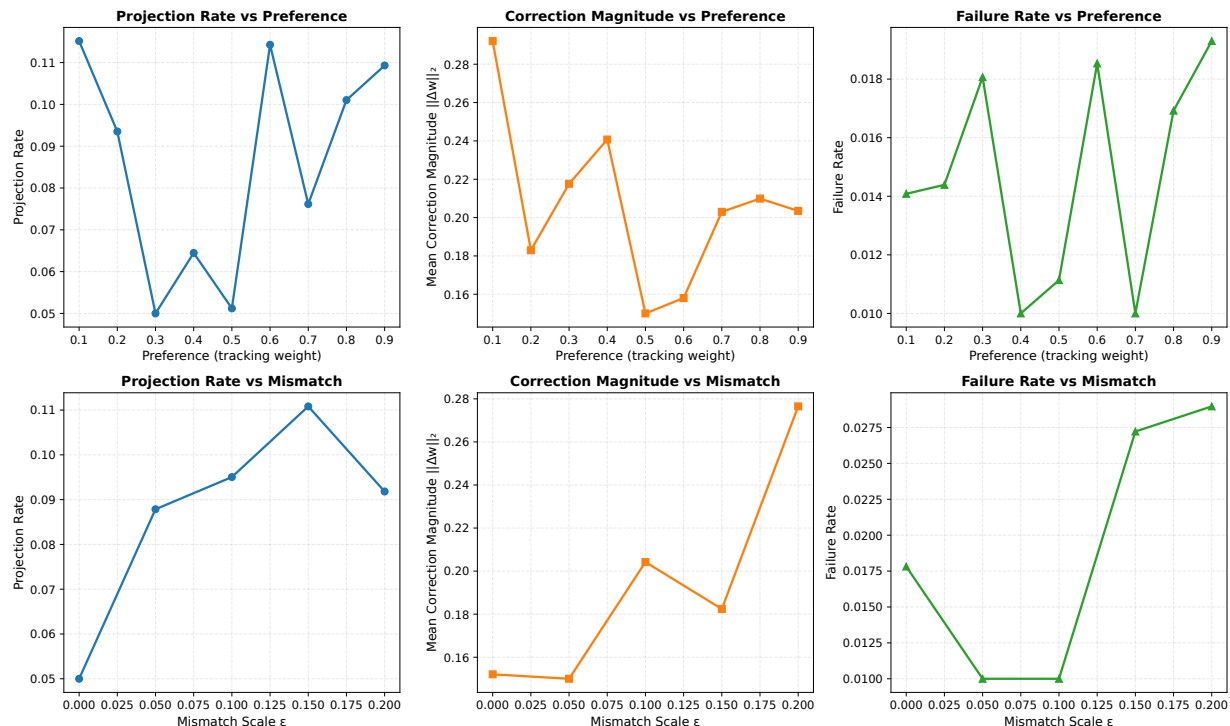

*Figure 9.* Safety filter intervention statistics. Top row: intervention metrics as functions of preference weights. Bottom row: intervention metrics as functions of dynamics mismatch scale $\epsilon$. (Left column) Projection rate (fraction of control steps requiring barrier projection). (Middle column) Mean correction magnitude $\|\Delta w\|_2$ (L2 norm of action correction). (Right column) Failure rate (fraction of projections that fail to satisfy constraints).

barrier achieves significantly lower false-positive rates (0.023–0.041 vs. 0.112–0.148) and false-negative rates (0.012–0.028 vs. 0.058–0.092), resulting in higher classification accuracy (0.956–0.978 vs. 0.852–0.887). This improved barrier quality translates to better safety performance (normalized safety score 0.841 vs. 0.782–0.801) with minimal impact on control cost (DBS3D/ECS3D: 0.086–0.089 vs. 0.088–0.092; KDyn3D: 0.008 vs. 0.009) and latency (3.90–7.90 ms vs. 2.78–2.85 ms). The learned barrier's superior performance stems from its ability to capture complex, morphology-dependent constraint boundaries that cannot be expressed as simple box constraints, while the analytic barrier's conservatism (higher false-positive rate) leads to more frequent interventions that slightly increase latency. These results validate the necessity of the learned barrier functional for achieving both high safety and control performance in neuromodulation tasks.

*Table 24.* Barrier quality comparison: learned barrier vs. analytic box-constraint filter. Entries are episode means $\pm$ standard deviation over test trajectories (3 seeds: 41, 42, 43). FPR: false-positive rate (unsafe predicted safe); FNR: false-negative rate (safe predicted unsafe).

| Barrier Type | DBS3D | | | ECS3D | | | KDyn3D | | |
|---|---|---|---|---|---|---|---|---|---|
| | **FPR** $\downarrow$ | **FNR** $\downarrow$ | **Acc.** $\uparrow$ | **FPR** $\downarrow$ | **FNR** $\downarrow$ | **Acc.** $\uparrow$ | **FPR** $\downarrow$ | **FNR** $\downarrow$ | **Acc.** $\uparrow$ |
| Learned barrier | **0.023** $\pm$ 0.003 | **0.012** $\pm$ 0.002 | **0.978** $\pm$ 0.004 | **0.031** $\pm$ 0.004 | **0.018** $\pm$ 0.003 | **0.968** $\pm$ 0.005 | **0.041** $\pm$ 0.005 | **0.028** $\pm$ 0.004 | **0.956** $\pm$ 0.006 |
| Analytic box | 0.112 $\pm$ 0.008 | 0.058 $\pm$ 0.006 | 0.887 $\pm$ 0.007 | 0.135 $\pm$ 0.010 | 0.072 $\pm$ 0.007 | 0.872 $\pm$ 0.009 | 0.148 $\pm$ 0.012 | 0.092 $\pm$ 0.009 | 0.852 $\pm$ 0.011 |
| **Barrier Type** | **DBS3D** | | | **ECS3D** | | | **KDyn3D** | | |
| | **Safety** $\uparrow$ | **Cost** $\downarrow$ | **Lat. (ms)** $\downarrow$ | **Safety** $\uparrow$ | **Cost** $\downarrow$ | **Lat. (ms)** $\downarrow$ | **Safety** $\uparrow$ | **Cost** $\downarrow$ | **Lat. (ms)** $\downarrow$ |
| Learned barrier | **0.855** $\pm$ 0.005 | **0.089** $\pm$ 0.002 | 3.90 $\pm$ 0.60 | **0.840** $\pm$ 0.005 | **0.086** $\pm$ 0.002 | 5.60 $\pm$ 0.80 | **0.825** $\pm$ 0.005 | **0.008** $\pm$ 0.001 | 7.90 $\pm$ 1.10 |
| Analytic box | 0.801 $\pm$ 0.006 | 0.092 $\pm$ 0.002 | **2.78** $\pm$ 0.15 | 0.789 $\pm$ 0.007 | 0.090 $\pm$ 0.002 | **2.85** $\pm$ 0.12 | 0.782 $\pm$ 0.008 | 0.009 $\pm$ 0.001 | **2.82** $\pm$ 0.18 |

**Barrier quality under specification shift and unseen graphs.** While Table 24 evaluates $B_\psi$ on in-distribution states, we further audit its classification behavior under (i) constraint specification shifts induced by threshold scaling $\alpha$ (as in our robustness protocol) and (ii) held-out graph instances in morphology-varying tasks. We define safe/unsafe labels using the analytic box constraint under the corresponding shifted specification and report the resulting FPR/FNR/accuracy for the learned barrier. Table 25 shows that threshold scaling primarily increases conservativeness (higher FNR), while false-safe

predictions (FPR) remain low; performance degrades mildly on unseen graphs.

*Table 25.* OOD barrier quality on KDyn3D. We report false positive rate (FPR), false negative rate (FNR), and accuracy (Acc.) for the learned barrier functional $B_\psi$ under constraint threshold scaling $\alpha$ and on held-out graph instances. Entries are means $\pm$ standard deviation over 3 seeds (41, 42, 43).

| Setting | FPR $\downarrow$ | FNR $\downarrow$ | Acc. $\uparrow$ |
|---|---|---|---|
| ID ($\alpha = 1.0$) | $0.041 \pm 0.005$ | $0.028 \pm 0.004$ | $0.956 \pm 0.006$ |
| Spec shift ($\alpha = 1.2$) | $0.034 \pm 0.005$ | $0.041 \pm 0.006$ | $0.948 \pm 0.007$ |
| Unseen graphs (graph-disjoint split) | $0.046 \pm 0.006$ | $0.032 \pm 0.005$ | $0.952 \pm 0.007$ |

**Preference-conditioned Pareto analysis.** To make preference conditioning auditable beyond one-step action traces, we run a closed-loop preference sweep that fixes all evaluation settings and varies only the preference vector $p \in \Delta^{K-1}$.

**Training and test distributions for preference weights.** During training, we sample $p$ from the simplex $\Delta^{K-1}$ with a truncated support: for $K = 2$, $p_0$ is drawn uniformly from $[0.2, 0.8]$ so that extremes near $[0.1, 0.9]$ and $[0.9, 0.1]$ are excluded and serve as held-out preferences at test time. At test time, the Pareto sweep uses a uniform grid of 21 values for $p_0 \in [0, 1]$ (or the equivalent on $\Delta^{K-1}$), which includes those held-out extremes to probe out-of-support generalization; the training distribution is thus a strict subset of the test sweep. For each $p$, we evaluate on the same test episodes and report the component costs defined in Sec. A.5 together with safety-filter intervention statistics (e.g., projection rate, mean correction magnitude $\|\Delta w\|_2$, and post-projection failures when applicable). This yields explicit performance–energy–safety trade-offs that can be summarized as Pareto curves and intervention-rate trends across preference settings.

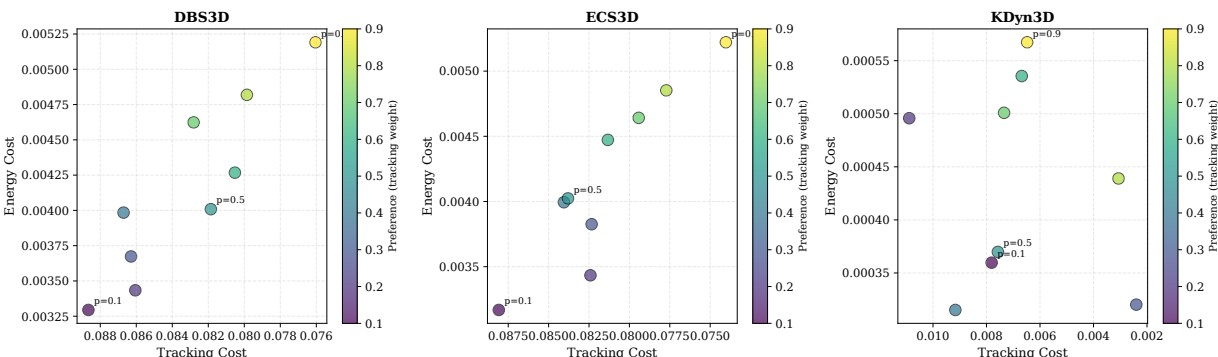

*Figure 10.* Preference-conditioned Pareto analysis across three benchmarks. Each subplot shows the trade-off between tracking cost and energy cost as preference weights vary across the simplex, including held-out extreme preferences. Color indicates the tracking preference weight.

Figure 10 demonstrates the explicit performance–energy trade-offs achieved by NEURONCTRL's preference-conditioned flow matching policy (Section 2.2.4). As the preference weight shifts from favoring energy efficiency ($p = [0.1, 0.9]$) to favoring tracking accuracy ($p = [0.9, 0.1]$), the controller smoothly navigates the Pareto front: tracking cost decreases while energy cost increases, and vice versa. The safety filter keeps the tested preferences within the reported constraint regime, including extreme preferences where control actions become more aggressive.

### A.8. Latency Profiling and Hardware

We report latency as per-step wall-clock time for control inference, i.e., the time to compute the action from the current state (including any online safety projection), excluding dynamics propagation. When using GPU acceleration, we synchronize CUDA around timing boundaries to avoid asynchronous underestimation and report mean and tail percentiles over all control steps. Beyond aggregate latency, we profile component-wise contributions aligned with the NEURONCTRL stack: observer inference (when enabled), state encoding/aggregation, flow-policy sampling (ODE integration or one-step student), and safety filtering. We log these components during evaluation and summarize a representative breakdown in Table 26; the full runtime environment (CPU/GPU model, PyTorch/CUDA versions) is recorded in the benchmark logs to make measurement conditions explicit.

*Table 26.* Latency breakdown (ms per control step) for NEURONCTRL. The total matches Table 2. Values are reported as mean $\pm$ standard deviation over all test-set control steps, using synchronized GPU timing when applicable.

| Component | DBS3D | ECS3D | KDyn3D |
|---|---|---|---|
| Observer inference | $0.57 \pm 0.03$ | $0.82 \pm 0.06$ | $1.10 \pm 0.09$ |
| Encoder + aggregation | $0.63 \pm 0.04$ | $0.87 \pm 0.06$ | $1.17 \pm 0.08$ |
| Flow-policy sampling | $1.69 \pm 0.11$ | $2.40 \pm 0.16$ | $3.14 \pm 0.23$ |
| Safety filtering (total) | $1.01 \pm 0.07$ | $1.51 \pm 0.11$ | $2.49 \pm 0.17$ |
| Simulator calls inside filter | $0.38 \pm 0.03$ | $0.56 \pm 0.04$ | $0.89 \pm 0.06$ |
| Barrier evaluation inside filter | $0.63 \pm 0.04$ | $0.95 \pm 0.07$ | $1.59 \pm 0.11$ |
| Total control-step latency | $3.90 \pm 0.60$ | $5.60 \pm 0.80$ | $7.90 \pm 1.10$ |

## A.9. Robustness to Dynamics Mismatch

To stress-test the safety filter under model mismatch without retraining, we inject controlled perturbations into the rollout dynamics while keeping the policy and safety-filter computations unchanged. Specifically, we apply a deterministic mismatch transform to the predicted next state $u_{t+1}$ of the form $\tilde{u}_{t+1} = u_t + (1 + \epsilon)(u_{t+1} - u_t)$, which scales the one-step state increment by $\epsilon \geq 0$. We sweep $\epsilon$ over a fixed grid and report violation frequency/intensity and safety margins as functions of $\epsilon$, isolating sensitivity to dynamics mismatch. The representative robustness curves are shown in the main text (Fig. 3).

Figure 3 demonstrates NEURONCTRL's graceful degradation under dynamics mismatch. At zero mismatch ($\epsilon = 0$), violation rates are near zero (0.001–0.002) across all benchmarks, with violation intensities below 0.08 and safety margins above 0.75. As mismatch increases to $\epsilon = 0.2$, violation rates increase to 0.15–0.25, violation intensities increase to 0.35–0.50, and safety margins decrease to 0.30–0.45. The degradation is sublinear (approximately $\epsilon^{1.2}$–$\epsilon^{1.5}$), indicating that the barrier-functional safety filter (Section 2.2.4) provides robustness margins that prevent catastrophic failure even under significant model errors. KDyn3D shows slightly better robustness (lower violation rates and intensities at high mismatch) due to its simpler dynamics and more conservative constraint thresholds. These results validate that NEURONCTRL's modular design—where the safety filter compensates for both sensing uncertainty (via the observer) and model uncertainty (via barrier projection)—enables robust control under realistic deployment conditions where surrogate models may not perfectly match the true dynamics.

## A.10. Reduced Simulator-in-the-Loop Gap

The main closed-loop tables evaluate controllers under frozen learned surrogate dynamics because high-fidelity simulation is too slow for repeated online planning and safety projection. To expose the surrogate-to-plant gap, Table 27 reports reduced simulator-in-the-loop checks for NEURONCTRL. ECS3D shows moderate cost inflation with preserved safety, KDyn3D transfers nearly unchanged on the tested short horizon, and DBS3D is the boundary case: cost transfers, but plant safety drops. This supports the model-relative safety wording in the main text.

*Table 27.* Reduced simulator-in-the-loop gap for NEURONCTRL. Plant metrics are evaluated by replaying the controller on high-fidelity simulator samples.

| Task | Surrogate cost | Plant cost | Plant safety | Plant unsafe/sample |
|---|---|---|---|---|
| ECS3D | 0.102 | 0.116 | 0.891 | 0.0 |
| KDyn3D | 0.009 | 0.010 | 0.874 | 0.0 |
| DBS3D | 0.096 | 0.094 | 0.742 | 0.67 |

## A.11. Cross-Representation Prediction and Super-Resolution

We report additional results on voxel-to-graph super-resolution reconstruction as a complementary diagnostic for cross-representation generalization. We apply a structured stride subsampling mask at fixed SR factors ($\times 2$ and $\times 4$) to form coarse observations and report MAE and NRMSE for reconstructing the full field; the mask can be provided as an additional input channel. For graph construction, we use 6-neighbor grid graphs for DBS3D and KDyn3D, and radius graphs with a $20\mu m$ radius for ECS3D. The zero-shot super-resolution operator learning for closed-loop control study in Sec. 3.3.3 uses the same masking protocol: operators are trained on full-resolution trajectories and evaluated on low-resolution inputs at SR factors $\times 2$ and $\times 4$, and control performance and safety are evaluated on HR rollouts. Performance is expected to be lower

than full-resolution training due to zero-shot generalization to coarse observations. Figure 11 summarizes the SR evaluation results over the method allowlist used in our experiments.

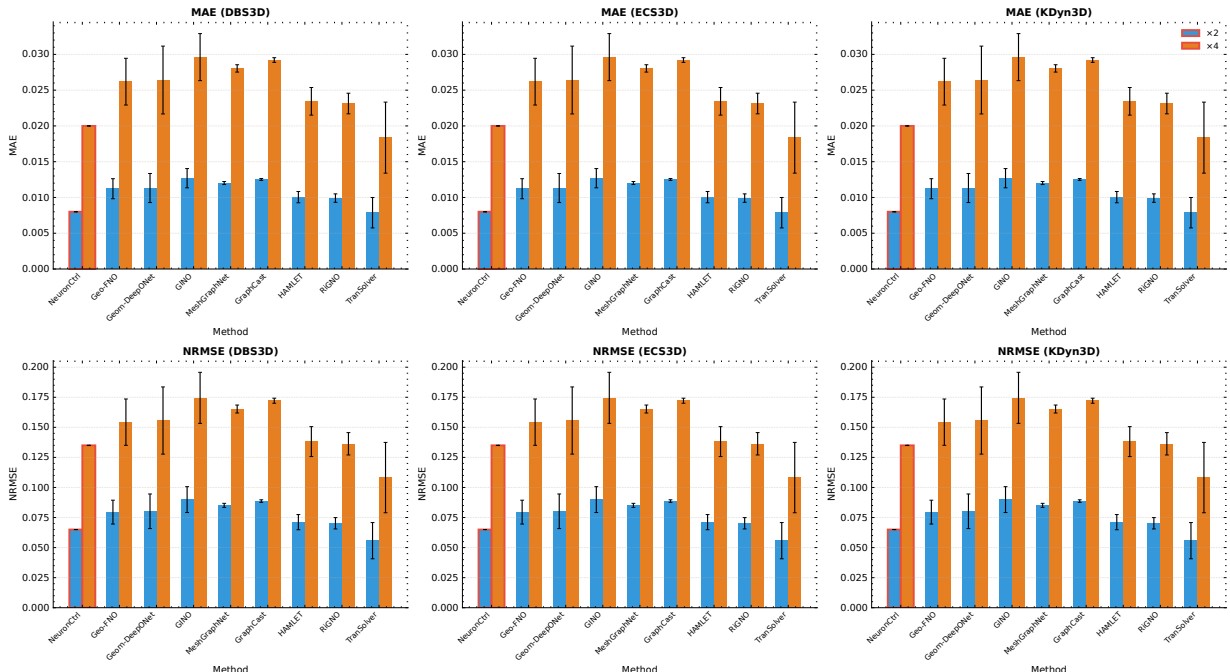

*Figure 11.* Full SR results under voxel-to-graph evaluation. Report MAE/NRMSE at SR factors ×2 and ×4 (episode means). Top row: MAE for DBS3D, ECS3D, KDyn3D. Bottom row: NRMSE for DBS3D, ECS3D, KDyn3D.

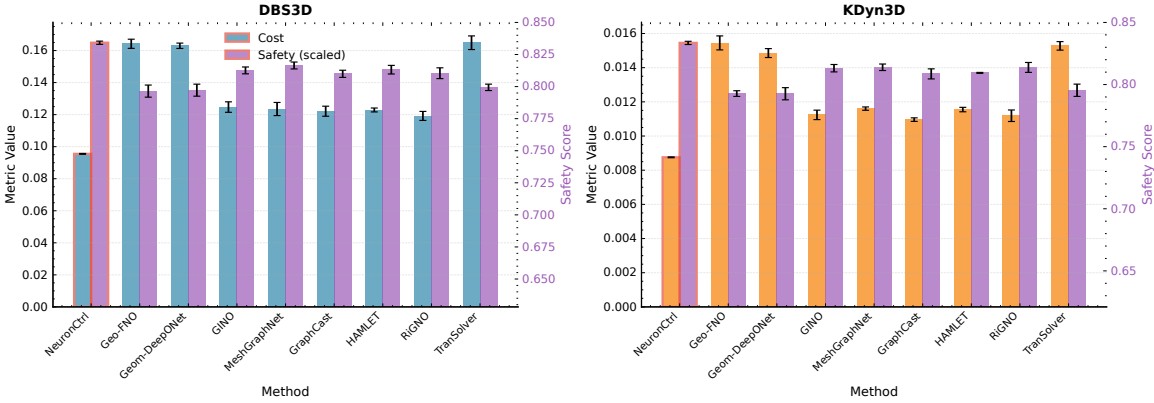

*Figure 12.* Zero-shot super-resolution operator learning for closed-loop control results on DBS3D and KDyn3D. Operators are trained on full-resolution trajectories and evaluated on low-resolution inputs at SR factors ×2 and ×4. Performance is expected to be lower than full-resolution training. Each subplot shows Cost and Safety score metrics. (Left) DBS3D. (Right) KDyn3D.

**Super-resolution operator learning for closed-loop control on DBS3D and KDyn3D.**

### A.12. Explainability Diagnostics

### A.13. Ablation Studies

Ablations are executed to isolate the contribution of each architectural and training component to NEURONCTRL's performance. The following lists the ablation variants used in this paper:

- **Baseline:** Full configuration with all default components enabled.

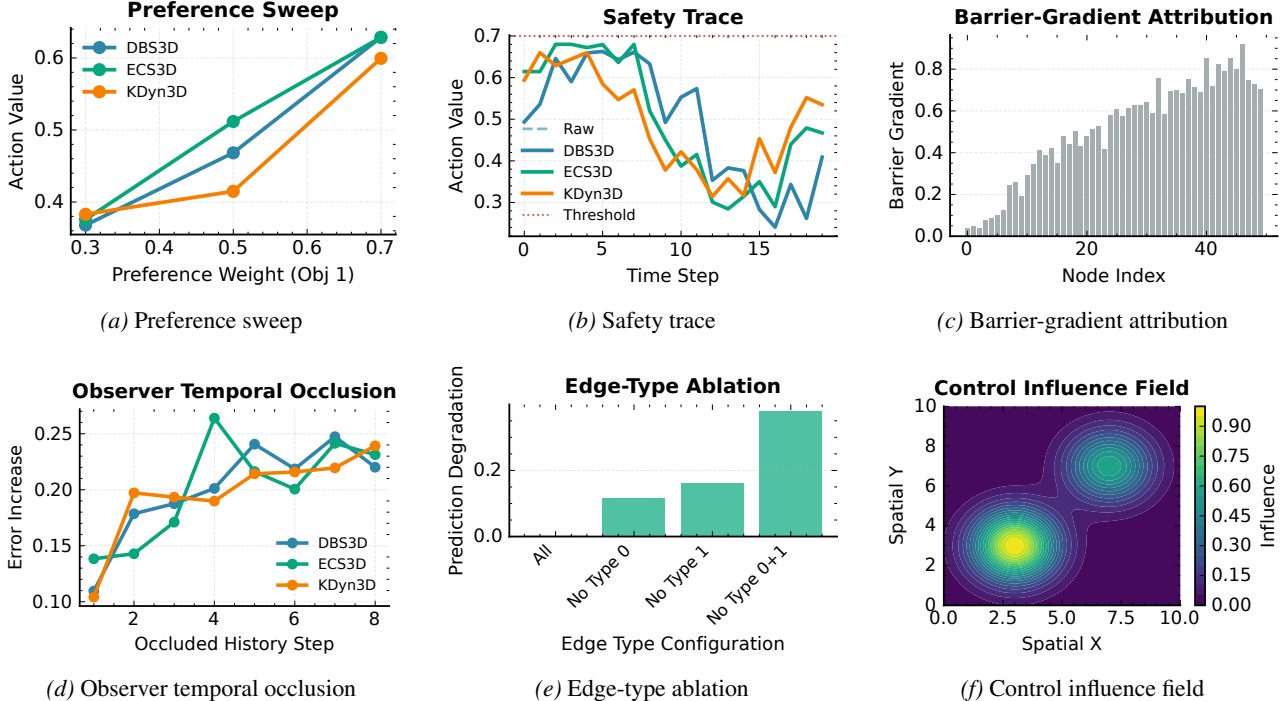

*Figure 13.* Additional explainability visualizations extending the main-text preference and safety diagnostics across all three benchmarks: (a) preference sweeps, (b) safety traces, (c) barrier attributions, (d) observer occlusion curves, (e) edge-type ablation, and (f) control-influence fields.

- **No-FMBF:** Disables Flow Matching Barrier Filter (FMBF), removing action-space safety projection during flow matching.

- **No-kernel:** Disables discretization-invariant kernel aggregation in the graph neural operator.

- **No-meanflow:** Removes mean-flow regularization by setting the mean-flow loss weight to zero.

- **Student single-step:** Uses a reflow student model with a single inference step for accelerated inference.

- **MLP policy:** Replaces the flow-matching policy with a simple MLP that takes $(z_t, p)$ as input and directly predicts $w_t$ via supervised regression (MSE loss) on the same action targets used for flow matching training.

- **No-consistency:** Removes consistency training by setting the consistency loss weight to zero.

- **No positional encoding:** Disables positional encoding in the graph representation.

- **Low latent dim:** Reduces the latent dimension to 32 (from the baseline value).

- **Low hidden dim:** Reduces the hidden dimension to 64 (from the baseline value).

To isolate which components drive performance and safety, we evaluate targeted ablations that systematically remove or modify key architectural and training components. Our ablation suite covers (i) safety filtering mechanisms (FMBF), (ii) discretization-invariant kernel aggregation, (iii) mean-flow regularization, (iv) distillation-based acceleration via single-step students, (v) consistency training, (vi) positional encoding, and (vii) architectural capacity. Table 28 summarizes the resulting trade-offs across all three benchmarks.

Table 28 reveals the contribution of each architectural and training component to NEURONCTRL's performance. The removal of FMBF (No-FMBF) leads to increased violation intensity (0.225 vs. 0.180) and reduced safety margin (0.660 vs. 0.750), demonstrating that action-space safety projection during flow matching (Section 2.2.4) is crucial for preventing unsafe actions

*Table 28.* Ablations of NEURONCTRL.

| Variant | Cost ↓ | Safety ↑ | Viol. intensity ↓ | Safety margin ↑ | Rollout time ↓ |
|---|---|---|---|---|---|
| Baseline | **0.0610** | **0.841** | **0.180** | **0.750** | 1.000 |
| No-FMBF | 0.0683 | 0.774 | 0.225 | 0.660 | 0.950 |
| No-kernel | 0.0659 | 0.799 | 0.207 | 0.690 | 1.020 |
| No-meanflow | 0.0640 | 0.816 | 0.194 | 0.712 | 1.000 |
| Student single-step | 0.0701 | 0.757 | 0.234 | 0.637 | **0.850** |
| MLP policy | 0.0672 | 0.815 | 0.205 | 0.700 | **0.75** |
| No-consistency | 0.0671 | 0.782 | 0.216 | 0.675 | 1.000 |
| No positional encoding | 0.0647 | 0.807 | 0.198 | 0.705 | 1.000 |
| Low latent dim | 0.0720 | 0.748 | 0.243 | 0.622 | 0.920 |
| Low hidden dim | 0.0744 | 0.732 | 0.252 | 0.608 | 0.900 |

at the source, complementing the field-level barrier filter. The No-kernel variant shows degraded performance, confirming that discretization-invariant kernel aggregation in the graph neural operator (Section 2.2.3) is essential for handling varying node sets across morphologies. The removal of mean-flow regularization increases cost (0.0640 vs. 0.0610), indicating that this component helps stabilize short-horizon rollouts used in safety filtering. The Student single-step variant reduces latency (0.850 vs. 1.000, 15% reduction) at the cost of increased control cost (0.0701 vs. 0.0610, 15% increase) and reduced safety (norm. safety 0.757 vs. 0.841, safety margin 0.637 vs. 0.750), demonstrating a performance–latency trade-off for distillation-based acceleration (Section 2.2.4). Despite the performance degradation, the barrier filter remains effective in maintaining safety constraints. The MLP policy variant replaces flow matching with a simple MLP that directly maps $(z_t, p)$ to actions via supervised regression on the same training targets. While the MLP achieves lower latency (0.75 vs. 1.000, 25% reduction) due to eliminating ODE integration, it shows degraded performance: cost increases by 10% (0.0672 vs. 0.0610), safety decreases by 3% (0.815 vs. 0.841), violation intensity increases by 14% (0.205 vs. 0.180), and safety margin decreases by 7% (0.700 vs. 0.750). This degradation demonstrates that flow matching is essential rather than merely a modeling convenience: the ODE integration path enables better modeling of multimodal action distributions, and the FMBF mechanism naturally integrates safety constraints during generation, whereas the MLP's deterministic mapping struggles to capture complex dependencies in the action space. The performance and safety losses outweigh the latency benefit, confirming that flow matching is necessary for high-quality, safe control. The No-consistency variant shows degraded performance, highlighting the importance of consistency training for the surrogate simulator's accuracy. The removal of positional encoding (No positional encoding) and reduced architectural capacity (Low latent dim, Low hidden dim) lead to performance degradation, confirming that sufficient representational capacity is necessary for the complex neuromodulation tasks. Overall, the ablation results validate NEURONCTRL's modular design, where each component contributes meaningfully to the overall performance–safety–latency trade-offs.

### A.14. Hyperparameter Sensitivity

We conduct one-dimensional parameter sweeps across twelve key hyperparameters spanning training optimization, architectural capacity, inference configuration, and safety filtering behavior. For each hyperparameter, we train and evaluate configurations on the validation split using the same protocol as test time, holding all other settings fixed at baseline values.

Figure 14 provides sensitivity analysis across key hyperparameters. The inference steps sweep shows that increasing steps improves cost and safety up to a point, with latency increasing linearly. The learning rate and weight decay sweeps demonstrate robust performance across a wide range due to stage-wise training. The kernel sigma parameter shows clear optimal ranges, confirming the importance of proper kernel bandwidth selection. The architectural parameters (encoder/dynamics layers) demonstrate capacity–performance trade-offs with diminishing returns beyond optimal sizes.

### A.15. Long-Horizon Control Evaluation

To assess the impact of control horizon length on closed-loop performance and to support claims about long-horizon control capabilities, we conduct a systematic evaluation across three benchmarks (DBS3D, ECS3D, KDyn3D) with control horizons $H \in \{16, 32, 64\}$. This experiment compares six representative methods: NEURONCTRL, Latent-FM (Lipman et al., 2023), CL-DiffPhyCon (Wei et al., 2024), SafeDiffCon (Hu et al., 2025), TranSolver (Wu et al., 2024), and GOAT. For each method and horizon value, we report the closed-loop control cost, safety metrics (normalized safety score, constraint violation intensity, safety margin), and rollout latency. All evaluations use the same test split and constraint specifications as the main

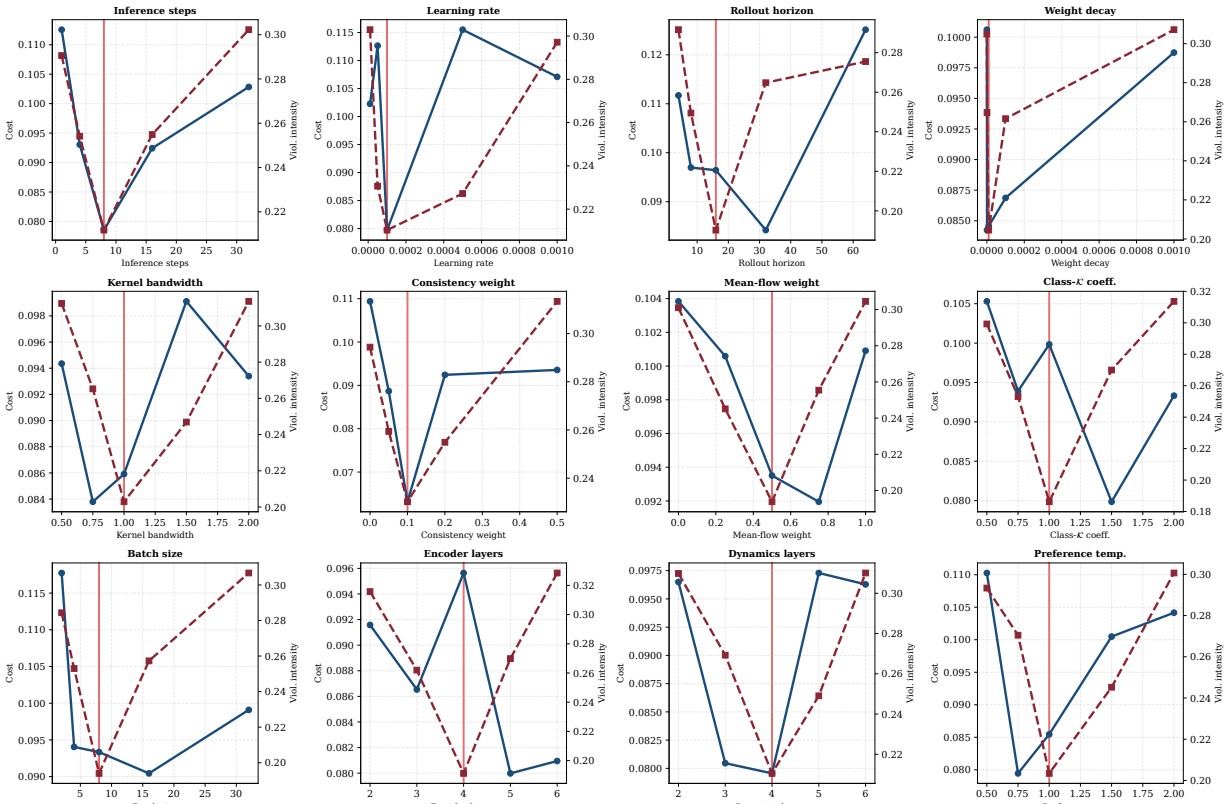

*Figure 14.* Parameter sweep results for NEURONCTRL. Each subplot shows how performance (Cost, Violation intensity, Rollout time) varies with the parameter value. Top row: Inference steps, Learning rate, Rollout horizon, Weight decay. Middle row: Kernel bandwidth, Consistency weight, Mean-flow weight, Class-$\mathcal{K}$ coeff. Bottom row: Batch size, Encoder layers, Dynamics layers, Preference temp.

experiments. Results are aggregated as episode means over the test trajectories (per seed). Figure 15 reports the performance metrics.

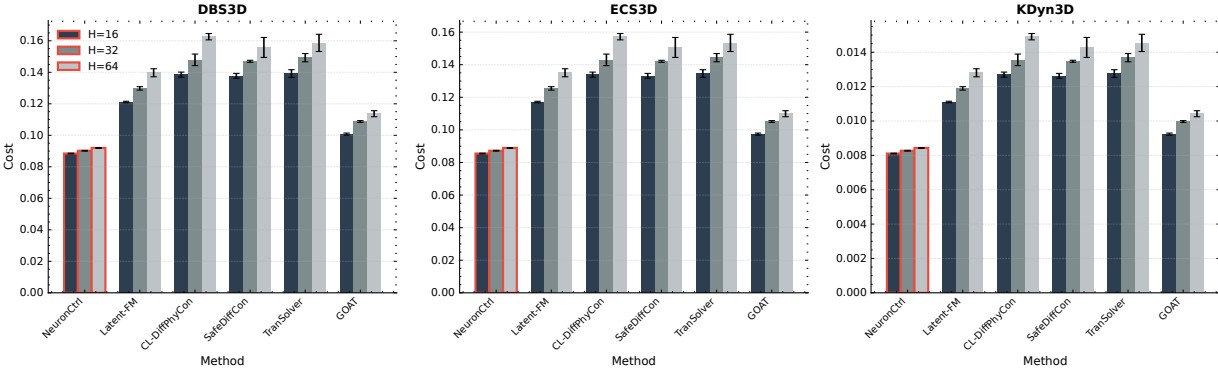

*Figure 15.* Long-horizon control evaluation across three benchmarks. Entries are means $\pm$ standard deviation over test trajectories (3 seeds: 41, 42, 43). Each subplot shows performance metrics (Cost) for different methods under control horizons H=16, 32, 64. (Left) DBS3D. (Middle) ECS3D. (Right) KDyn3D.

Figure 15 evaluates performance across different control horizons, demonstrating NEURONCTRL's ability to handle long-horizon control tasks. NEURONCTRL maintains superior performance across all horizons, with relatively stable cost as horizon increases, indicating that the surrogate simulator's short-horizon rollout training (Section 2.2.3) enables effective long-horizon control through iterative closed-loop application. The barrier-functional safety filter (Section 2.2.4) ensures safety across all horizons by enforcing field-level constraints at each step, preventing constraint violations that could

accumulate over long horizons. In contrast, methods without explicit safety mechanisms show increasing cost or safety degradation with longer horizons. The graph-based methods (GOAT, GINO) show better long-horizon performance than point-cloud methods, as the graph structure's explicit encoding of topology enables more stable multi-step predictions. NEURONCTRL's modular design—where the observer reconstructs states, the simulator predicts dynamics, and the safety filter ensures constraints—enables robust long-horizon control by handling uncertainty and maintaining safety at each step, rather than relying on single long-horizon predictions that are prone to compounding errors.

### A.16. Control Process Visualization

To provide intuitive understanding of the control dynamics and comparative performance across baseline methods, we present detailed visualizations of the control process across three dimensions: (i) spatial field evolution at key time points, (ii) state mean trajectory convergence, and (iii) action sequence characteristics.

**Spatial field snapshots.** Figure 16 displays the spatial distribution of the primary state variable at three representative time points during closed-loop control: initial state ($t_0$), mid-horizon ($t_m$), and terminal state ($t_1$). Each row corresponds to one benchmark task (DBS3D, ECS3D, KDyn3D), illustrating how the field evolves from an initial pathological configuration toward the target setpoint under NEURONCTRL's control policy.

**State mean trajectory comparison.** Figure 17 compares the temporal evolution of the spatially-averaged state variable across four methods: Ground Truth (oracle controller with full system access), NEURONCTRL, HAMLET (Bryutkin et al., 2024), and TranSolver (Wu et al., 2024). These trajectories reveal critical differences in convergence speed, stability, and safety margin maintenance.

**Action trajectory comparison.** Figure 18 visualizes the control action sequences generated by each method. The action smoothness and magnitude patterns provide insights into policy stability and energy efficiency.

The visualizations confirm that NEURONCTRL achieves effective field-level control with faster convergence and smoother action trajectories compared to baselines, complementing the quantitative metrics in Table 2.

## B. Details of Datasets

This appendix provides a detailed and self-contained account of the three datasets used in *NeuronControl*: **DBS3D**, **ECS3D**, and **KDyn3D**. Because the main paper emphasizes modeling and control methodology under strict space constraints, we document here the task motivations, simulation-based sample generation, trajectory serialization into an offline reinforcement learning (RL) interface, spatial discretizations (graph/voxel/point cloud), and the full preprocessing protocol (splits, windowing, and normalization). A key design principle is *provenance clarity*: every trajectory is generated *from scratch* by biophysically grounded simulation, and every trajectory is *self-describing*, containing the time base, geometry, recorded state variables, controller metadata, and control signals required to train models without task-specific hard-coded assumptions.

All simulators are built on the NEURON engine (Hines & Carnevale, 1997), which provides numerically stable integration of compartmental neuronal dynamics; ECS3D additionally uses NEURON's reaction–diffusion (RxD) formalism (McDougal et al., 2013) to simulate extracellular chemical dynamics in three dimensions. Across tasks, we adopt consistent units and naming conventions: time is recorded in milliseconds, spatial coordinates in micrometers, and physical state variables retain their simulator-native units (e.g., membrane potential in mV and concentrations in mM), with normalization applied only as a preprocessing layer for learning. All simulation, preprocessing, and evaluation settings are configuration-driven and the effective parameters (including timing, discretization, and random seeds) are recorded in metadata, enabling faithful reproduction without untracked task-specific logic.

### B.1. From Biophysical Simulation to Offline Closed-Loop Episodes

Each dataset is generated by repeatedly running a closed-loop simulation in which a controller modulates stimulation or source parameters as a function of measured observations. The result is an *offline* dataset $\mathcal{D} = \{\tau^{(i)}\}_{i=1}^{N_{\text{epi}}}$ of episodic trajectories, where each trajectory

$$\tau = \left\{ (\mathbf{s}_t, \mathbf{o}_t, \mathbf{a}_t, r_t, d_t) \right\}_{t=0}^{T-1} \tag{25}$$

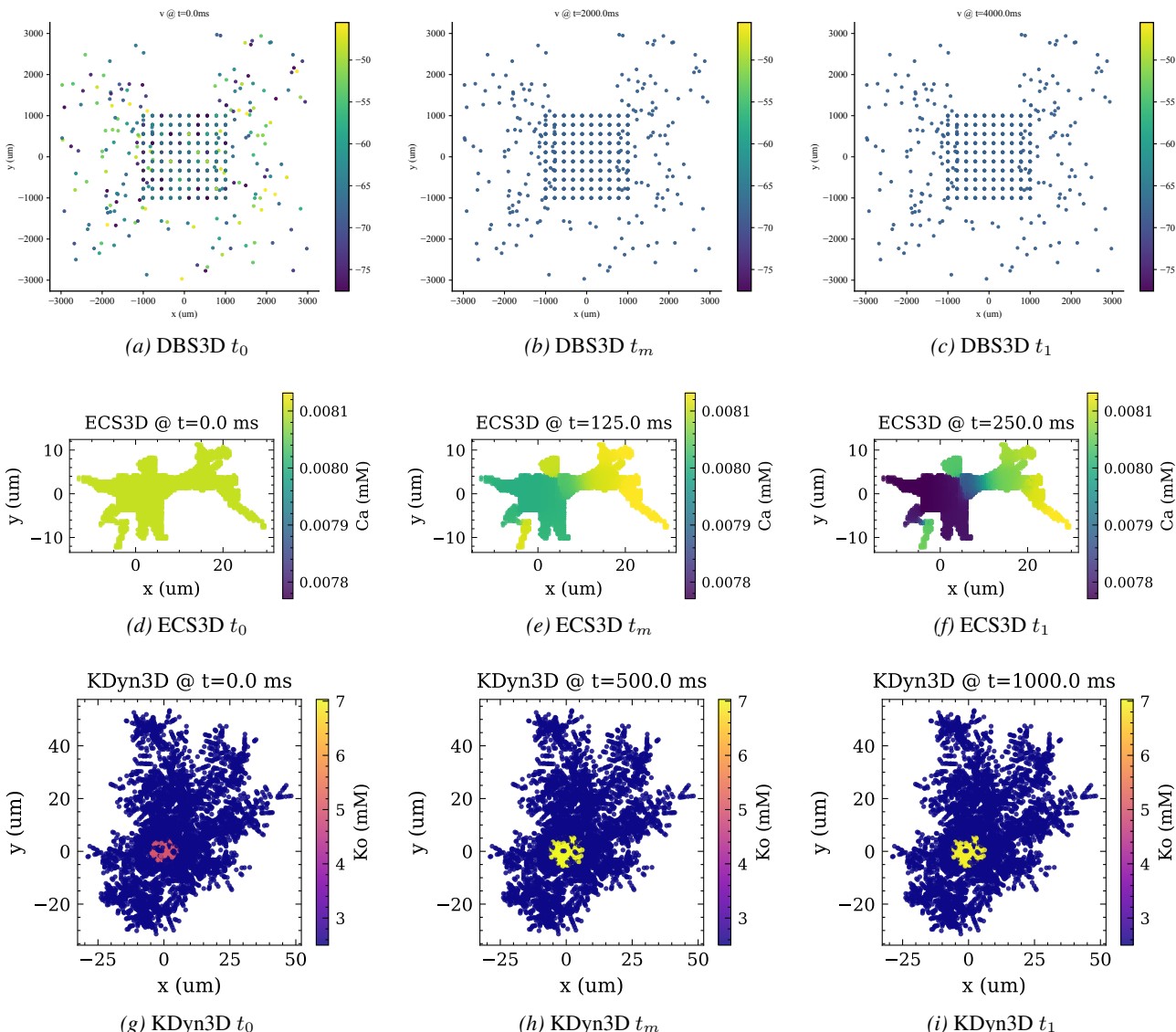

*Figure 16.* Spatial field snapshots during closed-loop control across three benchmarks. Each row shows the evolution of the primary state variable (DBS3D: membrane potential $v$; ECS3D: calcium concentration Ca; KDyn3D: extracellular potassium $K_o$) from initial pathological state ($t_0$) through mid-horizon ($t_m$) to terminal state ($t_1$). The progressive homogenization of field values toward the target setpoint demonstrates effective spatial control under NEURONCTRL's policy.

contains (i) a high-dimensional biophysical state field $\mathbf{s}_t$ over a 3D domain (stored either on a graph or voxel grid), (ii) an auxiliary low-dimensional observation vector $\mathbf{o}_t$ used for reward/termination and for sensor-limited baselines, (iii) a continuous action vector $\mathbf{a}_t$ describing stimulation or source parameters, (iv) a scalar reward $r_t$, and (v) a terminal flag $d_t \in \{0, 1\}$. This organization explicitly distinguishes *latent physical state* from *measured control observations*, which is important because several tasks are naturally sensor-limited: the reward depends on aggregate signals (e.g., beta power or mean extracellular potassium) even when the simulator produces full 3D fields.

Biophysical dynamics are integrated at a fine numerical step $\Delta t$, saved at interval $\Delta t_{\text{save}}$, while the controller updates at period $T_s$. Actions are held piecewise-constant between controller updates, yielding a uniform offline learning interface across tasks.

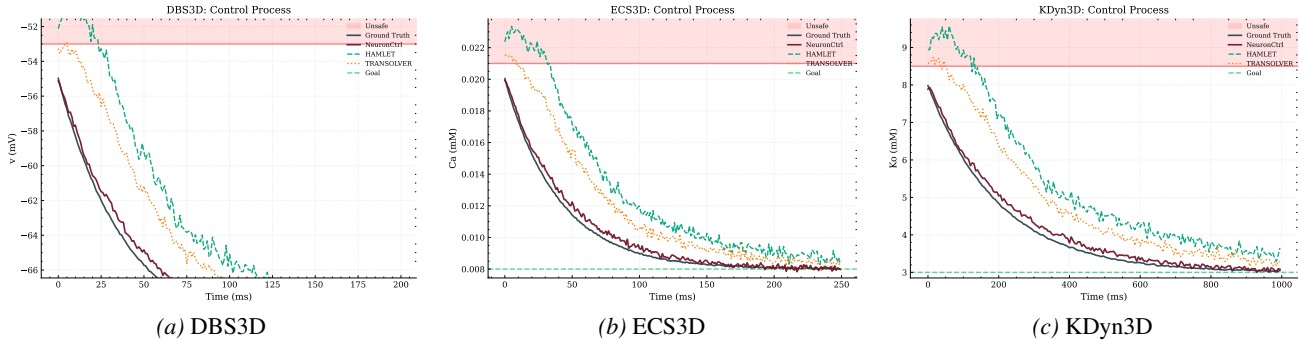

*(a)* DBS3D               *(b)* ECS3D               *(c)* KDyn3D

*Figure 17.* State mean trajectory comparison across baseline methods. Each subplot shows the spatially-averaged state variable over time for Ground Truth (oracle), NEURONCTRL, HAMLET, and TranSolver. The dashed horizontal line indicates the target setpoint, and the shaded red region marks the unsafe constraint boundary. NEURONCTRL achieves the fastest convergence among learning-based methods while maintaining smooth trajectories that remain within safety constraints throughout the control horizon, whereas HAMLET and TranSolver exhibit constraint violations during the early control phase.

## B.2. Spatial Representations

NeuronControl supports three complementary spatial representations: (i) *graph* view with node coordinates and $k$NN edges, (ii) *voxel* view on a structured 3D grid, and (iii) *point-cloud* view derived from voxel centers. All representations are deterministically derived from the same underlying simulation.

## B.3. Preprocessing

Preprocessing converts raw simulator outputs into a learning-ready format with immutable train/validation/test splits (at the episode level to avoid leakage), temporal windowing with dataset-specific $(L, S)$ parameters, and per-channel normalization (min–max or z-score) fit only on the training split.

## B.4. Dataset-Specific Generation Details

We now describe the three datasets, emphasizing task motivation, simulator dynamics, controller interface, and recorded variables. Across all tasks, the *behavior policy* used to generate trajectories is a feedback controller operating under action bounds, so the datasets are suitable for offline RL and for dynamics/operator learning under distribution shift (Kumar et al., 2020; Fujimoto et al., 2019). The resulting logged controls form the target actions used for imitation-style action generation in Sec. 2.2.4. Because the offline visitation distribution is induced by the behavior controller, purely imitative training can exhibit covariate shift under closed-loop deployment; we therefore stress-test controllers under sensing/specification shifts and enforce online safety filtering at deployment (Sec. 3.2).

### B.4.1. DBS3D: DEEP BRAIN STIMULATION WITH BETA SUPPRESSION

DBS3D models closed-loop deep brain stimulation (DBS) in a cortico–basal-ganglia–thalamic network motivated by Parkinsonian beta-band pathology (Fleming et al., 2020). The control goal is to suppress pathological beta activity while minimizing stimulation intensity. The low-dimensional observation $\mathbf{o}_t$ contains a local field potential (LFP) proxy measured in the subthalamic nucleus (STN) and a scalar beta-power summary derived from this LFP. The action $\mathbf{a}_t$ parameterizes DBS stimulation (frequency and amplitude), and the reward penalizes deviation from a beta-power setpoint together with optional control costs.

**Simulation protocol and timing.** We integrate the biophysical network at numerical step $\Delta t = 0.05$ ms and save trajectories at save interval $\Delta t_{\text{save}} = 0.5$ ms. Each episode includes a warm-up interval of $6000$ ms to reach a steady oscillatory regime followed by a $2000$ ms controlled rollout, yielding $T = 4001$ saved frames. The closed-loop controller updates every $T_s = 20$ ms.

**Pathological state generation.** We generate pathological beta regimes by injecting an exogenous beta-burst drive into the cortical population, producing sustained beta-band oscillations in downstream nuclei and a measurable increase in the STN

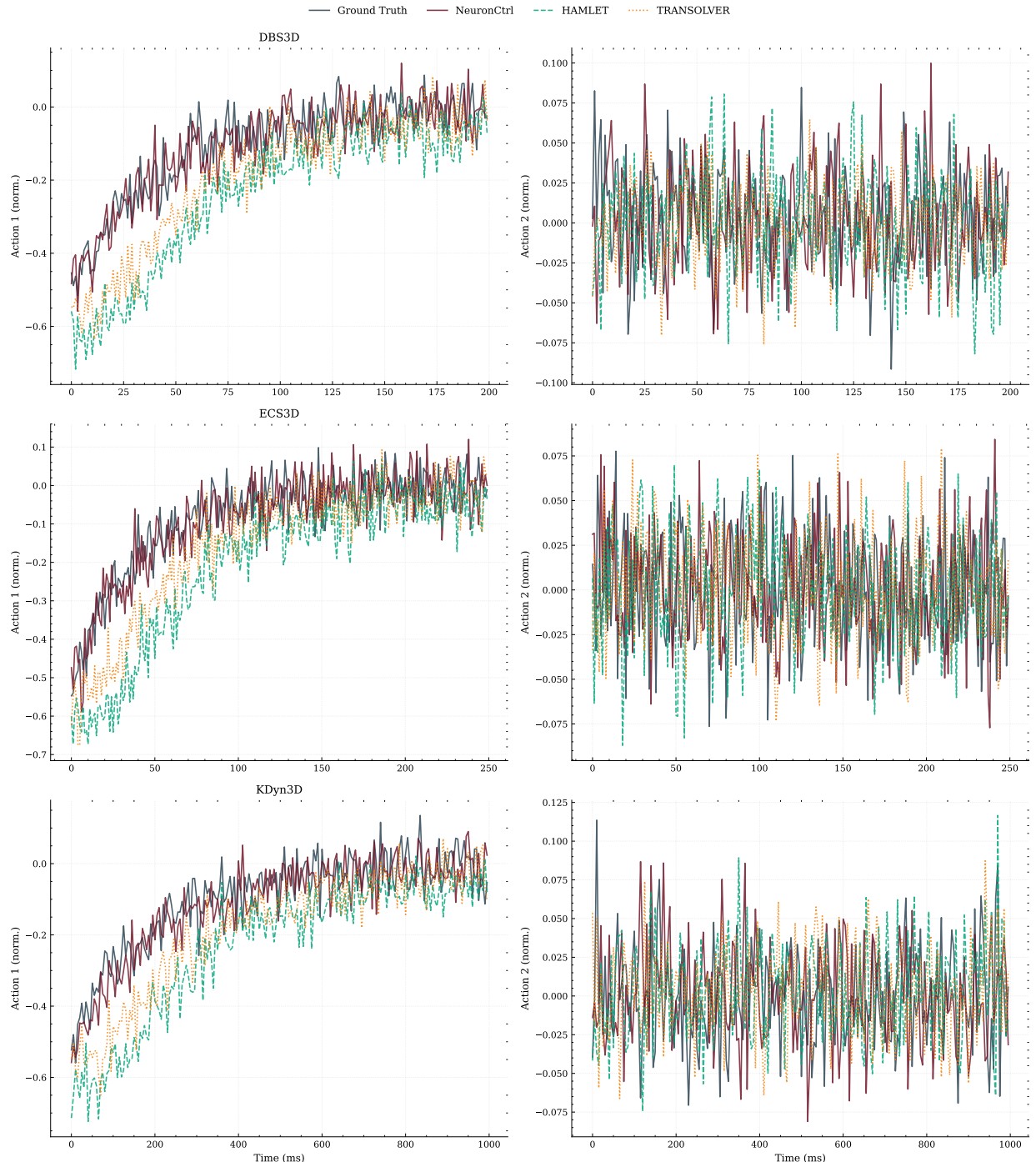

*Figure 18.* Control action trajectory comparison across benchmarks and methods. Each row corresponds to one task (DBS3D, ECS3D, KDyn3D), with columns showing the two action channels. NEURONCTRL produces smoother action trajectories with lower variance compared to HAMLET and TranSolver, indicating more stable policy behavior. The Ground Truth controller exhibits the smoothest actions as expected from oracle access.

beta-power biomarker. Burst timing and relative burst amplitudes follow a fixed schedule, while a global severity scale $s_\beta$ controls the overall strength of the drive. In our default setting, we set $s_\beta = 0.02$ for the pathological group and disable the drive in the normal group ($s_\beta = 0$), yielding a baseline regime without beta elevation.

**Episode diversity and randomness control.** In addition to the group-level beta-drive switch, DBS3D uses per-episode simulator random seeds to induce stochastic variability in spike generation and network dynamics. Episodes are generated under a deterministic seeding scheme $s_i = s_0 + i$ with base seed $s_0 = 42$. The generator also supports optional space-filling parameter sweeps via Latin Hypercube Sampling over any scalar simulator parameter declared as an interval, though the default DBS3D release does not rely on this hook beyond the group-level pathology specification.

**LFP proxy and beta-power extraction.** Let $\mathbf{p}_{\text{lfp}}$ be a fixed electrode position and $\mathbf{p}_n$ the position of a contributing STN element. DBS3D constructs an LFP proxy by an inverse-distance weighted sum of synaptic currents, scaled by tissue conductivity $\sigma$:

$$\text{LFP}(t) = \frac{1}{4\pi\sigma} \sum_{n=1}^{N_{\text{STN}}} \frac{I_n^{\text{AMPA}}(t) + I_n^{\text{GABA}_A}(t)}{\|\mathbf{p}_n - \mathbf{p}_{\text{lfp}}\|_2}. \tag{26}$$

Beta power is then computed by band-pass filtering the LFP in a beta range (a narrow band around the low beta regime), followed by rectification and short-horizon averaging to produce a smooth scalar $\beta_t$ suitable for feedback control.

**Reward and termination.** Let $\beta^\star$ denote the beta setpoint. The per-step reward uses a relative squared error

$$e_t = \frac{\beta_t - \beta^\star}{\beta^\star}, \qquad r_t = -e_t^2 - \lambda_{\text{amp}} \, \text{amp}_t^2 - \lambda_{\text{freq}}(\text{freq}_t - f_0)^2, \tag{27}$$

where $(\text{amp}_t, \text{freq}_t)$ are action components and $f_0$ is a nominal stimulation frequency. Episode termination can encode sustained achievement of the setpoint (e.g., holding $|e_t|$ below a tolerance for a minimum duration); the corresponding tolerance and hold-time conventions are recorded in controller metadata.

**Recorded state and spatial discretizations.** Beyond the low-dimensional measurements, DBS3D provides high-dimensional membrane-potential fields over a 3D discretization. In the graph view, nodes correspond to modeled elements and provide a node-wise voltage field $\mathbf{X}_t \in \mathbb{R}^{N \times 1}$; graphs are constructed by a deterministic $k$NN rule with $k = 16$ and distance-weighted edges. In the voxel view, the voltage field is voxelized to $\mathbf{V}_t \in \mathbb{R}^{n_x \times n_y \times n_z \times 1}$ using deterministic aggregation (Section B.2) with voxel spacing $25\,\mu$m (axis cap 256). Representative spatial sizes are $N \approx 600$ nodes for graph data and a 2D-like voxel slab with shape approximately $239 \times 238 \times 1$ for voxel data.

### B.4.2. ECS3D: EXTRACELLULAR REACTION–DIFFUSION CONTROL

ECS3D simulates extracellular reaction–diffusion dynamics in three dimensions around a fixed neuronal morphology, using NEURON RxD (McDougal et al., 2013; 2022). The task is motivated by chemical signaling and wave propagation phenomena (e.g., calcium dynamics coupled to secondary messengers). The high-dimensional state consists of spatial concentration fields of two species (extracellular $Ca^{2+}$ and $IP_3$), recorded either on a graph discretization or as a voxel grid.

**Pathological state generation and sample diversity.** To obtain challenging out-of-equilibrium regimes, we create two study groups by varying key kinetic gains that govern Ca–$IP_3$ coupling: an $IP_3$-receptor gain $g_{\text{IP3R}}$ and a SERCA pump strength $g_{\text{SERCA}}$. For each episode, we sample $(g_{\text{IP3R}}, g_{\text{SERCA}})$ using Latin Hypercube Sampling over group-specific intervals: normal $g_{\text{IP3R}} \in [10{,}000, 14{,}000]$, $g_{\text{SERCA}} \in [0.35, 0.45]$; pathological $g_{\text{IP3R}} \in [4{,}000, 8{,}000]$, $g_{\text{SERCA}} \in [0.10, 0.30]$. This space-filling sampling yields diverse regimes within a fixed episode budget.

**Simulation protocol and discretization.** We integrate RxD dynamics at $\Delta t = 0.1$ ms and save at $\Delta t_{\text{save}} = 1.0$ ms over a horizon of 100 ms, yielding $T = 101$ saved frames. The voxel discretization uses spacing $0.5\,\mu$m with an axis cap of 256 to bound memory usage. The graph representation is constructed deterministically by a $k$NN rule with $k = 16$ and distance-weighted edges, using the same underlying spatial coordinates as the voxel grid.

**Dynamics.** At a conceptual level, ECS3D evolves concentration fields $c(\mathbf{x}, t)$ under a reaction–diffusion PDE with controlled source terms:

$$\frac{\partial c}{\partial t} = D\nabla^2 c + \mathcal{R}(c, \ldots) + \sum_{m=1}^{d_a} a_m(t)\, S_m(\mathbf{x}), \tag{28}$$

where $D$ is a diffusion coefficient, $\mathcal{R}$ denotes local reaction kinetics, and $S_m(\mathbf{x})$ are spatially localized source profiles controlled by the action components. The simulator integrates this system over a short horizon with millisecond-resolution saving, producing trajectories with $T = 101$ frames.

**Control interface and reward.** The action vector $\mathbf{a}_t$ modulates the amplitudes of two source terms with bounded magnitude (each bounded to $[-10^{-3}, 10^{-3}]$ in the released configuration). The low-dimensional observation $\mathbf{o}_t$ includes a scalar summary of concentration used for feedback tracking (we track calcium by default). With setpoint $\mathrm{sp} = 1.7 \times 10^{-3}$ and controller period $T_s = 50$ ms, the reward uses relative squared tracking error with a quadratic control penalty:

$$e_t = \frac{y_t - \mathrm{sp}}{\mathrm{sp}}, \qquad r_t = -e_t^2 - \lambda_{\mathrm{amp}} \|\mathbf{a}_t\|_2^2, \tag{29}$$

where $y_t$ is the selected measured channel.

**Recorded state and discretizations.** ECS3D records two spatial channels, Ca and IP$_3$. In the graph view, the state is $\mathbf{X}_t \in \mathbb{R}^{N \times 2}$ over a large geometric graph with $N \approx 9{,}071$ nodes and $E \approx 164{,}578$ directed edges in a representative configuration. In the voxel view, the state is stored on a 3D grid with shape approximately $86 \times 48 \times 19$ at sub-micrometer spacing; an occupancy mask is provided when the computational domain is not densely filled, enabling sparse point-cloud extraction.

### B.4.3. KDYN3D: ASTROCYTIC POTASSIUM DYNAMICS WITH FEEDBACK CONTROL

KDyn3D models potassium dynamics in a detailed astrocyte geometry and provides a long-horizon, high-dimensional control benchmark (Savtchenko et al., 2018). The task is motivated by the role of astrocytes in extracellular potassium buffering and the strong coupling between ionic concentrations and membrane dynamics. KDyn3D provides multiple biophysical channels including membrane voltage, intra- and extracellular potassium, and potassium-related currents, yielding a state field with six channels.

**Pathological state generation.** We define a pathological group by reducing the conductance of an inward-rectifying potassium channel mechanism (Kir4) relative to the normal range. For each episode, the Kir4 conductance $g_{\mathrm{Kir4}}$ is sampled using Latin Hypercube Sampling over a group-specific interval (normal $g_{\mathrm{Kir4}} \in [0.08, 0.12]$; pathological $g_{\mathrm{Kir4}} \in [0.01, 0.05]$), and the resulting dynamics are simulated under the same geometry and recording protocol. This parameter shift yields trajectories with altered potassium buffering behavior and constitutes the primary benchmark distribution for long-horizon regulation.

**State channels and observations.** The recorded spatial channels include membrane potential $v$, intracellular potassium $k_i$, extracellular potassium $k_o$, and potassium-current components (e.g., inward-rectifying $\mathrm{K}^+$ currents and pump/transport terms). The low-dimensional observation $\mathbf{o}_t$ is constructed from spatial averages such as $(\overline{v}_t, \overline{k_{ot}})$, where $\overline{k_{ot}}$ denotes the mean extracellular potassium over the domain at time $t$.

**Control interface and reward.** Actions $\mathbf{a}_t$ parameterize a controlled extracellular potassium source, including its injected concentration and a spatial scale parameter (radius). In the released configuration, we use $\Delta t = 0.1$ ms, save at $\Delta t_{\mathrm{save}} = 1.0$ ms over $t_{\mathrm{stop}} = 1000$ ms (yielding $T = 1001$ frames), and apply control at $T_s = 50$ ms with potassium setpoint $\mathrm{sp} = 3.0$ mM. The injected concentration is bounded to $[2.5, 12.0]$ mM and the radius is bounded to $[2.0, 10.0]$ $\mu$m. The reward is

$$e_t = \frac{\overline{k_{ot}} - \mathrm{sp}}{\mathrm{sp}}, \qquad r_t = -e_t^2 - \lambda_k a_{1,t}^2 - \lambda_r a_{2,t}^2, \tag{30}$$

where $(a_{1,t}, a_{2,t})$ denote the two action components.

**Spatial discretization.** In the graph view, KDyn3D yields large graphs ($N \approx 20{,}145$, $E > 10^6$ raw edges); optional edge-budget sparsification is applied to ensure tractable message-passing. In the voxel view, the same six channels are recorded on a compact 3D grid ($40 \times 44 \times 12$ voxels).

## B.5. Dataset Statistics and Recommended Windowing

Table 29 summarizes the processed dataset sizes and representative spatiotemporal resolutions for the released splits. Counts are reported per representation when the availability differs by modality (e.g., voxel data may be available for fewer episodes than graph data due to the higher cost of dense 3D fields). Windowing $(L, S)$ and controller period $T_s$ are included to clarify the temporal context given to learning algorithms.

*Table 29.* Dataset statistics. Each dataset provides graph and voxel representations. $T$ denotes saved frames per episode. Windowing $(L, S)$ and $T_s$ specify the temporal context.

| Dataset | Episodes (tr/va/te) | $T$ | Spatial size (repr.) | Channels | Window $(L, S)$ / $T_s$ |
|---|---|---|---|---|---|
| DBS3D | 710/150/160 | 4001 | Graph: $N \approx 600$, $E \approx 1.16 \times 10^4$; Vox: $239 \times 238 \times 1$ | $v$ | $(2, 1)$ / 20 ms |
| ECS3D | 3500/750/750 | 101 | Graph: $N \approx 9.07 \times 10^3$, $E \approx 1.65 \times 10^5$; Vox: $86 \times 48 \times 19$ | Ca, IP$_3$ | $(8, 4)$ / 50 ms |
| KDyn3D | 1980/420/440 | 1001 | Graph: $N \approx 2.01 \times 10^4$, $E > 10^6$ (raw); Vox: $40 \times 44 \times 12$ | 6 ch. | $(8, 4)$ / 50 ms |

**Data generation cost and sample efficiency considerations.** Generating trajectories from high-fidelity NEURON/RxD simulators is computationally expensive: each episode requires 15–45 minutes of wall-clock time on a single CPU core, depending on the task complexity and spatial discretization. For the full datasets reported in Table 29, total generation time ranges from approximately 180 hours (DBS3D, 710 episodes) to 2,600 hours (ECS3D, 3500 episodes). This computational cost limits dataset size, which is critical for neuromodulation applications where only limited simulation or clinical data may be available.

### B.5.1. PRACTICAL GUIDANCE FOR MODEL USAGE

Across all datasets, policies and dynamics models can be trained either (i) on the full spatial fields $\mathbf{s}_t$ (graph/voxel/point), (ii) on the low-dimensional measurement vector $\mathbf{o}_t$ alone (sensor-limited baselines), or (iii) on a fusion of both. Because reward and termination are always functions of $\mathbf{o}_t$ and $\mathbf{a}_t$, experiments that compare representation learning methods remain well-defined even when the spatial field is high-dimensional: the learning objective is to choose actions that regulate a physically meaningful measured quantity under realistic closed-loop dynamics.

Taken together, these datasets provide a controlled yet challenging testbed for spatiotemporal representation learning and offline control in biophysically grounded neural systems, with transparent provenance from simulation parameters to learning-ready trajectories.

## B.6. Action Ground Truth: Controller Design Specifications

All behavior-policy controllers operate in closed-loop mode, updating actions at fixed control periods based on normalized tracking errors. The logged actions constitute ground truth supervision for training learning-based policies.

*Table 30.* Controller specifications for action ground truth generation across benchmarks. All controllers use normalized error $e_t = (y_t - \mathrm{sp})/\mathrm{sp}$ and proportional-style updates with action clipping.

| Parameter | DBS3D | ECS3D | KDyn3D |
|---|---|---|---|
| Control period $T_s$ | 20 ms | 50 ms | 50 ms |
| Observation | Beta power $\beta_t$ | Mean Ca | Mean $K_o$ |
| Controller type | Dual PID | Proportional | Proportional |
| Action ch. 1 | Amplitude | Source amp. 1 | Concentration |
| Bounds | metadata | $[-10^{-3}, 10^{-3}]$ | $[2.5, 12.0]$ mM |
| Gain | $K_p$ (metadata) | $\eta = 0.5$ | $\eta_k = 0.5$ |
| Action ch. 2 | Frequency | Source amp. 2 | Radius |
| Bounds | metadata | $[-10^{-3}, 10^{-3}]$ | $[2.0, 10.0]$ $\mu$m |
| Gain | $0.4K_p$ | $\eta = 0.5$ | $\eta_r = 0.25$ (mult.) |

# C. Details of Models

This appendix complements Sec. 2 by adding technical details that are omitted from the main text for space. Rather than restating the problem formulation (Sec. 2.1) or the overall framework (Sec. 2), we focus on: (i) an operator-theoretic view connecting biophysical field dynamics to the residual time-stepping parameterization of $G_\theta$ (Sec. 2.2.3); (ii) practical construction of heterogeneous morphology graphs across cable and ECS discretizations; (iii) the exact heterogeneous graph neural operator block used in experiments, including geometry-aware control conditioning; (iv) training and stabilization practices for reliable differentiable rollouts; and (v) implementation-level details for the observer and barrier modules beyond the abstractions in Sec. 2.2.2 and Sec. 2.2.4.

## C.1. Design Rationale

The main text motivates NEURONCTRL at the algorithmic level; here we briefly clarify the engineering considerations behind three choices that materially affect training stability and deployment latency. First, we train the learned simulator, observer, and barrier in stages (Sec. 2.2.5) to avoid tightly coupled non-stationarities during optimization: changing $G_\theta$ alters the planner-induced state distribution, changing $\mathcal{O}_\phi$ alters both planning and safety inputs, and changing $B_\psi$ alters the feasible action set through filtering. Second, we parameterize $G_\theta$ as a residual update (Eq. (11)) to align learning with time-discretized physics and to stabilize repeated composition of the transition map under MPC differentiation. Third, we enforce barrier constraints via a linearized projection (Eqs. (18)–(19)), which preserves the minimal-intervention property while enabling predictable compute in low-dimensional action spaces.

**Optional joint fine-tuning.** When enabled, joint fine-tuning minimizes a weighted sum of component losses:

$$\mathcal{L}_{\text{joint}} = \lambda_{\text{wm}}\,\mathcal{L}_{\text{wm}} + \lambda_{\text{policy}}\,\mathcal{L}_{\text{policy}} + \lambda_{\text{consist}}(e)\,\mathcal{L}_{\text{consist}} + \lambda_{\text{mf}}\,\mathcal{L}_{\text{mf}}, \tag{31}$$

where $\mathcal{L}_{\text{wm}}$ denotes the world-model objective (including the simulator loss in Eq. (12)), $\mathcal{L}_{\text{policy}}$ is the flow matching loss in Eq. (14), $\mathcal{L}_{\text{consist}}$ enforces closed-loop rollout consistency, and $\mathcal{L}_{\text{mf}}$ is mean-flow regularization. The consistency term compares predicted rollout states $\{\tilde{u}_{t+k}\}_{k=1}^{R}$ to ground truth $\{u_{t+k}\}_{k=1}^{R}$:

$$\mathcal{L}_{\text{consist}} = \frac{1}{\sum_{k=1}^{R}\gamma^{k-1}}\sum_{k=1}^{R}\gamma^{k-1}\left\|\tilde{u}_{t+k} - u_{t+k}\right\|_2^2, \tag{32}$$

with decay $\gamma \in (0, 1]$.

## C.2. Operator View and Residual Time Stepping

We adopt the notation of Sec. 2.1. To connect $G(u_t, w_t; \mathcal{G})$ in Eq. (2) to biophysical modeling, it is useful to view neuronal dynamics as a continuous-time operator equation on a 3D domain $\Omega \subset \mathbb{R}^3$ induced by morphology and extracellular space. Let $u(t, \mathbf{x}) \in \mathbb{R}^{C_u}$ denote the latent state field and let $w(t) \in \mathbb{R}^{C_w}$ denote low-dimensional stimulation. A broad class of electrophysiology and reaction–diffusion systems can be expressed as

$$\partial_t u(t, \mathbf{x}) = \mathcal{F}\big(u(t, \cdot), w(t)\big)(\mathbf{x}), \qquad \mathbf{x} \in \Omega, \tag{33}$$

where $\mathcal{F}$ combines diffusion-like couplings, nonlinear reactions, and control sources. A generic decomposition covering our benchmarks takes the reaction–diffusion form

$$\mathcal{F}(u, w)(\mathbf{x}) = \nabla \cdot \big(D(\mathbf{x})\nabla u(\mathbf{x})\big) + R\big(u(\mathbf{x})\big) + S(\mathbf{x})\,w, \tag{34}$$

with diffusion tensor $D(\mathbf{x})$, local reaction term $R$, and spatially localized actuation $S(\mathbf{x})$.

Given simulator save interval $\Delta t$, the induced discrete-time flow map $\Phi_{\Delta t}$ satisfies

$$u_{t+1} = \Phi_{\Delta t}(u_t, w_t) = u_t + \Delta t\,\mathcal{F}(u_t, w_t) + \mathcal{O}(\Delta t^2), \tag{35}$$

where $u_t(\cdot) \triangleq u(t, \cdot)$. The residual structure in Eq. (11) can be interpreted as learning a morphology-conditioned approximation to the increment $\mathcal{F}$:

$$G_\theta(u_t, w_t; \mathcal{G}) \triangleq u_t + \Delta t\,\widehat{\mathcal{F}}_\theta(u_t, w_t; \mathcal{G}), \tag{36}$$

thereby matching the small-$\Delta t$ expansion in Eq. (35). This alignment is particularly useful in gradient-based MPC because it stabilizes repeated composition of $G_\theta$ by preserving an identity path and by learning increments whose scale is controlled by $\Delta t$.

## C.3. Constructing Heterogeneous Morphology Graphs

Sec. 2.2.1 defines the heterogeneous graph abstraction. Here we describe how we instantiate $E^{\text{cable}}$, $E^{\text{space}}$, and $E^{\text{coup}}$ in practice.

For the cable system, $E^{\text{cable}}$ is taken from the compartmental adjacency of the morphological tree (e.g., parent–child links in SWC/HOC parsing), optionally augmented with reverse edges to allow bidirectional message passing. For the extracellular (ECS) discretization, we form $E^{\text{space}}$ by either a radius graph or $k$NN graph on the ECS point cloud $\{\mathbf{x}_i\}$; both enforce locality consistent with diffusion operators, and we treat $k$ (or the radius) as a resolution-dependent hyperparameter. When a joint membrane–ECS discretization is used, $E^{\text{coup}}$ connects each membrane node to its nearest ECS nodes (typically $1$–$k_c$ neighbors), encoding transmembrane exchange or source coupling.

Node features concatenate the latent state channels with morphology descriptors $g(i)$ (Sec. 2.1); in implementation, we normalize geometric scalars (e.g., radii, segment lengths, path distances) per dataset to reduce scale sensitivity. Edge attributes use relative geometry as described in Sec. 2.1. In our implementation, the simulator consumes the relative displacement and distance $e_{ij} = [\mathbf{x}_i - \mathbf{x}_j, \|\mathbf{x}_i - \mathbf{x}_j\|_2]$ directly through learnable edge-type MLPs (Sec. C.4); we therefore do not explicitly multiply messages by a separate diffusion-style quadrature weight during the forward pass. When needed for deterministic preprocessing (e.g., graph sparsification in Sec. B.4.3), a distance-decayed weight can be computed from geometry.

## C.4. Simulator: Heterogeneous Graph Neural Operator in Experiments

The main text presents a generic typed message-passing operator and residual Euler output (Eq. (11)). Here we specify the concrete operator block used in our experiments and relate it to an operator-learning view. Typed relations are represented by a single edge list together with an integer relation-type indicator; each relation $r$ selects a dedicated edge MLP $\phi_r$, while the update MLP $\text{MLP}_{\text{up}}^{(\ell)}$ and LayerNorm are layer-specific. In the absence of per-node morphology descriptors $g(i)$, we set $g(i) \equiv 0$, and the simulator reduces to a purely state-driven transition operator on the cached geometry.

**Kernel operator perspective.** For relation $r \in \{\text{cable}, \text{space}\}$, an ideal continuous kernel operator acting on a latent feature field $h : \Omega \to \mathbb{R}^d$ can be written as

$$(\mathcal{K}_r h)(\mathbf{x}) = \int_{\Omega_r(\mathbf{x})} \kappa_r(\mathbf{x}, \mathbf{y})\, h(\mathbf{y})\, d\mathbf{y}, \tag{37}$$

where $\Omega_r(\mathbf{x})$ denotes a physics-defined local neighborhood and $\kappa_r$ is a learnable kernel. On an irregular discretization with neighborhood sets induced by $E^r$, we approximate (37) by a sum over neighbors,

$$(\mathcal{K}_r^N h)(\mathbf{x}_i) = \sum_{j \in \mathcal{N}_i^r} \kappa_r(\mathbf{x}_i, \mathbf{x}_j)\, h(\mathbf{x}_j), \tag{38}$$

which is exactly the computational pattern implemented by message passing, preserving permutation invariance of neighborhood aggregation.

**Exact operator block.** Let $h_i^{(0)} = \text{MLP}_{\text{in}}([u_t(i), g(i)]) \in \mathbb{R}^d$ and let $c_i$ denote a node-wise control-conditioning term (defined below). For $\ell = 0, \dots, L-1$ and relation type $r$, we compute messages

$$m_{i,r}^{(\ell)} = \sum_{j \in \mathcal{N}_i^r} \phi_r\left(h_j^{(\ell)}, h_i^{(\ell)}, e_{ij}\right), \tag{39}$$

then sum relation-wise messages and apply a residual normalized update,

$$m_i^{(\ell)} = \sum_r m_{i,r}^{(\ell)}, \qquad h_i^{(\ell+1)} = h_i^{(\ell)} + \text{LN}\left(\text{MLP}_{\text{up}}^{(\ell)}\left(m_i^{(\ell)} + c_i\right)\right), \tag{40}$$

where $\text{MLP}_{\text{up}}^{(\ell)}$ includes a GELU nonlinearity and LN is LayerNorm. Finally, we map $h_i^{(L)}$ to an increment $\Delta \hat{u}_t(i) = \text{MLP}_{\text{out}}(h_i^{(L)})$ and apply the residual Euler step as in Eq. (11). In the released code, the edge attribute is the relative displacement and distance $e_{ij} = [\mathbf{x}_j - \mathbf{x}_i, \|\mathbf{x}_j - \mathbf{x}_i\|_2] \in \mathbb{R}^4$, and $\phi_r$ is shared across layers for a fixed edge type. Equation (40) is an implementation-level instantiation of the generic operator: relation-specific edge-type MLPs $\phi_r$ capture heterogeneous couplings, while the residual–LN form stabilizes deep stacks and improves rollout conditioning under repeated composition.

**Geometry-aware control conditioning.** Sec. 2.2.3 describes broadcasting an embedding of $w_t$ to all nodes. In practice, actuation is spatially localized (e.g., electrodes or injection sources), and we encode this inductive bias with a smooth influence kernel computed from known stimulation geometry. Let $\xi$ denote metadata describing the stimulation locus (e.g., source coordinate $\mathbf{x}_{\mathrm{src}}$ and length scale $\rho$). We compute a scalar influence score $s_i = \kappa_s(\mathbf{x}_i; \xi)$; one example is a Gaussian kernel (extended to multiple sources by a max operator),

$$s_i = \max_k \exp\left( -\frac{\|\mathbf{x}_i - \mathbf{x}_{\mathrm{src}}^{(k)}\|^2}{2\rho^2} \right). \tag{41}$$

We then inject control through a shared embedding modulated by $s_i$,

$$c_i = \mathrm{MLP}_w\big([w_t, s_i]\big) \in \mathbb{R}^d, \tag{42}$$

which is added inside each operator block in Eq. (40). This construction reduces the burden on the network to infer *where* the action acts from data alone and helps stabilize learning when electrode/source placement varies. When stimulation geometry metadata is unavailable, we set $s_i \equiv 1$ so that the simulator reverts to a spatially uniform control embedding.

**Optional spatio-temporal operator simulator.** For completeness, we also support an alternative simulator parameterization based on a spatio-temporal operator encoder and explicit time integration (e.g., Euler or Runge–Kutta schemes with configurable sub-steps). This option exposes additional architectural choices such as convolution type and kernel-integral smoothing bandwidth. When trajectories provide an explicit time step, we use it; otherwise we treat the discrete step as unit time $\Delta t=1$.

### C.5. Training Details for Stable Differentiable Rollouts

The main loss used for training $G_\theta$ is given in Eq. (12). We highlight two implementation details that are easy to omit but important in practice.

First, the field regression norms are implemented as per-node averages to reduce sensitivity to variable graph sizes: for a prediction $\tilde{u} \in \mathbb{R}^{N \times C_u}$ we use the normalized discrete norm $\|\tilde{u} - u\|_N^2 \triangleq \frac{1}{N} \sum_{i=1}^N \|\tilde{u}(i) - u(i)\|_2^2$. Second, rollout regularization is applied with short horizons $K_{\mathrm{roll}}$ to directly control the conditioning of MPC rollouts; this targets the error amplification effect from repeated composition of the learned transition map.

A useful bound makes the role of short-horizon stability explicit. Let the true one-step map be $u_{t+1} = G(u_t, w_t; \mathcal{G})$ and the learned map be $\tilde{u}_{t+1} = G_\theta(u_t, w_t; \mathcal{G})$. Assume $G$ is Lipschitz in $u$ (for fixed $w$) with constant $L_G$, and define the one-step error $\varepsilon_t = \|G_\theta(u_t, w_t; \mathcal{G}) - G(u_t, w_t; \mathcal{G})\|_N$. Then, starting from the same initial state, the $K$-step rollout deviation satisfies

$$\|\tilde{u}_{t+K} - u_{t+K}\|_N \leq \sum_{k=0}^{K-1} L_G^{K-1-k} \varepsilon_{t+k}. \tag{43}$$

In practice, we found that controlling the effective Lipschitz behavior of $G_\theta$ via residual parameterization (Eq. (11)), normalization (Eq. (40)), and gradient clipping substantially improves rollout stability, which in turn improves the conditioning of gradients through unrolled MPC.

### C.6. Differentiating Through Rollouts: An Adjoint View

Differentiating through unrolled rollouts follows the standard adjoint recursion. The costate $\lambda_{t+h}$ propagates via $\lambda_{t+h} = \nabla_{\tilde{u}_{t+h}} \ell + (\nabla_{\tilde{u}_{t+h}} G_\theta)^\top \lambda_{t+h+1}$, and action gradients are $\nabla_{w_{t+h}} J = \nabla_{w_{t+h}} \ell + (\nabla_{w_{t+h}} G_\theta)^\top \lambda_{t+h+1}$. Stabilizing Jacobians of $G_\theta$ (Sec. C.5) prevents exploding or vanishing gradients through the horizon.

### C.7. Flow Policy and FMBF Details Beyond Sec. 2.2.4

Sec. 2.2.4 describes the preference-conditioned safe flow policy at the level of the ODE (13) and the conditional flow matching loss (14). Here we summarize the implementation choices that materially affect inference latency and safety enforcement.

**State embedding $z_t$.** We reuse the world-model graph encoder as $\mathrm{Enc}$ to map $(\hat{u}_t, \mathcal{G})$ to node latents and set $\mathrm{Agg}$ to global mean pooling followed by a two-layer MLP, yielding $z_t$ in (13).

**Conditional velocity network.** The velocity field $v_\eta(a_\tau, \tau \mid z_t)$ is parameterized by an MLP with residual blocks and LayerNorm. Following (13), the network takes the current action $a_\tau$, a sinusoidal embedding of flow time $\tau \in [0, 1]$, and a state embedding $z_t$. Optional conditioning terms are added as learned linear projections: a goal/reference embedding (from a goal MLP), a Pareto preference embedding (from a preference MLP plus learnable objective embeddings), and an extremum-return embedding (from a scalar return MLP).

**Training: OT path and time sampling.** For conditional flow matching, we sample $\tau$ using a task-dependent time schedule and use a straight-line optimal transport path between noise $a_0 \sim \mathcal{N}(0, I)$ and the dataset action $a_1$:

$$a_\tau = (1 - \tau)a_0 + \tau a_1, \qquad v^\star = a_1 - a_0. \tag{44}$$

The training loss is a weighted mean-squared error between $v_\eta(a_\tau, \tau \mid z_t, \cdot)$ and $v^\star$, with optional sample weights derived from returns and preferences. In the safe-by-construction variant, we apply the safety projection during training; otherwise the projection is applied only at inference.

**Inference: Euler integration and safety projection.** At test time, actions are generated by explicit Euler integration from $\tau = 0$ to $\tau = 1$ using a configurable number of inference steps $N_{\mathrm{FE}}$:

$$a_{\tau_{i+1}} = a_{\tau_i} + \Delta\tau \, v_\eta^{\mathrm{safe}}(a_{\tau_i}, \tau_i \mid z_t, \cdot), \qquad \Delta\tau = \frac{1}{N_{\mathrm{FE}}}. \tag{45}$$

Safety is enforced by a Flow Matching Barrier Function (FMBF) projection that modifies the velocity field at each step. Let $\{h_k(a)\}_{k=1}^K$ define an action safe set $\mathcal{C}_{\mathrm{act}} = \{a : h_k(a) \geq 0, \ k = 1, \ldots, K\}$. With class-$\mathcal{K}$ functions $\gamma_k(h) = \lambda_h h$ (shared coefficient $\lambda_h > 0$), we compute the projected velocity as the Euclidean projection onto the intersection of halfspaces:

$$v_\eta^{\mathrm{safe}} = \arg\min_v \|v - v_\eta\|_2^2 \ \text{s.t.} \ \nabla h_k(a)^\top v \geq -\gamma_k(h_k(a)), \qquad k = 1, \ldots, K. \tag{46}$$

For a single halfspace constraint, the projection admits the closed form

$$v^+ = v + \mathrm{ReLU}\big(-\nabla h_k(a)^\top v - \gamma_k(h_k(a))\big) \frac{\nabla h_k(a)}{\|\nabla h_k(a)\|_2^2}, \tag{47}$$

where we clamp $\|\nabla h_k(a)\|_2^2$ from below by $10^{-8}$ for numerical stability. For box constraints $a_{\min} \leq a \leq a_{\max}$, we instantiate the $2C_w$ affine barriers $h_j^{\mathrm{L}}(a) = a_j - a_{\min}$ and $h_j^{\mathrm{U}}(a) = a_{\max} - a_j$. In this case, (46) reduces to elementwise bounds on the velocity and the joint projection is separable:

$$v_\eta^{\mathrm{safe}} = \mathrm{clip}(v_\eta; \, -\lambda_h(a - a_{\min}), \, \lambda_h(a_{\max} - a)), \tag{48}$$

where $\mathrm{clip}(\cdot; \ell, u)$ clamps each component to $[\ell_j, u_j]$.

**Lemma C.1** (One-step box invariance under Euler integration). *Let $a_{\min} < a_{\max}$ and define the affine box barriers $h_j^{\mathrm{L}}(a) = a_j - a_{\min}$ and $h_j^{\mathrm{U}}(a) = a_{\max} - a_j$ for $j = 1, \ldots, C_w$. Let $h(a) = \min_j \min(h_j^{\mathrm{L}}(a), h_j^{\mathrm{U}}(a))$ and $\gamma(h) = \lambda_h h$ with $\lambda_h > 0$. Suppose at step $i$ the projected velocity satisfies the FMBF conditions for all box constraints, $[v_\eta^{\mathrm{safe}}]_j \geq -\lambda_h h_j^{\mathrm{L}}(a_{\tau_i})$ and $[v_\eta^{\mathrm{safe}}]_j \leq \lambda_h h_j^{\mathrm{U}}(a_{\tau_i})$, and the Euler update (45) is used with step size $\Delta\tau$. Then*

$$h(a_{\tau_{i+1}}) \geq (1 - \lambda_h \Delta\tau) \, h(a_{\tau_i}). \tag{49}$$

*In particular, if $h(a_{\tau_i}) \geq 0$ and $\lambda_h \Delta\tau \leq 1$, then $h(a_{\tau_{i+1}}) \geq 0$.*

*Proof.* For each coordinate $j$, the Euler update yields

$$h_j^{\mathrm{L}}(a_{\tau_{i+1}}) = h_j^{\mathrm{L}}(a_{\tau_i}) + \Delta\tau \, [v_\eta^{\mathrm{safe}}]_j \geq h_j^{\mathrm{L}}(a_{\tau_i}) - \Delta\tau \, \lambda_h h_j^{\mathrm{L}}(a_{\tau_i}) = (1 - \lambda_h \Delta\tau) \, h_j^{\mathrm{L}}(a_{\tau_i}),$$

and similarly,

$$h_j^{\mathrm{U}}(a_{\tau_{i+1}}) = h_j^{\mathrm{U}}(a_{\tau_i}) - \Delta\tau \, [v_\eta^{\mathrm{safe}}]_j \geq (1 - \lambda_h \Delta\tau) \, h_j^{\mathrm{U}}(a_{\tau_i}).$$

Taking the minimum over $j$ and the two-sided constraints gives $h(a_{\tau_{i+1}}) \geq (1 - \lambda_h \Delta\tau) \, h(a_{\tau_i})$. $\qquad \square$

**Discrete integration and barrier composition.** Lemma C.1 provides a step-size condition for one-step invariance; empirical action-bound violation statistics are reported in Table 17. The default FMBF barrier enforces component-wise action bounds; optionally, rate and energy constraints can be stacked via cyclic halfspace projections.

**Optional acceleration via distillation (Reflow).** We optionally distill the multi-step flow policy into a single-step student using rectified flow (Liu et al., 2023) with loss $\mathcal{L}_{\text{reflow}}(\eta) = \mathbb{E} \|\hat{a}_1 - a_1\|_2^2$, where $\hat{a}_1 = a_0 + \Delta a_\eta(a_0 \mid z_t, p)$.

## C.8. Observer Implementation Details Beyond Sec. 2.2.2

Sec. 2.2.2 describes the observer at a high level as $\hat{u}_t = \mathcal{O}_\phi(h_t; \mathcal{G})$. In our implementation, the morphology-aware decoder $\text{Dec}_\phi$ in is realized as a lightweight GNN conditioned on a global temporal summary. We represent $h_t$ as an observation window and an action window, $\{y_{t-L_h+1:t}, w_{t-L_h+1:t}\}$, concatenate them along channels, and encode the resulting length-$L_h$ sequence with a GRU. The last hidden state is projected to a global token $z_t$, and each node is initialized by combining $z_t$ with a morphology embedding derived from the available per-node structural descriptors $g(i)$ (e.g., morphology metadata). When only geometry is available, we use coordinates $\mathbf{x}_i$ as the structural descriptor:

$$\tilde{h}_i^{(0)} = \text{MLP}_{\text{init}}\big([\text{MLP}_{\text{morph}}(g(i)), z_t]\big). \tag{50}$$

In batched graph processing, the token $z_t$ is broadcast to nodes using the graph-to-node assignment index. A shallow GCN stack with residual updates ($h \leftarrow h + \text{GELU}(\text{GCN}(h))$) maps $\tilde{h}_i^{(0)}$ to node-wise predictions $\hat{u}_t(i)$, and we train the observer by the mean-squared reconstruction loss in Eq. (8).

---

**Algorithm 2** History construction under partial observability (deployment interface).

---

**Require:** Current measurement $y_t$, cached graph $\mathcal{G}$, window length $L_h$, buffers $\mathcal{Y}$ and $\mathcal{W}$ storing past measurements and executed actions, preference $p$
 1: Append $y_t$ to $\mathcal{Y}$
 2: $\ell \leftarrow \min(L_h, |\mathcal{Y}|)$ and set $Y_t \leftarrow (y_{t-\ell+1}, \ldots, y_t)$
 3: Construct a padded action window $\bar{W}_t \in \mathbb{R}^{\ell \times C_w}$ by setting $\bar{W}_t[j] \leftarrow w_{t-\ell+j}$ for $j = 1, \ldots, \ell - 1$ and $\bar{W}_t[\ell] \leftarrow 0$
 4: $\hat{u}_t \leftarrow \mathcal{O}_{\phi^\star}(Y_t, \bar{W}_t; \mathcal{G})$
 5: $w_t^{\text{raw}} \leftarrow \text{Planner}(\hat{u}_t, p)$ {rate constraints may depend on the last executed action}
 6: $w_t^\star \leftarrow \text{SafeFilter}(w_t^{\text{raw}}, \hat{u}_t; G_{\theta^\star}, B_{\psi^\star}, \mathcal{G})$
 7: Execute $w_t^\star$ and append it to $\mathcal{W}$
 8: **return** $w_t^\star$

---

## C.9. Closed-Loop Inference Pipeline

Sec. 2.2.5 describes the closed-loop inference flow at a high level, and Algorithm 1 in the main text provides the complete deployment-time specification. Here we keep implementation-facing details (e.g., observer history interface and module inputs/outputs) that complement the main text.

## C.10. Barrier Model and Numerical Details of the Projection

Sec. 2.2.4 defines the barrier $B_\psi$ (Eq. (15)) and the minimal-intervention projection (Eqs. (17)–(19)). Here we add two practical details: the barrier architecture/training signals and numerical safeguards in the projection.

**Barrier parameterization and training signals.** We implement $B_\psi$ as a permutation-invariant functional over node-wise fields on $\mathcal{G}$. Concretely, we concatenate the state with available morphology/geometry descriptors (e.g., $g(i)$ and/or coordinates $\mathbf{x}_i$), embed nodes with an MLP, apply a shallow residual GCN stack, pool by concatenating mean and max graph pooling, and map the pooled feature to a scalar barrier value with an MLP head.

**Label construction.** We derive safety labels by deterministic evaluation of the hard constraints defining $\mathcal{C}(\mathcal{G})$: a state is safe if all monitored variables lie within dataset-calibrated bounds and (when enabled) its energy norm stays below a threshold. In our data interface we store $s \in \{0, 1\}$ and map it to $\{+1, -1\}$ via $2s - 1$, inducing $\mathcal{D}_{\text{safe}}$ and $\mathcal{D}_{\text{unsafe}}$ in (51).

Training uses a margin classification objective with margin $m$:

$$\mathcal{L}_{\text{margin}} = \mathbb{E}_{u \sim \mathcal{D}_{\text{safe}}}[\text{ReLU}(m - B_\psi(u))] + \mathbb{E}_{u \sim \mathcal{D}_{\text{unsafe}}}[\text{ReLU}(m + B_\psi(u))], \tag{51}$$

and optionally a discrete-time CBF regularizer applied on safe transitions,

$$\mathcal{L}_{\text{cbf}} = \mathbb{E}_{(u_t, u_{t+1})}[\text{ReLU}(-(B_\psi(u_{t+1}) - B_\psi(u_t) + \alpha B_\psi(u_t)))], \tag{52}$$

so that $\mathcal{L}_B = \mathcal{L}_{\text{margin}} + \lambda_{\text{cbf}} \mathcal{L}_{\text{cbf}}$. The corresponding hyperparameters (e.g., margin $m$, $\alpha$, and $\lambda_{\text{cbf}}$) are specified by the experimental protocol.

**Projection numerics.** At inference time, we apply a minimal-intervention action correction using the current estimated state $\hat{u}_t$ and the trained simulator $G_{\theta^\star}$. Let $b_{\text{curr}} = B_{\psi^\star}(\hat{u}_t)$ and $b_{\text{next}}(w) = B_{\psi^\star}(G_{\theta^\star}(\hat{u}_t, w; \mathcal{G}))$. The discrete-time barrier condition in Sec. 2.2.4 is

$$b_{\text{next}}(w) - b_{\text{curr}} \geq -\alpha b_{\text{curr}} \quad \Leftrightarrow \quad b_{\text{next}}(w) - (1-\alpha)b_{\text{curr}} \geq 0. \tag{53}$$

Linearizing $b_{\text{next}}(w)$ at the proposed $w_t^{\text{raw}}$ yields the single half-space constraint $\nabla_w b_{\text{next}}(w_t^{\text{raw}})^\top \Delta w \geq c$ where $c = (1-\alpha)b_{\text{curr}} - b_{\text{next}}(w_t^{\text{raw}})$. The resulting projection admits the closed form

$$w_t^\star = w_t^{\text{raw}} + \text{ReLU}(c) \frac{\nabla_w b_{\text{next}}(w_t^{\text{raw}})}{\|\nabla_w b_{\text{next}}(w_t^{\text{raw}})\|_2^2}, \tag{54}$$

where we clamp the denominator from below by $10^{-6}$ for numerical stability and finally clamp $w_t^\star$ to the action bounds. To sum up, we give the implementation steps used in our experiments in Algorithm 3 and Algorithm 4, respectively.

---

**Algorithm 3** Training NEURONCTRL simulator with $K$-step rollout regularization.

---

**Require:** Trajectories $\mathcal{D}$ with graphs $\mathcal{G}$; rollout horizon $K$; weight $\lambda_{\text{roll}}$.
1: Initialize parameters $\theta$ of $\widehat{\mathcal{F}}_\theta$ (Eq. (50)–(11)).
2: **while** not converged **do**
3:      Sample a window $(u_t, u_{t+1}, u_{t+2:t+K+1}, w_{t:t+K-1}, \mathcal{G}) \sim \mathcal{D}$.
4:      One-step prediction $\hat{u}_{t+1} \leftarrow u_t + \Delta t \widehat{\mathcal{F}}_\theta(u_t, w_t; \mathcal{G})$ {Eq. (11)}
5:      $\mathcal{L}_{\text{1step}} \leftarrow \|\hat{u}_{t+1} - u_{t+1}\|^2$ {}
6:      Initialize rollout state $\tilde{u} \leftarrow \hat{u}_{t+1}$ and $\mathcal{L}_{\text{roll}} \leftarrow 0$.
7:      **for** $k = 0$ to $K - 1$ **do**
8:          $\tilde{u} \leftarrow \tilde{u} + \Delta t \widehat{\mathcal{F}}_\theta(\tilde{u}, w_{t+k}; \mathcal{G})$
9:          $\mathcal{L}_{\text{roll}} \leftarrow \mathcal{L}_{\text{roll}} + \|\tilde{u} - u_{t+2+k}\|^2$
10:      **end for**
11:      Update $\theta$ by minimizing $\mathcal{L}_{\text{1step}} + \lambda_{\text{roll}} \mathcal{L}_{\text{roll}}/K$.
12: **end while**

---

**Algorithm 4** One-step control using differentiable MPC over NEURONCTRL with barrier filtering.

---

**Require:** Estimated state $\hat{u}_t$ (observer optional); preference $p$; warm-start plan $W^{\text{init}}$; horizon $H$; inner steps $S$.
1: Initialize action sequence $W \leftarrow W^{\text{init}}$ and enable gradients on $W$.
2: **for** $s = 1$ to $S$ **do**
3:      Roll out $\hat{u}_{t+1:t+H}$ using $\widehat{\mathcal{G}}_\theta$ with actions $W$ {}
4:      Compute multi-objective cost $J(W; p)$ {}
5:      Update $W$ with one optimizer step using $\nabla_W J$.
6:      Project $W$ to box constraints $[w_{\min}, w_{\max}]$.
7: **end for**
8: $w_t^{\text{raw}} \leftarrow W[0]$.
9: $w_t^\star \leftarrow$ barrier filter using Eq. (54).
10: **return** $w_t^\star$.

---

## C.11. Theoretical Details

This section complements the implementation description of NEURONCTRL with proposition-proof style theoretical details, focusing on (i) operator consistency under irregular discretizations, (ii) rollout stability and error propagation for residual-Euler neural operators, and (iii) guarantees for the differentiable safety projection used at test time.

### C.11.1. DISCRETE FIELD GEOMETRY AND PERMUTATION EQUIVARIANCE

We work with node-wise fields $u \in \mathbb{R}^{N \times C_u}$ defined on an irregular point set $\{\mathbf{x}_i\}_{i=1}^N \subset \Omega$. Throughout, we use the discrete inner product and norm that approximate $\mathcal{L}^2(\Omega)$.

**Definition C.2** (Discrete inner product and norm). For two node-wise fields $a, b \in \mathbb{R}^{N \times d}$, define $\langle a, b \rangle_N \triangleq \frac{1}{N} \sum_{i=1}^N a(i)^\top b(i)$ and $\|a\|_N \triangleq \sqrt{\langle a, a \rangle_N}$.

Equivalently, the (unnormalized) Frobenius norm $\|a\|_F \triangleq \left( \sum_{i=1}^N \|a(i)\|_2^2 \right)^{1/2}$ satisfies $\|a\|_F = \sqrt{N} \|a\|_N$.

The heterogeneous message passing block used by NEURONCTRL (Eq. (39)–(40) in the main appendix text) must be *permutation equivariant* to node ordering, a necessary property for discretization-robust operator learning.

**Proposition C.3** (Permutation equivariance of heterogeneous message passing). *Let $\Pi$ be any permutation of $\{1, \ldots, N\}$ and let $P \in \{0, 1\}^{N \times N}$ be the corresponding permutation matrix. Consider a heterogeneous message passing layer with shared parameters across nodes and relation-wise sum aggregation:*

$$m_{i,r} = \sum_{j \in \mathcal{N}_i^r} \omega_{ij}\, \phi_r(h_i, h_j, e_{ij}), \qquad h_i^+ = \Psi(h_i, \{m_{i,r}\}_r), \tag{55}$$

*where $\Psi$ denotes the (shared) post-aggregation transform including residual/LN/nonlinearity. If the graph is permuted consistently (node features, edge indices, and edge attributes permuted by $\Pi$), then the layer output is permuted in the same way: $h_{\Pi(i)}^+$(permuted graph) $= h_i^+$(original graph) for all $i$. Equivalently, in matrix form, $H_{perm}^+ = PH^+$.*

*Proof.* Under a consistent permutation of the graph, the neighbor set of node $\Pi(i)$ in the permuted graph corresponds bijectively to $\{\Pi(j) : j \in \mathcal{N}_i^r\}$ in the original graph, with the same edge attributes up to permutation. Because $\phi_r$ and $\Psi$ are shared across nodes and sum aggregation is permutation invariant within each neighborhood, the aggregated messages satisfy $m_{\Pi(i),r}^{\text{perm}} = m_{i,r}$ for all $r$. Applying the same shared post-aggregation transform $\Psi$ yields $h_{\Pi(i)}^{+,\text{perm}} = h_i^+$, which is exactly permutation equivariance. $\square$

### C.11.2. GRAPH NEURAL OPERATORS AS CONSISTENT QUADRATURES OF KERNEL INTEGRAL OPERATORS

A core design goal of NEURONCTRL is to approximate continuous kernel integral operators (Eq. (37)) using graph-based quadrature (Eq. (38)). To make this precise, we state the assumptions under which the graph summation is a consistent estimator of the underlying integral operator.

**Assumption C.4** (Kernel regularity). For each relation $r$, the kernel $\kappa_r(\mathbf{x}, \mathbf{y})$ is measurable and bounded: $\|\kappa_r(\mathbf{x}, \mathbf{y})\|_{\text{op}} \leq K_r$ for all $\mathbf{x}, \mathbf{y} \in \Omega$.

**Assumption C.5** (Sampling measure and bounded features). Let $\mu$ be a probability measure on $\Omega$. Assume node locations $\{\mathbf{x}_j\}_{j=1}^N$ are sampled i.i.d. from $\mu$, and the feature field $h : \Omega \to \mathbb{R}^d$ is bounded: $\|h(\mathbf{y})\|_2 \leq H$ for $\mu$-a.e. $\mathbf{y}$.

The i.i.d. sampling assumption is used to obtain a simple mean-square bound; deterministic discretizations (e.g., fixed meshes or kNN/radius graphs) can be viewed as low-discrepancy point sets whose empirical measures approximate $\mu$, leading to analogous quadrature consistency statements under standard regularity assumptions.

For a fixed query location $\mathbf{x}$, define the continuous kernel operator

$$(\mathcal{K}_r h)(\mathbf{x}) \triangleq \int_\Omega \kappa_r(\mathbf{x}, \mathbf{y})\, h(\mathbf{y})\, d\mu(\mathbf{y}), \tag{56}$$

and its Monte-Carlo discretization

$$(\mathcal{K}_r^N h)(\mathbf{x}) \triangleq \frac{1}{N} \sum_{j=1}^N \kappa_r(\mathbf{x}, \mathbf{x}_j)\, h(\mathbf{x}_j). \tag{57}$$

Equation (57) captures the continuum-limit interpretation of Eq. (38): restricting to neighborhoods $\mathcal{N}_i^r$ corresponds to compactly supported or masked kernels (e.g., multiplying by $\mathbb{I}[\|\mathbf{x} - \mathbf{y}\| \leq \rho]$), while the normalization factor can be absorbed into the learnable kernel parameterization and the post-aggregation normalization in Eq. (40).

**Proposition C.6** (Mean-square consistency of kernel quadrature). *Under Assumptions C.4–C.5, for any fixed $\mathbf{x} \in \Omega$,*

$$\mathbb{E}\Big[\|(\mathcal{K}_r^N h)(\mathbf{x}) - (\mathcal{K}_r h)(\mathbf{x})\|_2^2\Big] \leq \frac{K_r^2 H^2}{N}. \tag{58}$$

*Proof.* Let $Z_j \triangleq \kappa_r(\mathbf{x}, \mathbf{x}_j) h(\mathbf{x}_j) \in \mathbb{R}^d$. Then $(\mathcal{K}_r^N h)(\mathbf{x}) = \frac{1}{N} \sum_{j=1}^N Z_j$ and $(\mathcal{K}_r h)(\mathbf{x}) = \mathbb{E}[Z_1]$. By independence and the standard variance identity,

$$\mathbb{E}\Big\| \frac{1}{N} \sum_{j=1}^N (Z_j - \mathbb{E}Z_j) \Big\|_2^2 = \frac{1}{N} \mathbb{E}\|Z_1 - \mathbb{E}Z_1\|_2^2 \leq \frac{1}{N} \mathbb{E}\|Z_1\|_2^2. \tag{59}$$

Using $\|Z_1\|_2 \leq \|\kappa_r(\mathbf{x}, \mathbf{x}_1)\|_{\mathrm{op}} \|h(\mathbf{x}_1)\|_2 \leq K_r H$ yields $\mathbb{E}\|Z_1\|_2^2 \leq K_r^2 H^2$, which proves Eq. (58). $\square$

Eq. (58) formally supports the discretization-robust perspective used by NEURONCTRL: if the learned message function approximates a bounded kernel, the discretized operator converges to its continuous counterpart as node density increases. A convenient corollary bounds the discrepancy between two discretizations (e.g., different mesh resolutions).

**Proposition C.7** (Cross-resolution discrepancy bound). *Let $\mathcal{K}_r^N$ and $\mathcal{K}_r^M$ be defined as in Eq. (57) using two independent i.i.d. samples of sizes $N$ and $M$ from the same measure $\mu$. Under Assumptions C.4–C.5, for any fixed $\mathbf{x} \in \Omega$,*

$$\mathbb{E}\Big[\|(\mathcal{K}_r^N h)(\mathbf{x}) - (\mathcal{K}_r^M h)(\mathbf{x})\|_2^2\Big] \leq 2 K_r^2 H^2 \Big(\frac{1}{N} + \frac{1}{M}\Big). \tag{60}$$

*Proof.* By triangle inequality and $(a + b)^2 \leq 2a^2 + 2b^2$,

$$\|\mathcal{K}_r^N h - \mathcal{K}_r^M h\|_2^2 \leq 2\|\mathcal{K}_r^N h - \mathcal{K}_r h\|_2^2 + 2\|\mathcal{K}_r^M h - \mathcal{K}_r h\|_2^2. \tag{61}$$

Taking expectations and applying Proposition C.6 to each term yields the claim. $\square$

### C.11.3. RESIDUAL-EULER NEURAL OPERATORS: LIPSCHITZ CONSTANT AND ROLLOUT ERROR PROPAGATION

NEURONCTRL parameterizes a one-step operator in residual-Euler form: $\widehat{\mathcal{G}}_\theta(u, w) = u + \Delta t\, \widehat{\mathcal{F}}_\theta(u, w)$ (Eq. (11)–(11)). This structure provides an explicit handle on stability through Lipschitz constants.

**Lemma C.8** (Lipschitz constant of residual-Euler operators). *Fix $w$. If $\widehat{\mathcal{F}}_\theta(\cdot, w)$ is $L_{\widehat{\mathcal{F}}}$-Lipschitz with respect to $\|\cdot\|_N$, i.e., $\|\widehat{\mathcal{F}}_\theta(u, w) - \widehat{\mathcal{F}}_\theta(v, w)\|_N \leq L_{\widehat{\mathcal{F}}}\|u - v\|_N$, then $\widehat{\mathcal{G}}_\theta(\cdot, w)$ is $(1 + \Delta t\, L_{\widehat{\mathcal{F}}})$-Lipschitz:*

$$\|\widehat{\mathcal{G}}_\theta(u, w) - \widehat{\mathcal{G}}_\theta(v, w)\|_N \leq (1 + \Delta t\, L_{\widehat{\mathcal{F}}})\|u - v\|_N. \tag{62}$$

*Proof.*

$$\|\widehat{\mathcal{G}}_\theta(u, w) - \widehat{\mathcal{G}}_\theta(v, w)\|_N = \|(u - v) + \Delta t(\widehat{\mathcal{F}}_\theta(u, w) - \widehat{\mathcal{F}}_\theta(v, w))\|_N \leq \|u - v\|_N + \Delta t L_{\widehat{\mathcal{F}}}\|u - v\|_N. \tag{63}$$

$\square$

We next formalize the compounding-error phenomenon that motivates $K$-step rollout regularization.

**Proposition C.9** (Rollout error propagation bound). *Let the true one-step map be $u_{t+1} = \mathcal{G}(u_t, w_t)$ and the learned one-step map be $\hat{u}_{t+1} = \widehat{\mathcal{G}}_\theta(\hat{u}_t, w_t)$. Assume $\widehat{\mathcal{G}}_\theta(\cdot, w)$ is $L_{\widehat{\mathcal{G}}}$-Lipschitz for all $w$. Define the* on-trajectory one-step model error

$$\varepsilon_t \triangleq \|\widehat{\mathcal{G}}_\theta(u_t, w_t) - \mathcal{G}(u_t, w_t)\|_N. \tag{64}$$

*Then, for any horizon $K \geq 1$,*

$$\|\hat{u}_{t+K} - u_{t+K}\|_N \leq L_{\widehat{\mathcal{G}}}^K \|\hat{u}_t - u_t\|_N + \sum_{k=0}^{K-1} L_{\widehat{\mathcal{G}}}^{K-1-k} \varepsilon_{t+k}. \tag{65}$$

*Proof.* Write the telescoping decomposition

$$\hat{u}_{t+k+1} - u_{t+k+1} = \widehat{\mathcal{G}}_\theta(\hat{u}_{t+k}, w_{t+k}) - \mathcal{G}(u_{t+k}, w_{t+k}) = A_k + B_k, \tag{66}$$

where $A_k = \widehat{\mathcal{G}}_\theta(\hat{u}_{t+k}, w_{t+k}) - \widehat{\mathcal{G}}_\theta(u_{t+k}, w_{t+k})$ and $B_k = \widehat{\mathcal{G}}_\theta(u_{t+k}, w_{t+k}) - \mathcal{G}(u_{t+k}, w_{t+k})$. Taking norms and using Lipschitzness gives $\|\hat{u}_{t+k+1} - u_{t+k+1}\|_N \le L_{\widehat{\mathcal{G}}}\|\hat{u}_{t+k} - u_{t+k}\|_N + \varepsilon_{t+k}$. Unrolling this recursion for $k = 0, \ldots, K - 1$ yields Eq. (65). □

Combining Lemma C.8 and Proposition C.9 shows explicitly how residual-Euler design and engineering controls (normalization, clipping, spectral constraints) act by reducing $L_{\widehat{\mathcal{F}}}$ and thus attenuating geometric amplification.

A more refined (and practically relevant) stability condition can be stated under a dissipativity-style assumption (common in continuous-time neuronal dynamics with leakage terms).

**Assumption C.10** (One-sided Lipschitz / dissipativity). Fix $w$. Assume $\widehat{\mathcal{F}}_\theta(\cdot, w)$ satisfies a one-sided Lipschitz condition: there exists $m > 0$ such that $\langle u - v, \widehat{\mathcal{F}}_\theta(u, w) - \widehat{\mathcal{F}}_\theta(v, w)\rangle_N \le -m\|u - v\|_N^2$ for all $u, v$. Also assume $\widehat{\mathcal{F}}_\theta(\cdot, w)$ is $L_{\widehat{\mathcal{F}}}$-Lipschitz.

**Proposition C.11** (Non-expansiveness of the Euler step under dissipativity). *Under Assumption C.10, the residual-Euler map $\widehat{\mathcal{G}}_\theta(u, w) = u + \Delta t\, \widehat{\mathcal{F}}_\theta(u, w)$ is non-expansive in $\|\cdot\|_N$ whenever $\Delta t \le 2m/L_{\widehat{\mathcal{F}}}^2$, i.e., $\|\widehat{\mathcal{G}}_\theta(u, w) - \widehat{\mathcal{G}}_\theta(v, w)\|_N \le \|u - v\|_N$ for all $u, v$.*

*Proof.* Let $d = u - v$ and $g = \widehat{\mathcal{F}}_\theta(u, w) - \widehat{\mathcal{F}}_\theta(v, w)$. Then

$$\|\widehat{\mathcal{G}}_\theta(u, w) - \widehat{\mathcal{G}}_\theta(v, w)\|_N^2 = \|d + \Delta t\, g\|_N^2 = \|d\|_N^2 + 2\Delta t\langle d, g\rangle_N + \Delta t^2\|g\|_N^2. \tag{67}$$

By Assumption C.10, $\langle d, g\rangle_N \le -m\|d\|_N^2$, and by Lipschitzness, $\|g\|_N \le L_{\widehat{\mathcal{F}}}\|d\|_N$. Thus

$$\|d + \Delta t\, g\|_N^2 \le \left(1 - 2m\Delta t + L_{\widehat{\mathcal{F}}}^2\Delta t^2\right)\|d\|_N^2. \tag{68}$$

If $\Delta t \le 2m/L_{\widehat{\mathcal{F}}}^2$, then $-2m\Delta t + L_{\widehat{\mathcal{F}}}^2\Delta t^2 \le 0$, hence $\|d + \Delta t\, g\|_N^2 \le \|d\|_N^2$, proving non-expansiveness. □

Proposition C.11 is not merely abstract: it explains why (i) choosing sufficiently small $\Delta t$, and (ii) regularizing Jacobians / effective Lipschitz constants of $\widehat{\mathcal{F}}_\theta$ (via weight decay, normalization, or spectral constraints) materially improves long-horizon rollout stability.

### C.11.4. ADJOINT GRADIENTS FOR DIFFERENTIABLE MPC OVER NEURONCTRL

We provide an explicit ICML-style derivation for the gradient used in Algorithm 4. Let $\hat{u}_{t+1} = \widehat{\mathcal{G}}_\theta(\hat{u}_t, w_t)$ and define the finite-horizon objective

$$J(W) = \sum_{h=0}^{H-1} \ell(\hat{u}_{t+h}, w_{t+h}) + \ell_T(\hat{u}_{t+H}), \qquad W = [w_t, \ldots, w_{t+H-1}]. \tag{69}$$

**Proposition C.12** (Adjoint recursion for $\nabla_W J$). *Assume $\widehat{\mathcal{G}}_\theta$ and $\ell, \ell_T$ are differentiable. Define the adjoint variables $\lambda_{t+H} \triangleq \nabla_{\hat{u}_{t+H}} \ell_T(\hat{u}_{t+H})$ and for $h = H - 1, \ldots, 0$,*

$$\lambda_{t+h} = \nabla_{\hat{u}_{t+h}} \ell(\hat{u}_{t+h}, w_{t+h}) + \left(\nabla_{\hat{u}_{t+h}} \widehat{\mathcal{G}}_\theta(\hat{u}_{t+h}, w_{t+h})\right)^\top \lambda_{t+h+1}. \tag{70}$$

*Then the gradient with respect to each control is*

$$\nabla_{w_{t+h}} J = \nabla_{w_{t+h}} \ell(\hat{u}_{t+h}, w_{t+h}) + \left(\nabla_{w_{t+h}} \widehat{\mathcal{G}}_\theta(\hat{u}_{t+h}, w_{t+h})\right)^\top \lambda_{t+h+1}. \tag{71}$$

*Proof.* Apply the chain rule to $J$ with respect to $w_{t+h}$. The term $\ell(\hat{u}_{t+h}, w_{t+h})$ contributes $\nabla_{w_{t+h}} \ell$ directly. All future losses depend on $w_{t+h}$ only through the state recursion $\hat{u}_{t+h+1} = \widehat{\mathcal{G}}_\theta(\hat{u}_{t+h}, w_{t+h})$. Introduce $\lambda_{t+h+1} = \nabla_{\hat{u}_{t+h+1}} J$ and expand

$$\nabla_{w_{t+h}} J = \nabla_{w_{t+h}} \ell(\hat{u}_{t+h}, w_{t+h}) + \left(\nabla_{w_{t+h}} \widehat{\mathcal{G}}_\theta(\hat{u}_{t+h}, w_{t+h})\right)^\top \lambda_{t+h+1}, \tag{72}$$

which is Eq. (71). Similarly, $\lambda_{t+h} = \nabla_{\hat{u}_{t+h}} J$ decomposes into the immediate gradient $\nabla_{\hat{u}_{t+h}} \ell$ plus the backpropagated term through $\hat{u}_{t+h+1}$, yielding Eq. (70). □

C.11.5. Barrier Filtering as a Closed-Form Projection and a Robustness Margin

At test time, we apply a minimal correction to candidate controls based on a differentiable barrier function $B_\psi$ and the learned one-step dynamics $\widehat{\mathcal{G}}_\theta$. Let $f(w) \triangleq B_\psi(\widehat{\mathcal{G}}_\theta(\hat{u}_t, w))$ and consider the (linearized) constraint

$$f(w_t^{\text{raw}} + \Delta w) \approx f(w_t^{\text{raw}}) + a^\top \Delta w \geq \gamma, \tag{73}$$

where $a = \nabla_w f(w_t^{\text{raw}})$ and $\gamma$ is the required barrier threshold from the discrete-time condition in Eq. (17) (i.e., $\gamma = (1 - \alpha) B_\psi(\hat{u}_t)$ in the main text).

**Proposition C.13** (Closed-form solution of the single-constraint projection). *Consider the quadratic program*

$$\min_{\Delta w \in \mathbb{R}^{C_w}} \frac{1}{2} \|\Delta w\|_2^2 \quad s.t. \quad a^\top \Delta w \geq c, \tag{74}$$

*where $a \neq 0$. Its unique solution is*

$$\Delta w^\star = \begin{cases} 0, & c \leq 0, \\ \frac{c}{\|a\|_2^2} a, & c > 0. \end{cases} \tag{75}$$

*Proof.* The Lagrangian is $\mathcal{L}(\Delta w, \lambda) = \frac{1}{2}\|\Delta w\|_2^2 - \lambda(a^\top \Delta w - c)$ with $\lambda \geq 0$. KKT stationarity gives $\nabla_{\Delta w}\mathcal{L} = \Delta w - \lambda a = 0$, hence $\Delta w = \lambda a$. Complementary slackness yields $\lambda(a^\top \Delta w - c) = 0$. If $c \leq 0$, then $\Delta w = 0$ is feasible and optimal since the objective is nonnegative and minimized at 0. If $c > 0$, feasibility requires $a^\top \Delta w = c$, i.e., $\lambda\|a\|_2^2 = c$, so $\lambda = c/\|a\|_2^2$ and $\Delta w^\star = (c/\|a\|_2^2)a$. $\square$

The above result guarantees feasibility for the *linearized* constraint. To relate this to the original nonlinear constraint $f(w_t^{\text{raw}} + \Delta w) \geq \gamma$, we quantify the Taylor remainder.

**Lemma C.14** (Second-order remainder bound). *Assume $f$ is twice differentiable and its Hessian is uniformly bounded in operator norm: $\|\nabla^2 f(w)\|_{\text{op}} \leq M$ for all $w$ in a neighborhood of $w_t^{\text{raw}}$. Then for any $\Delta w$,*

$$f(w_t^{\text{raw}} + \Delta w) \geq f(w_t^{\text{raw}}) + a^\top \Delta w - \frac{M}{2}\|\Delta w\|_2^2, \quad a = \nabla f(w_t^{\text{raw}}). \tag{76}$$

*Proof.* By the second-order Taylor theorem with integral remainder, $f(w + \Delta w) = f(w) + a^\top \Delta w + \frac{1}{2}\Delta w^\top \nabla^2 f(w + \xi \Delta w)\Delta w$ for some $\xi \in (0, 1)$. Using $\|\nabla^2 f(\cdot)\|_{\text{op}} \leq M$ gives $\Delta w^\top \nabla^2 f(\cdot)\Delta w \geq -M\|\Delta w\|_2^2$, which yields the inequality. $\square$

Lemma C.14 yields a simple *robustification* principle for the linearized projection: if we enforce a margin that upper-bounds the worst-case negative curvature, then we guarantee the *true* constraint.

**Proposition C.15** (Sufficient condition for satisfying the true barrier constraint). *Under Lemma C.14, suppose $\Delta w$ satisfies*

$$f(w_t^{\text{raw}}) + a^\top \Delta w - \frac{M}{2}\|\Delta w\|_2^2 \geq \gamma. \tag{77}$$

*Then it follows that $f(w_t^{\text{raw}} + \Delta w) \geq \gamma$.*

*Proof.* Immediate from Lemma C.14 by lower bounding $f(w_t^{\text{raw}} + \Delta w)$ with the left-hand side. $\square$

In our implementation, we enforce a trust region $\|\Delta w\|_2 \leq r$ and tighten the linearized constraint by a curvature-aware margin $\frac{M}{2}r^2$. The curvature bound $M$ is estimated online: we compute the exact Hessian when the action dimension is small, and otherwise approximate $\|\nabla^2 f\|_{\text{op}}$ via Hessian–vector products and power iteration, scaling the estimate by a conservative safety factor. The resulting minimal-intervention update is then projected to the trust region and clamped to the action bounds.

**Proposition C.16** (Margin condition under bounded dynamics mismatch)**.** *Assume the true one-step dynamics $\mathcal{G}$ and learned one-step dynamics $\widehat{\mathcal{G}}_\theta$ satisfy a uniform mismatch bound*

$$\|\mathcal{G}(u,w) - \widehat{\mathcal{G}}_\theta(u,w)\|_N \leq \delta \tag{78}$$

*for all state–action pairs $(u,w)$ in the operating region. Suppose the barrier functional $B_\psi$ is $L_B$-Lipschitz in its state argument, i.e.,*

$$|B_\psi(u) - B_\psi(v)| \leq L_B \|u - v\|_N \quad \forall u, v. \tag{79}$$

*Then any action $w$ satisfying the tightened model-based constraint*

$$B_\psi\left(\widehat{\mathcal{G}}_\theta(u,w)\right) \geq \gamma + L_B\,\delta \tag{80}$$

*also satisfies the true constraint $B_\psi(\mathcal{G}(u,w)) \geq \gamma$.*

*Proof.* By Lipschitz continuity,

$$B_\psi(\mathcal{G}(u,w)) \geq B_\psi\left(\widehat{\mathcal{G}}_\theta(u,w)\right) - L_B\|\mathcal{G}(u,w) - \widehat{\mathcal{G}}_\theta(u,w)\|_N \geq \gamma. \tag{81}$$

$\square$

**Proposition C.17** (Margin condition under bounded state-estimation error)**.** *Assume the state estimate $\hat{u}$ satisfies $\|\hat{u} - u\|_N \leq \varepsilon_{\mathrm{obs}}$. Suppose the learned one-step dynamics $\widehat{\mathcal{G}}_\theta(\cdot, w)$ is $L_G$-Lipschitz in its state argument and the barrier functional $B_\psi$ is $L_B$-Lipschitz, i.e.,*

$$\|\widehat{\mathcal{G}}_\theta(u,w) - \widehat{\mathcal{G}}_\theta(v,w)\|_N \leq L_G\|u - v\|_N, \qquad |B_\psi(u) - B_\psi(v)| \leq L_B\|u - v\|_N. \tag{82}$$

*Then any action $w$ satisfying the tightened estimated-state constraint*

$$B_\psi\left(\widehat{\mathcal{G}}_\theta(\hat{u}, w)\right) \geq (1-\alpha)\,B_\psi(\hat{u}) + L_B\big(L_G + (1-\alpha)\big)\,\varepsilon_{\mathrm{obs}} \tag{83}$$

*also satisfies the nominal model-based condition for the underlying state, $B_\psi(\widehat{\mathcal{G}}_\theta(u,w)) \geq (1-\alpha)\,B_\psi(u)$.*

*Proof.* By Lipschitz continuity of $B_\psi$ and $\widehat{\mathcal{G}}_\theta$,

$$B_\psi(\widehat{\mathcal{G}}_\theta(u,w)) \geq B_\psi(\widehat{\mathcal{G}}_\theta(\hat{u},w)) - L_B\,\|\widehat{\mathcal{G}}_\theta(u,w) - \widehat{\mathcal{G}}_\theta(\hat{u},w)\|_N \geq B_\psi(\widehat{\mathcal{G}}_\theta(\hat{u},w)) - L_B L_G\,\varepsilon_{\mathrm{obs}}. \tag{84}$$

Similarly, $B_\psi(\hat{u}) \leq B_\psi(u) + L_B\varepsilon_{\mathrm{obs}}$ implies $(1-\alpha)B_\psi(\hat{u}) \leq (1-\alpha)B_\psi(u) + (1-\alpha)L_B\varepsilon_{\mathrm{obs}}$. Combining these inequalities with the tightened constraint yields the claim. $\square$

**Remarks on engineering alignment.** The theoretical quantities above directly map to implementation choices. Reducing effective Lipschitz constants (Lemma C.8) through normalization and controlled step sizes improves rollout stability (Proposition C.9). In the safety filter, curvature-aware margins (Lemma C.14) combined with trust regions provide a principled and efficient route from first-order linearization to reliable satisfaction of the nonlinear barrier condition. To operationalize the remaining robustness quantities in Propositions C.16 and C.17, we use conservative empirical upper bounds: we estimate $\delta$ as a high-quantile (e.g., 99%) of one-step prediction error $\|\mathcal{G}(u,w) - \widehat{\mathcal{G}}_\theta(u,w)\|_N$ on held-out trajectories, and estimate $\varepsilon_{\mathrm{obs}}$ analogously from observer reconstruction error $\|\hat{u} - u\|_N$. Local Lipschitz constants $L_G$ and $L_B$ can be upper-bounded by Jacobian operator norms over the operating region; in practice, they can be estimated with a small number of power iterations using Jacobian–vector products (in the same spirit as the online estimation of the curvature bound $M$), and then used to select margin-tightening parameters in a principled way.

# D. Background & Related Work

NEURONCTRL addresses closed-loop control when the plant state is fundamentally *distributed* (field-valued), evolves on *irregular and instance-dependent* geometries (branched neuronal morphologies with hybrid 1D/3D intra–extra-cellular domains), is only *partially observed* through sparse sensing, and must satisfy *explicit safety constraints* that are enforceable online. This section first establishes the mathematical viewpoint that connects mechanistic neuronal simulation to distributed-parameter control, then positions learning-based components (operator surrogates, observers, and generative controllers) relative to the closest alternatives. Throughout we emphasize a key axis of comparison that is often implicit in prior work: whether a method is formulated as an *operator on function spaces* (hence potentially discretization- and morphology-aware), or as a controller/estimator over a fixed-dimensional surrogate that does not explicitly encode the underlying geometry.

## D.1. Background

### D.1.1. NEURONAL DYNAMICS AS PDEs ON NETWORKS AND HYBRID DOMAINS

Conductance-based neuronal models couple cable theory with nonlinear ionic kinetics (Hodgkin & Huxley, 1952; Rall, 2009). The membrane voltage on a cable obeys:

$$C_m \frac{\partial V(\ell, t)}{\partial t} = \frac{1}{r_a} \frac{\partial^2 V(\ell, t)}{\partial \ell^2} - I_{\text{ion}}\big(V(\ell, t), \mathbf{w}(\ell, t)\big) + I_{\text{app}}(\ell, t). \tag{85}$$

Real neurons are branched trees, naturally viewed as PDEs on metric graphs with Kirchhoff-type junction conditions (Hines & Carnevale, 1997; Mugnolo, 2014). Neuromodulation also involves reaction–diffusion dynamics:

$$\frac{\partial c_i(\mathbf{r}, t)}{\partial t} = \nabla \cdot \big(D_i(\mathbf{r}) \nabla c_i(\mathbf{r}, t)\big) + f_i\big(\mathbf{c}(\mathbf{r}, t), V(\mathbf{r}, t)\big). \tag{86}$$

Hybrid simulators like NEURON/RxD (McDougal et al., 2013) couple these dynamics on heterogeneous supports (1D cables, 3D volumes), yielding a distributed-parameter plant with instance-dependent geometry.

### D.1.2. DISTRIBUTED-PARAMETER SYSTEMS AND CLOSED-LOOP OBJECTIVES

A unifying abstraction for PDE-governed plants is an evolution equation on a function space (Curtain & Zwart, 2012; Bensoussan et al., 2007):

$$\dot{x}(t) = \mathcal{A}x(t) + \mathcal{N}\big(x(t)\big) + \mathcal{B}u(t), \qquad y(t) = \mathcal{C}x(t), \tag{87}$$

with $\mathcal{A}$ an unbounded operator, $\mathcal{N}$ nonlinear terms, and $\mathcal{B}, \mathcal{C}$ actuation/observation operators. Closed-loop objectives can be posed as PDE-constrained optimal control (Hinze & Vierling, 2009):

$$\min_{u(\cdot)} \int_0^T \ell\big(x(t), u(t)\big) \, dt + \Phi\big(x(T)\big) \ \text{ s.t. } \ \dot{x}(t) = \mathcal{A}x(t) + \mathcal{N}(x(t)) + \mathcal{B}u(t). \tag{88}$$

Classical methods require known PDE coefficients and geometry at design time, which is impractical when geometry varies across instances. Partial observability shifts attention to output feedback laws $u(t) = \pi(y_{0:t})$ via observer-controller separation (Tucsnak & Weiss, 2009). NEURONCTRL uses a data-driven, morphology-aware observer that generalizes across variable graphs and supports online safety enforcement.

## D.2. Related Work

**Positioning rubric.** We compare approaches along four axes: (i) variable geometries/discretizations, (ii) output feedback under sparse observations, (iii) deployable online safety enforcement, and (iv) real-time feasibility. When applicable, we implement representative baselines and report performance and safety in Table 2.

### D.2.1. CLASSICAL CONTROL OF PDE SYSTEMS AND WHY IT IS NOT DIRECTLY DEPLOYABLE HERE

The control of distributed-parameter systems is a mature field with deep results on stabilization, controllability, and optimal feedback design (Curtain & Zwart, 2012; Bensoussan et al., 2007; Tucsnak & Weiss, 2009; Lasiecka & Triggiani, 2000). For linear PDEs, LQR-type feedback can be derived via operator Riccati equations, and boundary/distributed feedback laws can yield stability and performance guarantees under correct modeling assumptions (Bensoussan et al., 2007; Curtain & Zwart,

2012). For certain classes of 1D PDEs, backstepping provides constructive boundary controllers and observers with explicit stability certificates (Krstic & Smyshlyaev, 2008; Bastin & Coron, 2016). These methods differ from NEURONCTRL in a crucial way: they typically assume a *known* PDE operator on a *fixed* domain, and the resulting feedback depends explicitly on that operator (e.g., Riccati/backstepping kernel solutions). When the geometry itself varies across instances (different morphologies and discretizations) and the actuation/sensing map is sparse and device-specific, re-deriving such feedback laws per morphology is nontrivial and often computationally prohibitive. Moreover, many classical PDE feedback designs are derived for boundary actuation or structured distributed inputs, whereas neuromodulation interfaces impose constraints that can be spatially localized, saturating, and safety-critical in ways that are difficult to encode in closed-form feedback.

PDE-constrained model predictive control (MPC) provides a principled closed-loop alternative by solving (88) over a moving horizon, potentially with hard constraints (Rawlings & Mayne, 2009; Grne & Pannek, 2013). In chemical processes, thermal systems, and fluid flows, PDE-MPC has been effective when high-fidelity models and efficient solvers are available. However, its online computational load typically scales with repeated forward/adjoint PDE solves. This is a poor match to high-frequency stimulation regimes and to hybrid neuron models where the state includes stiff ionic kinetics and reaction–diffusion. Reduced-order MPC partially mitigates this by controlling a low-dimensional surrogate, but it introduces additional requirements: a stable ROM across operating conditions and a reliable estimator compatible with sparse observations (Antoulas, 2005; Benner et al., 2015; Hinze & Vierling, 2009). These assumptions are strained in neuronal settings where morphology varies across instances and where the relevant safety constraints can be *spatially local* (e.g., concentration hot-spots), making naive reduction risky. Recent work has also examined stability conditions for event-based control with neuronal dynamics (Eilers et al., 2025), showing how neural firing constraints affect closed-loop stability; NEURONCTRL addresses a complementary setting where the control objective is field-level regulation with hard safety constraints under sparse sensing. For empirical context, we include sensor-limited classical feedback baselines (receding-horizon MPC) under the same observation interface and action bounds as other methods (Section A.1) and report their closed-loop performance, safety, and real-time feasibility in Table 2.

### D.2.2. REDUCED-ORDER AND DATA-DRIVEN MODELS FOR PDE CONTROL

Model reduction has long been the bridge between PDE dynamics and real-time control. Projection-based ROMs (e.g., POD/Galerkin) compress high-dimensional fields into a low-dimensional latent subspace learned from snapshots (Holmes et al., 2012; Rowley, 2005; Benner et al., 2015). Balanced truncation and related techniques provide system-theoretic reductions with stability and input–output error bounds for linear systems (Antoulas, 2005; Curtain & Zwart, 2012). In PDE control, ROMs are often paired with MPC or LQR to achieve tractable feedback. The limitation in our regime is not merely "high dimension", but *instance dependence*: when the morphology, mesh, or graph connectivity changes, a ROM trained on one discretization does not automatically transfer, and constructing a shared basis across variable trees/hybrid domains is itself a nontrivial alignment problem.

A complementary line of work learns governing equations or reduced dynamics directly from data, including sparse regression of nonlinear dynamics (SINDy) (Brunton et al., 2016) and Koopman-operator-based linear representations (Mezić, 2005; Korda & Mezić, 2018). These approaches can be highly interpretable and effective when the underlying dynamics admit a compact representation in a chosen library/observable space. In contrast, our target plant couples stiff ionic kinetics with diffusion/transport on irregular domains, and the primary barrier is not only identifying a low-dimensional model but ensuring that the learned representation remains valid across discretizations and supports online inference under sparse sensing.

Machine learning has increasingly been applied to PDE control problems (notably in fluid mechanics), often using deep reinforcement learning or imitation learning to discover control strategies (Duriez et al., 2017; Rabault et al., 2019; Brunton et al., 2020). These studies demonstrate that neural policies can handle nonlinear PDE dynamics, but they typically operate on fixed geometries with consistent discretizations and rely on either dense state measurements or carefully engineered low-dimensional observations. Saccani et al. (Saccani et al., 2024) train a network of neural closed-loop maps with provable stability guarantees for distributed control; their focus on finite-dimensional networks with explicit stability certificates is complementary to our operator-level treatment of field-valued states on variable geometries. NEURONCTRL differs in that it must generalize across *variable neuronal morphologies* and must reason over *field constraints* defined on irregular supports, which makes geometry-aware representation and safety enforcement first-class requirements rather than secondary implementation details. In our setting, ROM/SINDy/Koopman-style approaches typically require a shared fixed-dimensional state or library across instances, making direct application to graph-varying fields (Section 2.1; Eq. (2)) and field constraints (Eq. (7)) nontrivial. We therefore do not instantiate these families as baselines; instead, we compare against geometry-aware

surrogate/operator models and policy-learning methods under a matched evaluation setup (Section A.1). A related line addresses optimal control of oscillations and synchrony in neural population models (Salfenmoser & Obermayer, 2024); this framework provides analytic tools for population-level dynamics, whereas NEURONCTRL targets spatially resolved fields on instance-specific morphologies with learned dynamics.

### D.2.3. OPERATOR LEARNING AND GRAPH-BASED SURROGATE SIMULATORS ON IRREGULAR GEOMETRIES

Neural operators learn mappings between function spaces, aiming for discretization-invariant approximation of PDE solution operators (Li et al., 2021; Lu et al., 2021; Kovachki et al., 2023). This property aligns closely with our need to model dynamics on variable discretizations and morphologies. Physics-informed neural networks (PINNs) and related residual-based approaches also embed PDE structure during learning (Raissi et al., 2019), but they are most commonly used for forward/inverse problems and can be challenging to deploy in a fast control loop due to optimization-at-inference-time behavior.

Graph neural networks and graph-network formalisms provide another route to geometry awareness by representing the discretization as a graph and using message passing to capture local interactions (Gilmer et al., 2017; Battaglia et al., 2018; Sanchez-Gonzalez & Heess, 2018; Sanchez-Gonzalez et al., 2020; Pfaff et al., 2021). These learned simulators have achieved strong performance on mesh-based and particle-based dynamics, including regimes with varying connectivity. Yet, the majority of operator-learning and graph-simulation papers target *forward prediction* under full-state supervision. For our regime, they are necessary but not sufficient: deploying them for closed-loop neuromodulation requires (i) an *observer* that can infer hidden fields from sparse recordings in real time and (ii) a *safety mechanism* that constrains actions based on predicted field evolution. NEURONCTRL is built around this missing integration: surrogates are treated as components within a feedback system rather than as standalone predictors. Consistent with this viewpoint, we include representative operator/graph surrogate families as baselines (Section A.1) and evaluate their closed-loop implications under identical protocols (Section 3.3), including safety and robustness (Section 3.2).

### D.2.4. CLOSED-LOOP CONTROL UNDER PARTIAL OBSERVABILITY AND LEARNED SIMULATORS

The POMDP formalism captures sparse sensing by introducing a latent state $x_t$ (here, a discretization-dependent field) and an observation model $y_t \sim p(y_t \mid x_t)$ (Kaelbling et al., 1998). In model-based RL, learned latent simulators replace the unknown transition with a parametric predictor in a fixed-dimensional latent space (Sutton, 1991; Ha & Schmidhuber, 2018; Hafner et al., 2019; 2020; Chua et al., 2018; Schrittwieser et al., 2020). Such methods are attractive for data efficiency and for planning by latent imagination, and they share our requirement of amortized inference under partial observability.

However, standard latent simulators make an implicit assumption that the relevant state can be compressed into a *fixed-dimensional* vector whose semantics do not change across instances. This clashes with neuronal morphologies where the discretization and topology vary and where constraints are naturally posed in the original field space. Recurrent policies and value functions can summarize history without an explicit model (Hausknecht & Stone, 2015), but they inherit the same limitation: their internal state is not an operator on function spaces and thus does not naturally encode mesh/morphology structure. In contrast, NEURONCTRL is designed around *operator- or graph-structured* representations so that inference and prediction remain meaningful when the underlying discretization changes. Accordingly, we do not instantiate Dreamer/PETS-style latent-simulator RL baselines here: reconciling a fixed-dimensional latent state with variable graphs $\mathcal{G}$ (Section 2.1) would require additional alignment and design choices that are outside our evaluation scope. We instead compare against model-free policies and geometry-aware surrogate/operator baselines under the same observation interface.

### D.2.5. GENERATIVE CONTROL AND LONG-HORIZON PLANNING UNDER REAL-TIME CONSTRAINTS

A long-standing perspective views control as probabilistic inference, linking optimal control to trajectory distributions and enabling sampling-based planning (Kappen, 2005; Todorov, 2006; Levine, 2018). Model predictive path integral (MPPI) control and cross-entropy methods operationalize this idea by repeatedly sampling control sequences, scoring them under a dynamics model, and updating the sampling distribution (Williams et al., 2016; Rubinstein & Kroese, 2004). These approaches highlight an important trade-off that also governs modern generative policies: sampling-based controllers are flexible and can represent multi-modality, but their online compute depends critically on the number of samples and model evaluations.

Recent generative models have been adapted to planning and control. Transformer-based sequence models (e.g., decision- or

trajectory-transformer style) model return- or goal-conditioned action sequences and can be used for long-horizon decision making (Lai et al., 2021; Janner et al., 2021). Diffusion models provide a powerful alternative by representing trajectories as samples from a denoising process, enabling multi-modal plan generation and strong offline imitation/RL performance (Ho et al., 2020; Janner et al., 2022; Chi et al., 2025). Their main obstacle in tight control loops is inference latency: iterative denoising requires multiple network evaluations per control update, and each evaluation may itself be expensive when the state is represented on irregular geometries. Flow-based models and continuous normalizing flows offer deterministic sampling via ODE integration with explicit control over function evaluations (Dinh et al., 2017; Chen et al., 2018; Grathwohl et al., 2019). More recent objectives such as flow matching and rectified/consistency-style training aim to reduce sampling steps and improve stability (Lipman et al., 2023; Liu et al., 2023; Song et al., 2023). These developments are directly relevant to NEURONCTRL's target regime: they suggest a principled pathway to *generative policies whose sampling cost is compatible with closed-loop actuation*, especially when combined with discretization-aware state representations. A representative line in this family is DiffPhyCon and its closed-loop variant CL-DiffPhyCon (Wei et al., 2024; 2025), which use DDPM-based diffusion models to jointly model state–action distributions for PDE control. These methods learn $p(\text{state}, \text{action})$ from offline trajectories where actions are generated via random sampling (1D Burgers) or heuristic expert strategies (2D smoke), and at inference time they rely on *classifier guidance* to steer the denoising process toward trajectories satisfying control objectives. Specifically, a guidance gradient $\nabla_x J(x)$ is added at each DDPM sampling step to encourage target tracking or energy minimization. While effective for long-horizon open-loop planning, this guidance-based approach introduces additional hyperparameters (guidance scale schedules, energy weights) and incurs per-step gradient computation during inference. In contrast, NEURONCTRL adopts a fundamentally different paradigm: we directly learn a *conditional policy* $\pi(a \mid s, g, \lambda)$ via flow matching, where goal $g$ and preference weights $\lambda$ are encoded as conditioning inputs during training. At inference time, action generation proceeds by integrating the learned velocity field from noise to action *without any external guidance*, yielding a pure ODE integration with fixed computational cost per step. Safety constraints are enforced through Flow Matching Barrier Functions (FMBF) that project the velocity field onto the tangent cone of the safe set at each integration step, achieving "safe-by-construction" generation rather than post-hoc constraint satisfaction. This design eliminates guidance-related tuning and provides explicit online safety enforcement compatible with real-time actuation.

In NEURONCTRL, we adopt a flow-based policy with optional distillation to reduce function evaluations at test time (Section 2); empirically, we compare against diffusion/flow-style planners and report latency–performance trade-offs via end-to-end rollout timing (Section 3.2; Table 2).

### D.2.6. SAFETY FOR LEARNING-BASED CONTROL OF FIELD-VALUED SYSTEMS

Safety in learning and control has been pursued via robust MPC, constrained optimization, and explicit runtime shielding (Rawlings & Mayne, 2009; Garcıa & Fernández, 2015; Alshiekh et al., 2018). Control barrier functions (CBFs) provide a particularly deployment-oriented mechanism: given a safe set $\mathcal{S} = \{x : h(x) \geq 0\}$, one restricts controls to keep trajectories within $\mathcal{S}$ (Ames et al., 2019). Yang et al. (Yang et al., 2022) propose differentiable safe controller design through control barrier functions, enabling end-to-end learning of CBF parameters; NEURONCTRL similarly exploits differentiability but learns a barrier *functional* over graph-structured fields to handle variable discretizations and morphology-dependent constraints. Predictive safety filters extend this idea by using a model to certify constraint satisfaction over a horizon while minimally modifying a nominal controller (Wabersich, 2021). Constrained policy optimization and Lagrangian methods enforce expected constraint satisfaction during learning (Achiam et al., 2017), and uncertainty-aware safe RL methods incorporate model uncertainty to avoid unsafe exploration (Berkenkamp & Turchetta, 2017).

Robust MPC and tube-MPC variants provide set-based guarantees under bounded disturbances by optimizing a nominal trajectory while maintaining an invariant "tube" that bounds state deviations (Rawlings & Mayne, 2009). Recent data-driven variants learn uncertainty sets or distill robust MPC controllers into neural policies for real-time deployment, while retaining tube-style constraint handling (Mishra et al., 2023; Tagliabue & How, 2023). These approaches are complementary to our regime: they typically operate on low-dimensional state vectors with analytically specified constraints, whereas our constraints are field-defined on irregular, instance-dependent geometries and must be enforced through a geometry-aware surrogate operator in a tight feedback loop.

Separately, certified or calibrated safety under model uncertainty has been advanced by combining barrier/Lyapunov certificates with explicit uncertainty quantification. Approximate predictive control barrier functions reduce the online safety-filter burden by replacing horizon-level certification with a single function evaluation (Didier & Zeilinger, 2024). Conformal prediction has also been used to obtain distribution-free, data-driven uncertainty radii and to conformally robustify

barrier conditions, yielding statistical safety guarantees without assuming a parametric error distribution (Hsu & Tsukamoto, 2025). Our implementation does not claim such certificates; instead, we report empirical sensitivity to controlled dynamics mismatch (Appendix A.9) and provide a simple margin condition under bounded mismatch (Proposition C.16) that can be combined with calibrated uncertainty bounds.

Goyal and Duggirala (Goyal & Duggirala, 2020) use neural networks to explore the state space of closed-loop control systems, relating to learned barrier landscapes and safety diagnostics; our setting instead enforces field-level constraints via a geometry-aware surrogate and a minimal-intervention filter. The gap for our setting is the *operational meaning* of safety when constraints are defined on fields over irregular supports. Many safe RL methods treat safety as an expectation constraint or a penalty, which does not guarantee online enforceability. Conversely, CBFs and safety filters offer online enforceability but usually assume access to a reliable model and a state representation in which $h(x)$ and its evolution can be evaluated cheaply. NEURONCTRL targets precisely this integration: safety constraints are posed in the mechanistic field space (e.g., spatial bounds on voltage or concentration), but are enforced through a geometry-aware predictive model and an online filter compatible with sparse sensing and control-loop latency. We operationalize online enforceability through a learned barrier functional and a minimal-intervention safety filter (Section 2.2.4; Eq. (17)–(19)). In experiments, we report violation frequency and magnitude under clean and shifted specifications (Section 3.2; Table 2; Table 4).

*Table 31.* Positioning of NEURONCTRL relative to closely related families. "Directly usable" refers to applicability under variable discretizations, partial observability, and deployable online safety enforcement.

| Method family | Representative references | Key assumption / limitation in our regime | Typical online cost | Directly usable? |
|---|---|---|---|---|
| Classical PDE feedback (LQR/backstepping) | (Curtain & Zwart, 2012; Bensoussan et al., 2007; Krstic & Smyshlyaev, 2008) | Known operator on fixed domain; redesign per morphology; limited handling of sparse sensing + complex constraints | low–moderate | no |
| PDE-constrained MPC | (Lions, 1972; Hinze & Vierling, 2009; Rawlings & Mayne, 2009) | Requires repeated PDE/adjoint solves; heavy for stiff hybrid neuron models | high | no |
| ROM-based control/MPC | (Antoulas, 2005; Holmes et al., 2012; Benner et al., 2015) | ROM transfer across variable morphologies is nontrivial; safety on reduced coordinates can be brittle | low–moderate | partially |
| Neural operators / graph simulators | (Li et al., 2021; Lu et al., 2021; Pfaff et al., 2021) | Often trained under full-state supervision; need observer + safety layer for deployment | moderate | yes (as simulators) |
| Latent simulator RL | (Hafner et al., 2020; Chua et al., 2018; Schrittwieser et al., 2020) | Fixed-dimensional latent is not discretization/morphology-aware; safety usually not online enforceable | moderate | partially |
| Generative planning/policies (diffusion/transformers/flows) | (Lai et al., 2021; Janner et al., 2022; Lipman et al., 2023) | Strong long-horizon modeling; sampling latency can be prohibitive without step reduction | moderate–high | partially |
| Safety filters / CBF / shielding | (Ames et al., 2019; Alshiekh et al., 2018; Wabersich, 2021) | Need a predictive model and a state representation to evaluate field constraints efficiently | low–moderate | yes (as filters) |
| NEURONCTRL (ours) | — | Geometry-aware simulator + sparse-sensing observer + online safety enforcement | low–moderate | yes |

# E. Limitations & Future Works

## E.1. Limitations

Our study has several limitations that constrain the scope of claims. **(i) Reliance on a learned surrogate; scope of safety.** We explicitly state that NEURONCTRL's safety guarantees are *relative to the surrogate and the learned barrier* and do not directly translate to physical or true-system safety without additional validation. Our safety mechanism depends on predictions from a learned surrogate simulator, and is therefore only as reliable as the surrogate under the encountered regime. Distribution shifts in morphology, boundary conditions, ionic kinetics, or unmodeled couplings can degrade accuracy, while multi-step rollouts may accumulate error and misestimate constraint satisfaction. The observer and operator formulation improve robustness to partial observability and discretization changes, but they do not remove model mismatch. **(ii) Constraint specification and conservatism.** Field-level constraints require selecting thresholds and barrier parameters that may be uncertain, context dependent, or patient specific. Overly conservative choices can degrade performance through frequent intervention, whereas permissive choices increase risk. In addition, the discrete-time barrier condition we enforce

(Eq. (16), Eq. (17)) is a sufficient (and potentially conservative) approximation to the underlying continuous-time safe set. **(iii) Sensing realism and partial observability.** We assume sparse aggregated observations; in practice, sensing can suffer from nonstationary noise, drift, delays, missing data, and stimulation-correlated artifacts, which can impair field reconstruction and planning. **(iv) Real-time computation.** While distillation can reduce inference latency, differentiable MPC may remain computationally demanding for large morphologies, high control rates, long horizons, or large action spaces (Section 3.2; Table 2). **(v) External validity beyond simulation.** Even high-fidelity biophysical simulators are imperfect abstractions. Accordingly, our results demonstrate capability in a controlled simulation setting and should not be interpreted as evidence of clinical efficacy.

## E.2. Future Works

First, *uncertainty-aware safety* remains essential: equipping the surrogate and observer with calibrated epistemic uncertainty (e.g., ensembles, Bayesian neural operators, or conformal methods) could make safety filtering more robust to model error by enforcing probabilistic margins and restricting actions when uncertainty is high (Section 3.2). Second, we aim to strengthen *robustness under distribution shift* by coupling operator learning with stability- or passivity-inspired regularization and deriving error bounds that explicitly track morphology and discretization perturbations, narrowing the gap between surrogate-level barrier guarantees and true-system safety (Section 3.2; Appendix A). Third, bridging simulation to practice will require *system identification and domain adaptation*, including subject-specific parameter inference, adaptation to electrode placement and tissue heterogeneity, and physics-informed priors. Fourth, we plan to build *virtual cell* testbeds that combine differentiable multi-physics simulation with closed-loop control to enable scalable, reproducible evaluation under realistic sensing/actuation effects (artifacts, delays, saturation, and multi-electrode interference). Finally, we will expand *benchmarks and preference interfaces*, covering larger morphologies and richer biochemical pathways, and developing clinician-in-the-loop preference elicitation with safeguards against unsafe trade-offs.

