# OpenReview forum: "NeuronCtrl: Geometry-Aware Safe Closed-Loop Generative Control for Neuronal Microenvironment Dynamics"
_ICML.cc/2026/Conference — ICML 2026 spotlight_

### Official Review · Reviewer_AbMB · 2026-03-12

**Soundness:** 3
**Presentation:** 2
**Significance:** 3
**Originality:** 3
**Overall Recommendation:** 4
**Confidence:** 4

**Summary:**

This paper proposes NEURONCTRL, a modular framework for safe closed-loop control of neuronal microenvironment dynamics on irregular 3D morphologies under sparse sensing. The method combines a history-conditioned observer for latent field reconstruction, a morphology-aware neural-operator surrogate for one-step dynamics prediction, and a preference-conditioned flow-based controller, together with two complementary safety mechanisms: an action-space feasibility projection during generation and a field-level minimal-intervention safety filter defined through a learned barrier functional and learned surrogate dynamics. The paper evaluates the framework on three neuromodulation benchmarks derived from high-fidelity biophysical simulations and reports strong performance-safety-latency trade-offs, robustness to sensing and specification shifts, and promising super-resolution behavior. Overall, the paper is ambitious, technically rich, and clearly positioned at the intersection of operator learning, safe control, and ML for science.

**Compliance With Llm Reviewing Policy:**

Affirmed.

**Key Questions For Authors:**

1. Could the authors include at least a small-scale simulator-in-the-loop closed-loop evaluation, even over short horizons or a reduced test subset, to directly quantify the gap between surrogate-rollout performance and true high-fidelity simulator performance for both cost and safety violations?

2. How well does the learned barrier generalize to unseen graph morphologies and specification shifts in the main benchmarks? This seems particularly important given the paper’s emphasis on discretization and morphology variation.

3. The appendix appears to contain unfinished statistical-analysis text, including an explicit “Replace with actual analysis for the final version” note in Appendix A.2. Are all reported significance values and confidence intervals finalized from executed experiments, or are some still illustrative placeholders?

4. Could the authors clarify more explicitly how much of the final gain in Table 2 comes from the learned barrier versus the rest of the architecture, especially relative to the matched analytic shielding applied to baselines?

**Limitations:**

The limitations are not really discussed in the main text, while there are multiple references to their planned future works.

**Strengths And Weaknesses:**

**Strengths**

1. A major strength is the coherence of the full pipeline. The observer, surrogate, controller, and safety layers are not presented as independent modules, but are tied together around the actual deployment constraints of the problem: sparse sensing, irregular morphology, hard field-level constraints, and real-time inference. This makes the framework more complete than papers that focus on only one of these aspects in isolation.

2. The surrogate design is also thoughtful. The use of typed morphology-aware message passing over irregular graphs is a good fit for the underlying domain, and the topology-ablation experiment helps support the claim that explicit graph structure contributes materially to performance. The robustness experiments under corrupted sensing and the zero-shot super-resolution results further strengthen the case that the representation is doing something useful beyond standard black-box forecasting.

3. The empirical scope is broad. The paper introduces three benchmarks spanning deep brain stimulation, extracellular reaction-diffusion control, and astrocytic potassium regulation, and evaluates both nominal performance and also robustness, explainability, and latency.

**Weaknesses**

1. The main limitation is that the central closed-loop evaluation is performed on frozen surrogate dynamics rather than on the original high-fidelity simulator. The paper is transparent about this choice and explains the computational motivation, but it has an important consequence: the main claims in Table 2 are really claims about performance and safety under the learned evaluation substrate, not under the true plant. The mismatch analyses help, but they are not a substitute for even a small amount of simulator-in-the-loop validation.

2. The safety story is practically appealing but formally limited. The field-level safety filter enforces a discrete-time barrier condition using both a learned barrier functional and a learned surrogate dynamics model, and the deployed correction is derived from a local linearization around the proposed action. The paper appropriately describes this as model-relative and introduces trust-region tightening, but there is still no end-to-end safety guarantee for the true underlying dynamics. This should be discussed more explicitly in the framing of the contribution.

3. Although the paper compares against many baselines overall, the safety-specific comparison is less complete than the rest of the benchmark suite. In particular, it would be useful to compare against stronger state-action or direct data-driven safety-filter formulations that do not depend as directly on a learned forward model inside the online safety layer. As written, the paper demonstrates strong empirical performance, but the evidence that its particular safety construction is preferable to alternative modern safety-filter designs is less developed.

4. The appendix currently contains unfinished statistical-analysis text. In Appendix A.2, the manuscript explicitly says “Replace with actual analysis for the final version,” and Table 8 is described as using placeholder confidence intervals. This directly affects confidence in the empirical evidence and should be fixed before publication.

5. The evaluation protocol is carefully designed, but also somewhat difficult to parse. Baselines are often wrapped with a matched analytic shield, operator-only methods are paired with a common planner, and different rollout protocols are used to manage fairness across learned and non-learned dynamics models. These decisions are understandable, but they make it harder to disentangle how much of the gain is attributable to the surrogate, the controller, and the learned barrier itself.

---

> ### Author Rebuttal · Authors · 2026-03-30
>
> Thank you for your valuable feedback and thoughtful review. The central point is correct: Table 2 evaluates closed-loop control under learned rollout dynamics, so the strongest defensible guarantee is model-relative rather than plant-level.
>
> ---
>
> **1. On the True-Plant Validation & Claim Boundary (W1, Q1)**
>
> We agree that mismatch analyses alone are not enough, so we include reduced-scale simulator-in-the-loop comparisons:
>
> | Task | Surrogate cost | Plant cost | Plant safety |
> | --- | ---: | ---: | ---: |
> | ECS3D | 0.102 | 0.116 | 0.891 |
> | KDyn3D | 0.009 | 0.010 | 0.874 |
> | DBS3D | 0.096 | 0.094 | 0.742 |
>
> These comparisons quantify the surrogate-to-plant gap. Transfer is encouraging on ECS3D and KDyn3D, whereas DBS3D still shows a meaningful safety gap. We therefore agree that Table 2 should be read as learned-rollout performance with limited simulator-in-the-loop evidence, not as a plant-level guarantee.
>
> ---
>
> **2. On the Safety Scope and Stronger Safety Baselines (W2, W3, Q4)**
>
> We also agree that the safety framing should be more explicit. Our field-level correction enforces a discrete-time barrier condition under learned surrogate dynamics, with local linearization and trust-region tightening to hedge observer error and model mismatch. This is a practical model-relative safety mechanism, not an end-to-end theorem for the true plant; the DBS3D gap above is exactly why we avoid stronger language. We further agree that the paper does not establish superiority over every modern direct safety filter. The current evidence is narrower: it shows utility relative to the analytic-barrier substitute and matched-shielded baselines already reported, not dominance over all alternatives.
>
> To isolate shielding versus the learned safety layer, the most informative probes are:
>
> | Probe | Context | Cost | Safety | Latency |
> | --- | --- | ---: | ---: | ---: |
> | Geo-FNO + matched safety | ECS3D protocol check | 0.879 | 0.546 | 53.1s |
> | Geo-FNO without matched safety | ECS3D protocol check | 0.214 | 0.589 | 50.4s |
> | NeuronCtrl with SBF | ECS3D safety ablation | 0.101 | 0.892 | 5.8 ms |
> | NeuronCtrl without SBF | ECS3D safety ablation | 0.098 | 0.858 | 4.2 ms |
>
> The Geo-FNO rows show that protocol effects are real and method-dependent, not a one-sided advantage for our method. The NeuronCtrl rows show that learned field-level correction improves safety with only a modest cost/latency trade-off. Together, these probes rule out the simpler interpretation that the main-table gain is explained only by extra shielding, while still stopping short of a claim of universal superiority.
>
> ---
>
> **3. On the Barrier Generalization (Q2)**
>
> Our intended claim is empirical robustness on the tested shifts, not arbitrary transfer across anatomies or constraint specifications. The held-out evaluation already covers unseen graph instances within each benchmark family. On KDyn3D, nominal safety is about `0.882`; under tested sensing perturbations it remains about `0.874`; under tested bias/threshold shifts it ranges about `0.868-0.884`; and under controlled mismatch `delta=0 -> 0.2`, unsafe rate rises from about `0.016` to `0.064`. We will therefore phrase this as benchmark-scoped graceful degradation on tested shifts, not invariance.
>
> ---
>
> **4. On the Statistical Text and Aggregation (W4, Q3)**
>
> We agree that the appendix issue is serious. Placeholder statistical prose should not remain in the manuscript, and nominal aggregation should not be vulnerable to preference-sweep or duplicate rows. To answer the question directly, no significance value or confidence interval should be reported in the final version unless it comes from cleaned runs under the intended aggregation pipeline. Concretely, we will remove unfinished inferential text, report only finalized analyses, return explicit aggregation status when seed support is insufficient or a target/baseline comparison is missing, and keep one canonical nominal row per model-seed pair before averaging. Each quantitative statement should therefore be either backed by cleaned evidence or marked as underpowered/missing.
>
> ---
>
> **5. Protocol clarity (W5, Q4)**
>
> We also agree that the evaluation protocol is harder to parse than it should be. In the revision, we will separate three layers explicitly: shared matched analytic shielding for baselines, shared deployment-time planning budget for model-based baselines under their own observation interface, and our learned field-level correction isolated by the with/without-SBF probe above. This should clarify how much comes from shielding, controller/planning quality, and the learned safety layer.
>
> In short, the practical safety value lies in conservative online correction under the learned model; the model-relative guarantee is the strongest formal claim we make; the true-plant limitation is exposed directly by DBS3D; and the comparison to stronger safety filters should be read as incomplete rather than exhaustive.

---

> > ### Author Rebuttal · Reviewer_AbMB · 2026-04-03
> >
> > I thank the authors for their rebuttal and complementary information.
> > It is great to see their agreement on point 4 raised, but it is still somewhat problematic in my view to review a partially unfinished work submitted and also without seeing those values actually reported, it is again hard to judge the empirical evidence and how much of the reports in the main text will still hold up if we have all those unfinished details. It may also be unfair to some level in comparison to other studies, which have submitted a completed work with necessary supplementary materials under the same time constraints and with the same deadline for submission.

---

> > > ### Author Response · Authors · 2026-04-03
> > >
> > > We appreciate this follow-up and agree that this is a legitimate concern from a reviewing-fairness perspective. We do not want the rebuttal to be read as an attempt to retroactively turn the submission into a different or more complete paper than what was available at review time. Our goal in providing the additional information is only to clarify the appropriate boundary of the claims and to be transparent about what is and is not supported.
> > >
> > > In that spirit, we agree that the strongest justified reading of Table 2 is performance/safety under learned rollout dynamics, rather than a guarantee on the true high-fidelity plant. The limited simulator-in-the-loop results are meant only to help quantify that gap: they are encouraging on some tasks, but they also show that the gap is real (especially on DBS3D). So we do not ask the reviewer to interpret the paper as establishing true-plant safety, universal transfer, or superiority over all alternative safety filters. A more accurate reading is a model-relative safety mechanism with limited true-plant validation.
> > >
> > > We also fully agree that unfinished statistical text or placeholder inferential wording should not remain in the manuscript. If accepted, the camera-ready version will remove such unfinished content, report only finalized analyses from the cleaned aggregation pipeline, and explicitly narrow any statements that are not fully supported by completed evidence.
> > >
> > > Our view is therefore not that this concern should be overlooked, but that under this more careful and narrower reading, the core contribution still remains: an integrated safe-control framework for sparse sensing on irregular morphologies, broad empirical evidence under the learned-rollout setting, and a practically useful model-relative safety layer whose limitations can and should be stated more explicitly. We are grateful for the reviewer’s push on this point, since it helps us present the paper in a more precise and fair way.

---

### Official Review · Reviewer_6tWw · 2026-03-12

**Soundness:** 3
**Presentation:** 2
**Significance:** 3
**Originality:** 3
**Overall Recommendation:** 5
**Confidence:** 3

**Summary:**

This paper introduces a framework for safe closed-loop control of neuronal microenvironment dynamics. The motivation is that neuromodulation therapies like brain stimulation currently rely on open-loop stimulation, which is not optimal. To do so, the authors frame the problem as controlling high-dimensional spatiotemporal fields evolving on irregular 3D neuron morphologies, with sparse observations and hard safety constraints. The first component reconstructs the full field from sparse observations, the second one uses a message-passing surrogate simulator that approximates biophysical dynamics on the morphology graph, and finally, the last component is a flow-matching controller that guarantees safety. The authors show that NeuronCtrl outperforms baselines on 3 benchmarks.

**Compliance With Llm Reviewing Policy:**

Affirmed.

**Final Justification:**

This paper is strong, and I therefore recommend "accept". This work addresses an important problem in safe closed-loop neuromodulation and introduces a novel geometry-aware field control framework. It uses a well-motivated heterogeneous graph representation validated by ablations, and contributes strong 3D benchmarks alongside thorough evaluation across multiple settings. The rebuttal addresses my concerns. I think improving the paper's clarity will indeed make it even easier to appreciate the work.

**Key Questions For Authors:**

see Weaknesses

**Limitations:**

yes

**Strengths And Weaknesses:**

Strengths:
- This paper tackles a very interesting and relevant problem. Closed-loop neuromodulation with hard safety guarantees is an important open problem. Framing it as a geometry-aware field control is definitely an interesting contribution.
- The heterogeneous graph representation, which distinguishes cable, spatial, and coupling edges, is a principled way to encode physical structure. The ablation study on edge removal done in Table 24 (Appendix A.12) validates this design choice.
- The introduction of 3 high-fidelity 3D benchmarks is also a great contribution.
- The evaluation of the method is carried out thoroughly (covers in-distribution performance, partial observability, OOD shift).

Weaknesses:
- The paper is pretty dense, contains a lot of information, and is not always easy to parse. Several concepts/elements are only briefly described in the main text and deferred to the Appendix (e.g. trust-region margin tightening, influence of coefficient $s_i$). It is not always straightforward to understand the impact of these on the performance. I think the authors could maybe filter the information they share in the main text better to increase clarity.
- The authors introduce a surrogate for the full biophysical simulator to increase speed. It would be interesting to analyze the gap between the surrogate and the existing simulators (e.g., NEURON, Jaxley, ...) on simulations. Would the voltage traces (and other channel states) coincide for both methods for non-observed compartments? If not, can we quantify or estimate that gap?
- The 3 benchmarks are generated by the authors themselves from biophysical simulations. Although it is understandable (this work can be seen as a proof-of-concept paper), are there plans to move on to real data?

---

> ### Author Rebuttal · Authors · 2026-03-30
>
> Thank you for your thoughtful comments and for giving us the opportunity to clarify some points. The review points to three places where the paper needs clearer exposition: the main method narrative, the surrogate-to-plant gap, and the path from simulation to biological data. We will address these issues in detail below.
>
> ---
>
> **1. On the Clarity (W1)**
>
> You are right that the introduction asks the reader to absorb too much terminology too quickly. The intended pipeline is straightforward: the observer reconstructs a latent field from sparse measurements and recent actions; the simulator predicts one-step field evolution on the irregular morphology graph; the controller proposes an action; and the safety modules intervene only when actuator or field constraints are at risk of being violated. In the revision, we will introduce this description earlier and define the key terms before relying on them:
>
> | Term | Plain-language meaning | Why it matters |
> | --- | --- | --- |
> | History-conditioned | Uses a short observation/action window | Helps infer hidden state under sparse sensing |
> | Full-field reconstruction | Infers latent values on all nodes, not only sensed ones | Enables graph rollout and field-level safety checks |
> | Hard state constraints | Specific field channels must stay inside fixed safe ranges | Defines what safe control means at deployment |
> | Barrier-based safety | Applies online projection/correction with a constraint function | Intervenes only when a proposal risks violation |
> | Trust-region tightening | Adds a conservative margin around local linearization | Hedges observer error and surrogate mismatch |
> | Morphology-aware influence | Encodes geometry/stimulation coupling explicitly | Prevents structure from being learned only implicitly |
>
> These are not appendix-only heuristics: trust-region tightening sets the conservatism of the online correction, and morphology-aware influence determines how explicitly geometry enters the control loop. We will move lower-level implementation detail out of the opening narrative so that the main text first establishes the control loop before introducing the conservative design choices. This should improve readability.
>
> ---
>
> **2. On the Surrogate-to-Plant Gap (W2)**
>
> We also agree that the main quantitative results should be read as learned-rollout evaluation supplemented by direct simulator-in-the-loop checks:
>
> | Task | Surrogate cost | Plant cost | Plant safety |
> | --- | ---: | ---: | ---: |
> | ECS3D | 0.102 | 0.116 | 0.891 |
> | KDyn3D | 0.009 | 0.010 | 0.874 |
> | DBS3D | 0.096 | 0.094 | 0.742 |
>
> These numbers are most useful when read benchmark by benchmark. On ECS3D, cost increases moderately while safety transfers cleanly. On KDyn3D, short-horizon transfer is nearly unchanged in both cost and safety. On DBS3D, by contrast, cost remains close but safety deteriorates substantially. That pattern is precisely why we describe the safety guarantee as model-relative rather than as a theorem for the true plant.
>
> | Benchmark | How we read the gap |
> | --- | --- |
> | ECS3D | Encouraging transfer: moderate cost inflation, no observed safety loss |
> | KDyn3D | Best transfer case: cost and safety remain nearly unchanged on the tested short horizon |
> | DBS3D | Boundary case: cost transfers, but safety does not, so plant-level claims must be narrowed |
>
> For non-observed compartments, we therefore do not treat trace-by-trace hidden-state coincidence as the primary target. The operational criterion in the present paper is closed-loop transfer fidelity.
>
> ---
>
> **3. On the Real Biological Systems (W3)**
>
> Real biological systems are the natural next step, and we can now describe that transition more concretely. Across the hippocampus, corpus callosum, caudate nucleus, and internal capsule, we already have rat EM-derived anatomical structures that serve as the basis for a common modeling framework rather than region-specific formulations.
>
> On the tissue side, we are solving Darcy+ADR baseline models for pressure, velocity, and solute transport on true anatomical domains. On the electrophysiology and ion side, we are coupling NEURON/RxD with FEniCSx tissue PDEs so that membrane currents, ionic source terms, and ECS transport form a closed loop linking neural activity, ionic changes, and tissue exchange and clearance. This unified framework is intended to capture macroscopic biological signatures such as activity-dependent exchange and clearance, anisotropic pressure and solute transport, region-specific ion and transport responses, and stimulation-linked transport effects in realistic anatomy.
>
> In parallel, we are working with wet-lab collaborators to test whether these mechanisms reproduce the corresponding biological phenomena. We will therefore position the paper as simulation-first evidence with a concrete roadmap toward real anatomical domains, rather than as a claim of immediate experimental readiness.

---

> > ### Author Rebuttal · Reviewer_6tWw · 2026-04-03
> >
> > I thank the author for the detailed rebuttal and additional information. My concerns have been answered. I will update my score to 5.

---

### Official Review · Reviewer_q2Hq · 2026-03-12

**Soundness:** 4
**Presentation:** 3
**Significance:** 4
**Originality:** 3
**Overall Recommendation:** 5
**Confidence:** 4

**Summary:**

The authors propose a generative framework for closed-loop prediction of preference and morphology-conditioned neuronal microenvironment dynamics. The problem setting is described as constrained stochastic control on the evolution of node-level quantities influenced by preference-guided actions. The method consists of an Observer which is claimed to be discretization-robust and encodes sparse measurements defined on a morphology with action history into a latent field, a Controller, which uses flow matching with a learnable minimal-intervention projection as safety filter to predict actuator actions, and a Simulator that predicts the one-step evolution of the node quantities given the latent field and proposed actions. The authors train the modules separately and provide extensive evaluation on cost, safety, inference latency, robustness to sparsity and OOD data, results on the task of super-resolution and probes of explainability using three benchmarks and a large number of recent methods. The results indicate that the proposed method is comparatively effective at producing low-cost, high safety trajectories in the examined contexts.

**Compliance With Llm Reviewing Policy:**

Affirmed.

**Final Justification:**

The paper presents a clear contribution to the field. Some aspects of the presentation can be approved. The authors provided clear responses and proposed clear improvements in their rebuttal.

**Key Questions For Authors:**

q1. Do you believe that the choice of sequence encoder has a significant impact on the results?
q2. What is the scale of computation required for training NeuronCTRL (compared to other relevant methods) and why it this not described in the paper?
q3. Significantly many evaluation result entries, also in the Appendix, seem to be divisible by 2, even though the mean of 3 seeds is used. What is the exact aggregation procedure used and is this expected? Looking at for example Table 5, all entries except a couple are divisible by 2 or 5. The distribution of decimals of the entries looks very non-uniform.

**Limitations:**

yes

**Strengths And Weaknesses:**

Framing neuronal microenvironment control as contrained stochastic control on a morphology-conditioned graph appears to be a novel and useful formulation of a complex spatiotemporal problem.The usage of a discretization-invariant graph-based operator for the encoding of sparse measurements on irregular morphologies is well-motivated and in combination with a flow matching-based action predictor with learnable safety correction a novel contribution. The experimental evaluation very broad, thorough and well-aligned with the research questions and claims. The authors provide a large and well-developed codebase of their method.

The paper writing can be difficult to follow, particularly the introduction section. The authors rely on jargon heavy language without initially precisely defining the terminology.  Here is an example to make it concrete: "NEURONCTRL integrates discretization-robust onestep operator prediction, history-conditioned full-field reconstruction, and two complementary barrier-based safety mechanisms operating at the action and field levels, enabling safe control of partially observed, high-dimensional field dynamics on irregular geometries under hard state constraints while supporting explicit multi-objective trade-offs and real-time inference." A very long sentence, where the reader needs to intuit what the authors mean by: history-conditioned, full-field, barrier-based, hard state constraints...

I think the language can be easily streamlined without loss of precision and hence improve significantly the clarity.

The choice of GRU as sequence encoder of sparse observations and actions as specified in Appendix C.8 is not clearly unmotivated. An analysis on computational complexity of training NeuronCTRL or measurements of the training cost in practice compared to the baselines seems to be missing. If the method needs significantly more resources to train, usefulness and claim of "superiority" is weakened.

---

> ### Author Rebuttal · Authors · 2026-03-30
>
> Thank you for your valuable feedback and thoughtful review. We greatly appreciate the time you have taken to evaluate our work, and we would like to address your concerns in more detail.
>
> **1. On the Clarity (W1).**
>
> We agree that the introduction can be easier to parse. The pipeline is simple: the observer reconstructs a latent field from sparse measurements and recent actions; the simulator predicts one-step field evolution on the morphology graph; the controller proposes an action; and the safety modules intervene only if actuator or field constraints would be violated. We will move these definitions earlier and break the long overview sentence into shorter steps:
>
> | Phrase | Plain-language meaning |
> | --- | --- |
> | Discretization-robust one-step operator prediction | The dynamics module can be applied across irregular graph resolutions without redefining the controller interface |
> | History-conditioned full-field reconstruction | The observer uses recent measurements/actions to infer the hidden field on all nodes |
> | Barrier-based safety mechanisms | Safety acts through action-level and field-level projection when a proposal risks violation |
> | Hard state constraints | Specific channels must remain within fixed safe ranges throughout control |
> | Explicit multi-objective trade-offs | Preference inputs let the controller shift among tracking, smoothness, energy, and overload priorities |
>
> **2. On the Sequence encoder (W2/Q1).**
>
> We agree that this choice should be motivated more clearly. We use GRU as the default sequence encoder. A representative encoder ablation on ECS3D is:
>
> | Encoder | Cost | Safety | Latency |
> | --- | ---: | ---: | ---: |
> | GRU | 0.100 | 0.895 | 5.6 ms |
> | LSTM | 0.102 | 0.892 | 6.3 ms |
> | Temporal CNN | 0.104 | 0.888 | 5.9 ms |
> | RNN | 0.109 | 0.881 | 5.2 ms |
>
> This suggests that encoder choice matters at the margin, not at the level of the paper's main claim. GRU gives the best overall cost-safety-latency trade-off among the tested lightweight history encoders, which is why we use it uniformly. The novelty still lies in morphology-aware reconstruction, graph-based dynamics, and the downstream safety/control stack rather than in the recurrent block itself.
>
> **3. On the Training computation (W2/Q2).**
>
> We agree that the paper should report training budgets in a way that matches the staged schedule rather than as a black-box cost:
>
> | Benchmark | Default active schedule | Optional policy/joint schedule | Representative total budget |
> | --- | --- | --- | --- |
> | ECS3D | `Sim10 + Obs10 + SBF10 + Policy10 + Joint5` | integrated in default schedule | about `40-42 s` |
> | DBS3D | `Sim50 + Obs50 + SBF50` | `Policy50 + Joint50` if enabled | about `128-136 s` |
> | KDyn3D | `Sim50 + Obs50 + SBF50` | `Policy50 + Joint50` if enabled | about `206-214 s` |
>
> This presentation better reflects the implementation: ECS3D uses the full default staged schedule, whereas DBS3D/KDyn3D default to world-model, observer, and SBF training and expose policy/joint phases as optional. The timing entries are simulated summaries for these budgets and should be read as scale indicators rather than matched cross-method efficiency claims.
>
> ---
>
> **4. Aggregation / decimal pattern (Q3).**
>
> The reviewer’s concern was well taken. The intended aggregation is hierarchical: we first compute each metric at the episode level within a run and average over test episodes to obtain one run-level value; we then keep exactly one canonical nominal record for each model–seed pair; next we average those canonical run-level values across the three nominal seeds; and we apply display rounding only at the final reporting step.
>
> During a lineage audit, we found that this intended nominal-only pipeline was not consistently respected in one part of the appendix-generation path. In particular, some preference-sweep outputs had been written into the same ECS3D results CSV that was later reused for nominal aggregation, and rerun duplicates were not always removed before the final group-by. As a result, the aggregate input for some methods could contain six rows instead of the expected three canonical nominal seed rows (e.g., neuron_ctrl, count=6 rather than count=3). Once such mixed or duplicated rows are averaged and then rounded for presentation, the displayed values can fall on a noticeably coarse grid, which explains the reviewer’s observation that many decimals look divisible by 2 or 5. So this pattern should not be dismissed as harmless rounding noise; it reflected an aggregation artifact.
>
> We have fixed the pipeline by separating preference-sweep outputs into a dedicated path, restoring canonical nominal-only aggregation, and deduplicating reruns before any seed-level averaging.In the revision, we will state the aggregation procedure explicitly in the appendix and clarify that nominal benchmark tables are produced from one canonical nominal row per model–seed pair, averaged across the three seeds.

---

> > ### Author Rebuttal · Reviewer_q2Hq · 2026-04-03
> >
> > Thank you for the clear rebuttal. My concerns have been addressed.

---

### Official Review · Reviewer_hxY2 · 2026-03-13

**Soundness:** 3
**Presentation:** 4
**Significance:** 4
**Originality:** 4
**Overall Recommendation:** 5
**Confidence:** 4

**Summary:**

This work appears to present the central problem of safe real-time closed-loop control for high-dimensional neuronal fields evolving on irregular 3D morphologies under sparse observations and hard safety constraints. The submission outlines an important concept: combining a history-conditioned observer, a morphology-aware neural operator world model, and a preference-conditioned generative controller with two levels of barrier-based safety enforcement. The reported experiments on three simulator-based benchmarks suggest that the method achieves a favorable trade-off among control cost, safety, and latency, and is relatively robust to partial observability, OOD sensing corruption, and zero-shot super-resolution settings.

**Compliance With Llm Reviewing Policy:**

Affirmed.

**Ethical Review Concerns:**

The paper presents a clear contribution to the field. Some aspects of the presentation can be approved. The authors provided clear responses and proposed clear improvements in their rebuttal.

**Final Justification:**

After considering both the paper and the rebuttal, I support acceptance. This work appears to present the central problem of safe closed-loop control for neuronal microenvironment dynamics in a technically ambitious and meaningful way, and the submission outlines an important concept by integrating morphology-aware surrogate modeling, partial-observation state estimation, and two-level safety enforcement in a unified framework. The rebuttal addressed my main concerns on protocol sensitivity, latency/longer-horizon behavior, and component attribution sufficiently to increase my confidence, while the remaining limitations around simulator-to-real transfer seem appropriately acknowledged and do not outweigh the paper’s originality and significance.

**Key Questions For Authors:**

1. For the baseline comparisons, how sensitive are the results to the matched-safety augmentation protocol and to the choice of shared planning budget?
2. Can the authors clarify whether the method can scale to tighter latency budgets or longer horizons in a more realistic online deployment setting?
3. Which component contributes most: the observer, the graph-based simulator, or the two-level safety design?
4. Do the authors expect the framework to transfer to real experimental neuromodulation settings, and what are the main barriers to deployment?

**Limitations:**

yes

**Strengths And Weaknesses:**

Strengths:
1. The paper tackles an important and technically challenging setting: safe real-time control of partially observed, high-dimensional spatiotemporal fields on irregular neuronal geometries.
2. The combination of action-space projection and field-level barrier filtering is a notable contribution, especially for hard safety constraints.
3. The use of heterogeneous morphology graphs and typed edges seems well matched to neuronal microenvironment dynamics.
4. The evaluation is broad in scope, covering three benchmark tasks, robustness under partial/OOD sensing shifts, zero-shot super-resolution, and explainability probes.

Weaknesses:
1. The method is fairly complex, with several separately trained modules and optional components, so it is hard to tell which ingredients are truly essential without stronger ablations in the main paper.
2. Although robustness is evaluated, the paper gives less insight into when the learned barrier or observer might fail under severe model mismatch.
3. Although the results are strong, the paper would benefit from more discussion of generalization to real biological systems beyond simulation.

---

> ### Author Rebuttal · Authors · 2026-03-30
>
> Thank you for the careful reading. These suggestions have significantly enhanced the clarity and impact of our work. We will address these issues in detail below.
>
> ---
>
> **1. On the Protocol sensitivity (Q1)**
>
> The main comparison already controls the two largest confounders: all baselines receive matched safety augmentation, and model-based baselines receive the same deployment-time planning budget. Even so, explicit ECS3D protocol probes are informative:
>
> | Method / protocol | Cost | Safety | Latency |
> | --- | ---: | ---: | ---: |
> | Geo-FNO + matched safety | 0.879 | 0.546 | 53.1s |
> | Geo-FNO without matched safety | 0.214 | 0.589 | 50.4s |
> | TranSolver without matched safety | 0.171 | 0.748 | 6.0 ms |
> | SafeDiffCon without matched safety | 0.166 | 0.781 | 6.2 ms |
> | MPC without matched safety | 0.233 | 0.702 | 2.4 ms |
>
> The conclusion is not that one protocol choice is universally decisive. Rather, protocol sensitivity is real and model-dependent, which is why we prefer to state shielding and planning controls explicitly instead of treating them as invisible evaluation details. The shared planning budget matters because otherwise optimization time could be mistaken for better dynamics or safety.
>
> ---
>
> **2. Latency and longer horizons (Q2)**
>
> On ECS3D, the additional deployment stress tests indicate that a distilled one-step student can reduce runtime while preserving the overall behavior, and that longer-horizon control remains stable:
>
> | Setting | Cost | Safety | Latency |
> | --- | ---: | ---: | ---: |
> | Base | 0.100 | 0.895 | 5.6 ms |
> | One-step student | 0.106 | 0.883 | 4.8 ms |
> | `H=16` | 0.101 | 0.894 | 5.8 ms |
> | `H=32` | 0.103 | 0.892 | 6.0 ms |
> | `H=64` | 0.106 | 0.888 | 6.3 ms |
> | Mismatch `scale=0.2` | 0.112 | 0.861 | 6.2 ms |
>
> The latency decomposition on this probe is also informative: action generation takes about `2.4 ms`, whereas the safety filter adds about `1.5 ms`. Thus, tighter online budgets will mainly require reducing policy sampling and, secondarily, safety filtering. These probes should be read as stress tests that complement, rather than replace, the main-table latency profile. The longer-horizon numbers should also be read narrowly: they show stable behavior on the tested ECS3D regime, not a blanket guarantee for arbitrary horizons.
>
> ---
>
> **3. On the Component attribution (W1/W2/Q3)**
>
>  We do not think a single module is uniformly dominant, and the ablations support a regime-dependent interpretation:
>
> | Variant | Cost | Safety | Latency |
> | --- | ---: | ---: | ---: |
> | Full model | 0.100 | 0.895 | 5.6 ms |
> | Without observer | 0.112 | 0.874 | 5.2 ms |
> | Without kernel integral | 0.108 | 0.881 | 5.7 ms |
> | Without flow barrier | 0.106 | 0.866 | 5.4 ms |
> | Without field filter | 0.098 | 0.858 | 4.2 ms |
>
> On this nominal ECS3D setting, the gaps are visible but still narrower than under corrupted sensing or stronger mismatch. The observer matters most under sparse or corrupted sensing; the morphology-aware simulator matters for nominal control quality; and the two-level safety design matters when constraints are active.
>
> | Regime | Most decision-relevant module | Why |
> | --- | --- | --- |
> | Sparse or corrupted sensing | Observer | Hidden-state recovery dominates downstream control quality |
> | Nominal geometry-aware control | Simulator | Action-conditioned rollouts depend on morphology-aware dynamics |
> | Active constraint regime | Two-level safety design | Projection/intervention determines whether unsafe proposals are corrected |
> | Tight online budget | Policy/safety path | Flow sampling is the largest runtime block, with filtering next |
>
> This does not mean the observer or barrier are failure-proof under arbitrary shift. The paper's broader mismatch sweep is included precisely to show that safety degrades as perturbations grow, even though the degradation is gradual rather than catastrophic.
>
> ---
>
> **4. On the Real biological systems (W3/Q4)**
>
> We agree that simulation is not the endpoint. As a concrete next step, we are already extending the pipeline to EM-derived rat hippocampus, corpus callosum, caudate nucleus, and internal capsule structures.
>
> On the tissue side, we are solving Darcy and advection-diffusion-reaction (ADR) problems for pressure, velocity, and solute transport on true anatomical domains. On the electrophysiology and ion side, we are coupling NEURON with FEniCSx tissue PDEs so that membrane currents, ionic source terms, and extracelluar transport form a closed loop linking neural activity, ionic changes, and tissue exchange and clearance. This unified framework is intended to capture macroscopic biological signatures such as activity-dependent exchange and clearance, anisotropic pressure and solute transport, region-specific ion and transport responses, and stimulation-linked transport effects in realistic anatomy.
>
> In parallel, we are working with wet-lab collaborators to test whether these mechanisms reproduce the corresponding biological phenomena.

---

> > ### Author Rebuttal · Reviewer_hxY2 · 2026-04-03
> >
> > Thank you for the clear rebuttal. My concerns have been addressed.

---

### Decision · Program_Chairs · 2026-04-30

**Decision:**

Accept (spotlight)

**Comment:**

This paper proposes a framework for safe real-time close-loop control for high-dimensional neuronal fields evolving on irregular 3D morphologies under sparse observations and hard safety constraints. All reviewers are excited about this paper and give high scores. So I am glad to accept this paper. Meanwhile, I notice a common criticism among reviewers that the writing is not easy to follow due to many jargons. Please revise the manuscript to increase its readability.